# Catabolism of extracellular glutathione supplies cysteine to support tumours

Fabio Hecht[1,2,14 ✉], Marco Zocchi[1,2,14], Emily T. Tuttle[1,2], Nathan P. Ward[3], Fatemeh Alimohammadi[1,2], Amal Afzal Khan[1,4], Veronica C. Gomes[1,2], Bradley Smith[2], Jennifer J. Twardowski[2], Bradley N. Mills[2,5], Kevin A. Welle[6], Sina Ghaemmaghami[6,7], Zhuoran Zhou[1], Yuhan Gan[1], Yun Pyo Kang[3], Juliana Cazarin[1,2], Zamira G. Soares[1,2], Mete Emir Ozgurses[8], Huiping Zhao[8], Colin Sheehan[9], Guillaume Cognet[9], Lila D. Munger[1,2], Dhvani Trivedi[1,2], Gloria Asantewaa[1,2,10], Sara K. Blick-Nitko[1,2], Jason J. Zoeller[11], Ying Chen[12], Vasilis Vasiliou[12], Bradley M. Turner[4], Stephano S. Mello[1,2], Brian J. Altman[1,2], Alexander Muir[9], Jonathan L. Coloff[8], Joshua Munger[2,10], Gina M. DeNicola[3] & Isaac S. Harris[1,2,13 ✉]

Restricting amino acids from tumours is an emerging therapeutic strategy with substantial promise[1]. Although typically considered an intracellular antioxidant with tumour-promoting capabilities[2], glutathione (GSH), as a tripeptide of cysteine, glutamate and glycine, can be catabolized to release amino acids. The extent to which GSH-derived amino acids are essential to cancers is unclear. Here we show that depletion of intracellular GSH does not alter tumour growth and extracellular GSH is highly abundant in the tumour microenvironment, highlighting the potential importance of GSH outside tumours. Supplementation with GSH rescues cancer cell survival and growth in cystine-deficient conditions, and this rescue depends on the catabolic activity of γ-glutamyltransferases. Finally, pharmacological targeting of the activity of γ-glutamyltransferases prevents the breakdown of circulating GSH, reduces tumour cysteine levels and slows tumour growth. Our findings indicate a non-canonical role for GSH in supporting tumours by acting as a reservoir of amino acids. Depriving tumours of extracellular GSH or inhibiting its breakdown is potentially a therapeutically tractable approach for patients with cancer. Furthermore, these findings change our view of GSH and how amino acids, including cysteine, are supplied to cells.

Amino acids are crucial for cancer initiation, progression and drug resistance[3]. Despite their indispensability, amino acids are often scarce in the tumour microenvironment[4], which drives tumours to develop strategies to import and synthesize them[5–7]. Thus, interfering with these mechanisms and restricting amino acid access holds potential as an anticancer strategy[1]. One amino acid that has garnered substantial attention as an anticancer target is cysteine[8,9]. Beyond being a building block for proteins, cysteine has roles that include supporting the generation of antioxidants (for example, glutathione and persulfide species)[10], metabolites (for example, $H_2S$, CoA and hypotaurine)[11,12] and iron–sulfur clusters for mitochondria[13]. The intracellular cysteine pool is thought to be maintained by at least three sources: import through alanine, serine, cysteine, threonine transporter 1 (ASCT1); uptake of cystine through system $x_c^-$ (xCT and CD98) and subsequent reduction by thioredoxin reductase (TXNRD1); and de novo synthesis from methionine through the transsulfuration pathway. Cystine uptake

through system $x_c^-$ is thought to be the predominant source of cysteine in cancer cells[14]. Notably, the deletion of *Slc7a11* (which encodes xCT) in animals results in viable offspring[15], which suggests that tissues (and potentially tumours) can obtain cysteine from another origin. The generation of cysteine from the transsulfuration pathway is a potential source. However, this pathway is reported to be inactive in most tissues, and in the few tissues in which it is active, its function is reduced in tumours from the corresponding tissue[16]. Together, these results suggest that tumours have an alternative mechanism to acquire cysteine.

GSH is an antioxidant that regulates oxidative stress, drug detoxification and post-translational modifications[2]. Disruption of GSH production can impair tumorigenesis[17–19], but the exact mechanisms involved are unclear. GSH is a tripeptide of glutamate, cysteine and glycine, and it can be broken down into its individual constituent amino acids. The rate-limiting step in extracellular GSH catabolism is controlled by γ-glutamyltransferases (GGTs), which cleave the γ-glutamyl bond

[1]Department of Biomedical Genetics, University of Rochester Medical Center, Rochester, NY, USA. [2]Wilmot Cancer Institute, University of Rochester Medical Center, Rochester, NY, USA. [3]Department of Metabolism and Physiology, Moffitt Cancer Center and Research Institute, Tampa, FL, USA. [4]Department of Pathology, University of Rochester Medical Center, Rochester, NY, USA. [5]Department of Surgery, University of Rochester Medical Center, Rochester, NY, USA. [6]University of Rochester Mass Spectrometry Resource Laboratory, Rochester, NY, USA. [7]Department of Biology, University of Rochester, Rochester, NY, USA. [8]Department of Physiology and Biophysics, University of Illinois College of Medicine, University of Illinois Cancer Center, Chicago, IL, USA. [9]Ben May Department of Cancer Research, University of Chicago, Chicago, IL, USA. [10]Department of Biochemistry and Biophysics, University of Rochester Medical Center, Rochester, NY, USA. [11]Department of Cell Biology, Harvard Medical School, Boston, MA, USA. [12]Department of Environmental Health Sciences, Yale School of Public Health, New Haven, CT, USA. [13]Department of Pharmacology and Physiology, University of Rochester Medical Center, Rochester, NY, USA. [14]These authors contributed equally: Fabio Hecht, Marco Zocchi. ✉e-mail: fabio_hechtcastromedeiro@ urmc.rochester.edu; isaac_harris@urmc.rochester.edu

in GSH to release glutamate and dipeptide cysteinylglycine. After its release, cysteinylglycine is further broken down by peptidases to produce cysteine and glycine. GSH catabolism is proposed as a source of amino acids for cells[20,21]. This idea dates back to seminal work from Eagle, who showed that supplementation with extracellular GSH supports cell growth in cystine-free conditions[22]. Indeed, catabolism of mitochondrial GSH supplies cysteine for iron–sulfur cluster synthesis[23]. In animals, unlike the deletion of *Slc7a11*, the deletion of the GGT family member *Ggt1* results in increased GSH levels in urine, reduced cysteine levels in tissues and perinatal lethality[24]. Notably, patients with mutations in *GGT1* often present with developmental disorders, accumulation of GSH (glutathionaemia and glutathionuria) and reduced circulating levels of cystine[25]. Together, these findings indicate that GSH catabolism by GGTs is an important source of amino acids, including cysteine. Whether tumours can co-opt GSH catabolism to fuel their growth is poorly understood.

Here we identify an underappreciated role for GSH as a cysteine reservoir for tumours. Compared with cystine, GSH is highly abundant in the microenvironment of tumours, and supplementation with GSH or its product cysteinylglycine rescues cancer cell growth in cystine-depleted conditions. Mechanistically, GGT activity is necessary and sufficient to promote survival by catabolizing GSH and supplying amino acids to surrounding cells. Furthermore, supplementation with extracellular GSH renders cancer cells resistant to drugs that block cystine uptake or its reduction into cysteine. Finally, inhibition of GGT activity deprives tumours of cysteine and slows their growth. These results reveal an actionable pathway of nutrient acquisition in cancer with direct therapeutic implications. Furthermore, these findings change our perspectives about GSH biology and the acquisition of amino acids by cells.

## GSH production in tumours is dispensable

GSH facilitates tumorigenesis[17–19]; however, it is unclear whether intracellular GSH production by tumours themselves is required. To explore this idea, we bred the MMTV-PyMT transgenic mouse strain (which spontaneously develops breast tumours)[26] with an uninduced *Gclc^{f/f} Rosa26^{creERT2}* mouse[27] in which the rate-limiting step in GSH synthesis, the glutamate–cysteine ligase catalytic subunit (GCLC), can be knocked out throughout the body after induction with tamoxifen. After allowing tumours to develop in these mice, the tumours were excised and orthotopically implanted into recipient immunocompetent wild-type (WT) mice (C57BL/6). The recipient mice were treated with tamoxifen, which leads to the activation of Cre recombinase activity and subsequent deletion of *Gclc* specifically in the implanted tumours (Fig. 1a). *Gclc* mRNA and GSH levels were reduced in mice with *Gclc* specifically knocked out in the tumour (Fig. 1b,c). However, GSH levels were not completely abolished, which may be due to the presence of GCLC-expressing cells in the tumour microenvironment (Extended Data Fig. 1a–c). Notably, the deletion of *Gclc* in tumours did not affect their growth (Fig. 1d,e), which suggests that the ability of the tumour to synthesize intracellular GSH is dispensable. Moreover, tumours did not show signs of oxidative stress or compensatory metabolic changes, such as accumulation of cysteine (Extended Data Fig. 1d–j). Similar results were observed when tumours were implanted into the flanks of mice, a result that suggests that this phenotype is not specific to the mammary gland environment (Extended Data Fig. 2). Similar phenotypes were also observed when *GCLC* was deleted in HCC-1806 cells (a human breast cancer cell line), which showed complete depletion of GSH levels (Fig. 1f–i). The lack of an antitumour effect from the loss of intracellular GSH suggested that extracellular GSH is instead crucial for tumour growth. To explore this idea, we examined total GSH (tGSH; calculated as the sum of reduced GSH and two times the amount of oxidized glutathione (GSH + 2 × GSSG)) levels in the serum and in the tumour interstitial fluid (TIF), the extracellular compartment surrounding tumours. Examination of mouse and human breast tumours revealed that tGSH levels

were enriched in the TIF compared to serum (Fig. 1j,k). Similar trends were observed in a mouse model of pancreatic ductal adenocarcinoma (PDAC) (*LSL-Kras^{G12D/+}Trp53^{f/f}Pdx1^{cre}* (KP^{−/−}C) mice[28]) and in humans with renal cell carcinoma[29] (Extended Data Fig. 3a,b). Levels of tGSH in the TIF were much higher than reported amounts added to formulations of cell culture media (Extended Data Fig. 3c). Analyses of TIF from tumours with WT *Gclc* and with *Gclc* knocked out (KO) showed that intracellular GSH synthesis partially contributed to tGSH levels in the TIF but not serum (Extended Data Fig. 3d–h). There was no difference between tGSH and cystine levels in the serum of mice without tumours (Extended Data Fig. 3i). Notably, serum tGSH (but not serum cystine) levels were lower in mice with tumours than in mice without tumours (Extended Data Fig. 3j,k). Overall, these findings indicate that although intracellular GSH does not affect tumour growth and survival, extracellular GSH is highly abundant and potentially supports tumour growth. Furthermore, in the presence or absence of GSH synthesis, tumours have an adequate supply of upstream metabolites (that is, cysteine) to sustain growth.

## Extracellular GSH supplies cysteine

The uptake of extracellular cystine through the cystine–glutamate antiporter xCT is thought to be a major source of cysteine for tumours (Fig. 2a). Extracellular glutamate levels, which can counteract xCT-mediated cystine uptake, were enriched in the TIF compared to serum (Extended Data Fig. 3l–n), whereas cystine levels were not (Fig. 2b,c). GSH is a cysteine-containing tripeptide that can be broken down by GGT enzymes into cysteinylglycine and glutamate. Cysteinylglycine can be further processed to produce intracellular cysteine. We sought to determine whether GSH or its catabolic product cysteinylglycine can replace cystine as a source of cysteine for cancer cells. Supplementation with GSH, at a concentration within the range found in the TIF, or with cysteinylglycine rescued cancer cell growth in cystine-free conditions (Fig. 2d and Extended Data Fig. 4a). Moreover, GSH or cysteinylglycine rescued both tumour survival and proliferation under cystine-free conditions (Fig. 2e,f).

This rescue by GSH was not observed after deprivation of other amino acids (serine, glycine, glutamine or glutamate) or after inhibition of amino acid synthesis (via the glutaminase 1 inhibitor CB-839) (Extended Data Fig. 4b–e). Moreover, this rescue did not depend on intracellular re-synthesis of GSH, as pharmacological inhibition of GSH synthesis did not prevent GSH rescue in cystine-free conditions (Extended Data Fig. 4f,g). Furthermore, the L-isomer of *N*-acetylcysteine (NAC), which is both a cysteine source and an antioxidant, but not D-NAC, which is only an antioxidant[30], rescued cancer cell growth in cystine-free conditions (Extended Data Fig. 4h). Cysteine deprivation is linked to ferroptosis, a non-apoptotic form of cell death that involves lipid peroxidation[31]. Notably, supplementation with ferrostatin-1 or trolox, both of which are radical-trapping antioxidants and potent inhibitors of ferroptosis, rescued survival but not proliferation under cystine-free conditions (Extended Data Fig. 5a–c). Unlike GSH, supplementation with these antioxidants did not supply cysteine, which suggests that preventing lipid peroxidation after cystine deprivation is not sufficient to sustain cancer cells. This idea was further supported when similar phenotypes were observed across ferroptosis-sensitive and ferroptosis-resistant cancer cell lines (Extended Data Fig. 5d–j). Next, metabolite levels were compared across cells cultured with the following medium: control (cystine supplemented); cystine-free; and cystine-free with GSH supplemented. Supplementation with GSH resulted in time-dependent accumulation of extracellular cysteinylglycine (Fig. 2g and Extended Data Fig. 6a), which suggests that there is catabolism of extracellular GSH by cancer cells. Levels of intracellular cysteinylglycine and cysteine were not rescued by GSH supplementation in cystine-free conditions (Extended Data Fig. 6b,c). However, this effect may be due to the large amount of cystine provided in the control conditions (208 μM), as reduction of cystine levels in the medium did not impair growth but abolished intracellular

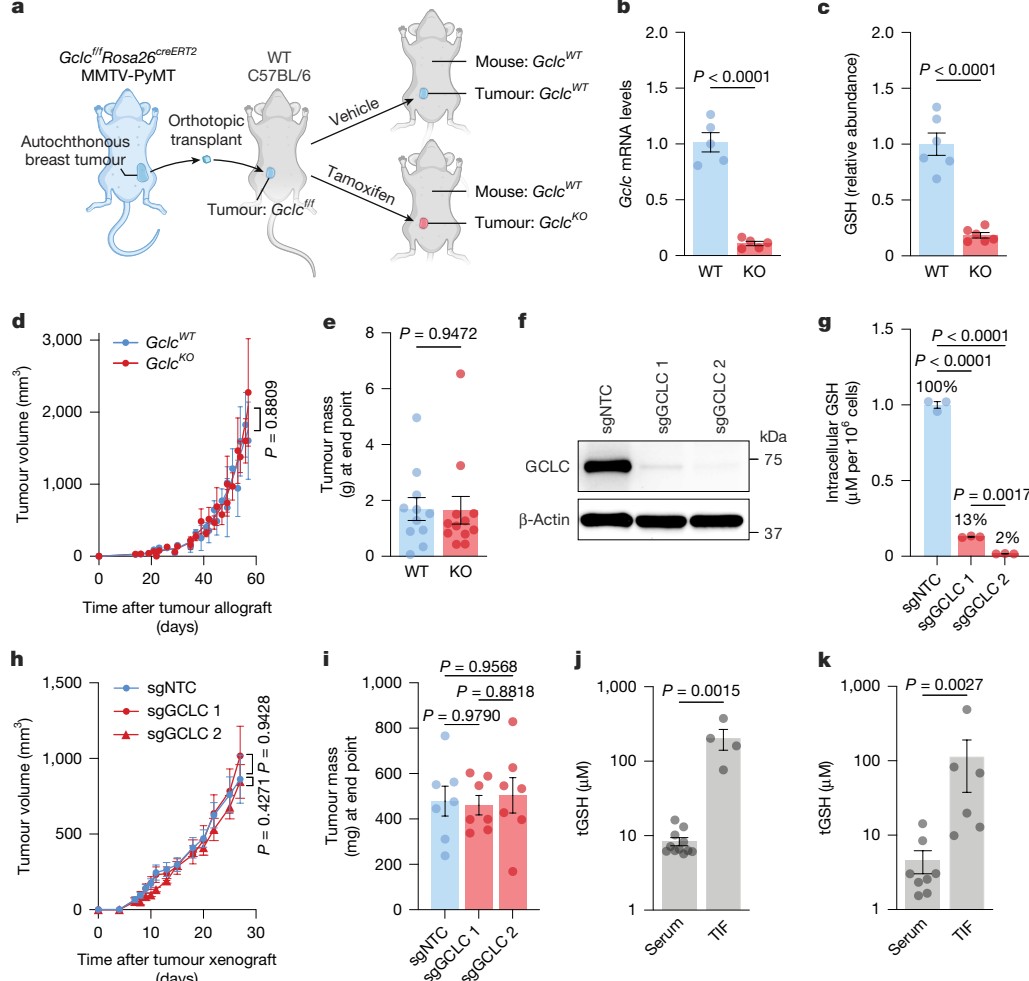

**Fig. 1 | Intracellular production of GSH is dispensable for tumour growth.**
**a**, Schematic of the tumour-specific *Gclc* knockout mouse model. Autochthonous tumours from *Gclc^f/f^Rosa26^creERT2^* MMTV-PyMT mice were excised and orthotopically transplanted into mammary fat pads of WT C57BL/6 mice. C57BL/6 mice were treated with vehicle (corn oil; WT) or 50 mg kg$^{-1}$ tamoxifen for 5 days (KO). **b**, Relative *Gclc* mRNA levels of WT and KO tumours ($n = 5$ representative animals from 3 independent experiments). **c**, GSH levels of WT and KO tumours ($n = 6$ representative animals from 3 independent experiments). **d**,**e**, Volume over time (**d**) and mass at end point (**e**) of WT tumours ($n = 11$) and KO tumours ($n = 12$) from a representative of 3 independent experiments. **f**,**g**, Immunoblot of GCLC protein levels (**f**) and GSH levels (**g**) in HCC-1806 human breast cancer cells transduced with lentiCRISPRv2 containing

non-targeting guides (sgNTC) and guides against *GCLC* (sgGCLC 1 and sgGCLC 2) ($n = 3$ representative samples from 2 independent experiments). For gel source data, see Supplementary Fig. 1. **h**,**i**, Volume over time (**h**) and mass at end point (**i**) of orthotopically implanted HCC-1806 breast cancer cell treated with sgNTC, sgGCLC 1 or sgGCLC 2 ($n = 7$ representative animals from 2 independent experiments). **j**, Concentration of tGSH (GSH + 2 × GSSG) in serum ($n = 11$) and the TIF ($n = 4$) from *Gclc* WT MMTV-PyMT autochthonous tumours from **a**. **k**, Concentration of tGSH in serum ($n = 8$) and the TIF ($n = 6$) from patients with breast cancer. Significance was assessed by two-tailed unpaired *t*-test (**b**,**c**,**e**), Mann–Whitney test (**j**,**k**), ordinary two-way analysis of variance (ANOVA) (**d**,**h**) or one-way ANOVA followed by Tukey's multiple comparisons test (**g**,**i**). Data are the mean ± s.e.m.

cysteine and cystine levels (Extended Data Fig. 6d–f). Notably, GSH supplementation rescued the levels of downstream cysteine-related products (Extended Data Fig. 6g–i) and prevented the accumulation of ophthalmic acid (Extended Data Fig. 6j), a metabolite that is produced by GCLC in the absence of cysteine[32,33]. To examine the incorporation of GSH-derived cysteine into downstream pathways, we used an inverse stable-isotope labelling approach[34,35]. Cancer cells were grown in medium with $^{13}$C-cystine and then switched to cystine-free medium with unlabelled GSH (Fig. 2h). Cysteine and cysteine-dependent metabolites (glutamylcysteine, GSH and hypotaurine) were labelled when cells were grown in medium with $^{13}$C-cystine and became unlabelled when cancer cells were grown in cystine-free medium with unlabelled GSH (Fig. 2i and Extended Data Fig. 6k,l). Similar findings were observed using $^{15}$N$^{13}$C-cystine and by performing high-resolution proteomics (Fig. 2j,k and Extended Data Fig. 6k,m). Moreover, the level of residual labelling of individual peptides positively correlated with their protein half-lives (measured independently in primary human fibroblasts[36]) (Extended

Data Fig. 6n,o). This result suggests that the residual cystine labelling reflects the slow turnover of these peptides. Finally, as intracellular GSH synthesis partially contributed to tGSH levels in the TIF, we explored the impact of blocking GSH export through drug transporters[37]. Pharmacological inhibition of drug transporters in breast cancer cells did not affect the ability of GSH to rescue tumour growth in cystine-free conditions (Extended Data Fig. 7a–d), even though it promoted the accumulation of GSH in cells (Extended Data Fig. 7e–h). However, further research is required to fully understand the interplay between GSH export from cancer cells and GSH levels in the TIF in vivo. Together, these findings indicate that extracellular GSH is broken down to fuel the intracellular metabolic and proteogenic processes in cancer cells.

## GGT1 drives GSH catabolism and survival

Several enzymes have GGT activity[38], and GGT1 is the most catalytically active isoform[39]. We used three separate approaches to examine

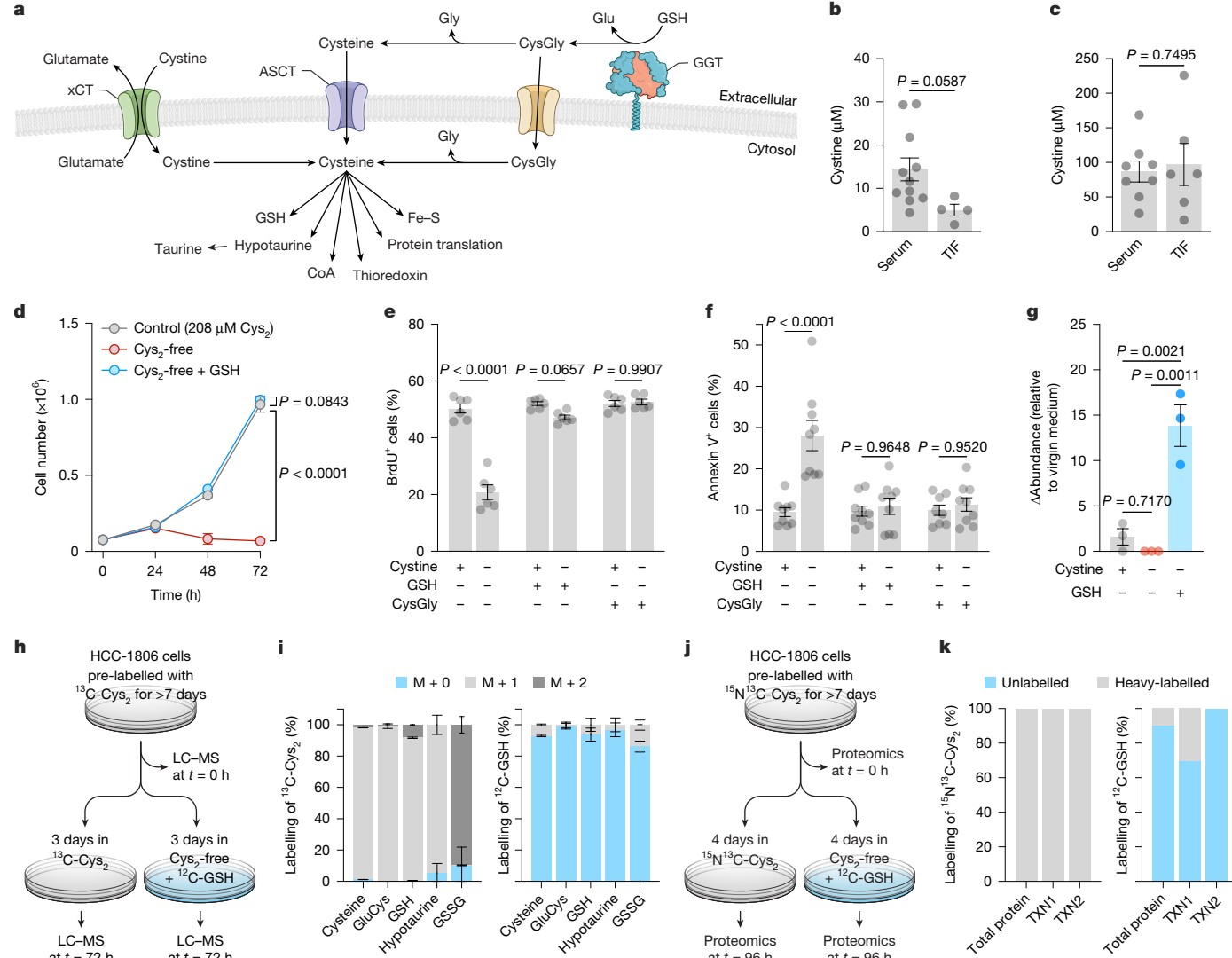

**Fig. 2 | Extracellular GSH supplies amino acids to promote cancer cell growth and survival in cystine-free environments. a**, Schematic of the different mechanisms of cysteine acquisition and use. **b**, Concentration of cystine in serum ($n = 11$) and the TIF ($n = 4$) from *Gclc* WT MMTV-PyMT autochthonous tumours from Fig. 1a. **c**, Concentration of cystine in serum ($n = 8$) and the TIF ($n = 6$) from patients with breast cancer. **d–f**, HCC-1806 breast cancer cells were grown in the following medium: control (208 μM cystine; Cys$_2$); cystine-free (Cys$_2$-free); cystine-free with GSH supplemented (750 μM, Cys$_2$-free + GSH); or cystine-free with cysteinylglycine (CysGly) supplemented (750 μM). Cell numbers ($n = 3$ technical replicates representative of 4 independent experiments) (**d**), percentages of proliferative cells (positive for bromodeoxyuridine (BrdU$^+$) at $t = 48$ h; $n = 6$ technical replicates from 3 independent experiments) (**e**) and apoptotic cells (Annexin V$^+$ at $t = 72$ h; $n = 9$ technical replicates from 3 independent experiments) (**f**) were determined at the indicated time points. **g**, Levels of extracellular CysGly in the medium at 72 h after HCC-1806 breast cancer cells were grown in the indicated medium ($n = 3$ representative technical replicates from 2 independent experiments). **h**, Schematic of the $^{13}$C-cystine stable-isotope labelling approach and metabolomics. **i**, Per cent labelling of $^{13}$C-cystine stable-isotope in the indicated species at 72 h ($n = 5$ technical replicates). **j**, Schematic of the $^{15}$N$^{13}$C-cystine stable-isotope labelling approach and proteomics. **k**, Per cent labelling of $^{15}$N$^{13}$C-cystine stable-isotope in the indicated species at 96 h (average of $n = 3$ technical replicates). TXN1, thioredoxin 1; TXN2, thioredoxin 2. Significance was assessed by unpaired two-tailed $t$-test (**b,c**), ordinary two-way ANOVA followed by Šídák's multiple comparisons test (**e,f**), Dunnett's multiple comparisons test (**d**) or one-way ANOVA followed by Tukey's multiple comparisons test (**g**). Data are the mean ± s.e.m.

the necessity of GGT1 to sustain GSH catabolism and cysteine supply to cancer cells. We generated *GGT1* knockdown cell lines using CRISPRi approaches and *GGT1* KO cell lines (polyclonal and single-cell clones) using CRISPR–Cas9 approaches. *GGT1* knockdown cell lines had minimal *GGT1* mRNA (Extended Data Fig. 8a) but still retained substantial GGT activity (Extended Data Fig. 8b) and grew in a similar manner to control cell lines when cultured under cystine-replete and cystine-depleted with GSH-supplemented conditions (Extended Data Fig. 8c). *GGT1* KO cell lines showed decreased GGT activity (especially in *GGT1* KO single-cell clones). However, in both models, cancer cell growth was rescued under cystine-free with GSH-supplemented conditions (Extended Data Fig. 8d–g). Cancer cells express multiple

GGT isoforms (Extended Data Fig. 9a–d), but none of these isoforms displayed a dependency in cancer cell line genetic screens (Extended Data Fig. 9e). Expression of GGT isoforms varied across tumour subtypes, but in most cases, there was increased expression in tumours compared with normal tissue (Extended Data Fig. 9f,g). However, GGT isoforms were rarely found mutated in cancers (Extended Data Fig. 9h). Moreover, expression of *Ggt1* in mouse tumours did not depend on *Gclc* expression or the site of tumour growth (Extended Data Fig. 9i). Further studies are required to better understand how GGT1, other GGT family members and uncharacterized proteins with GGT activity contribute to extracellular GSH catabolism. Separately, we speculated that GGT1 is sufficient to support the breakdown of GSH in the extracellular

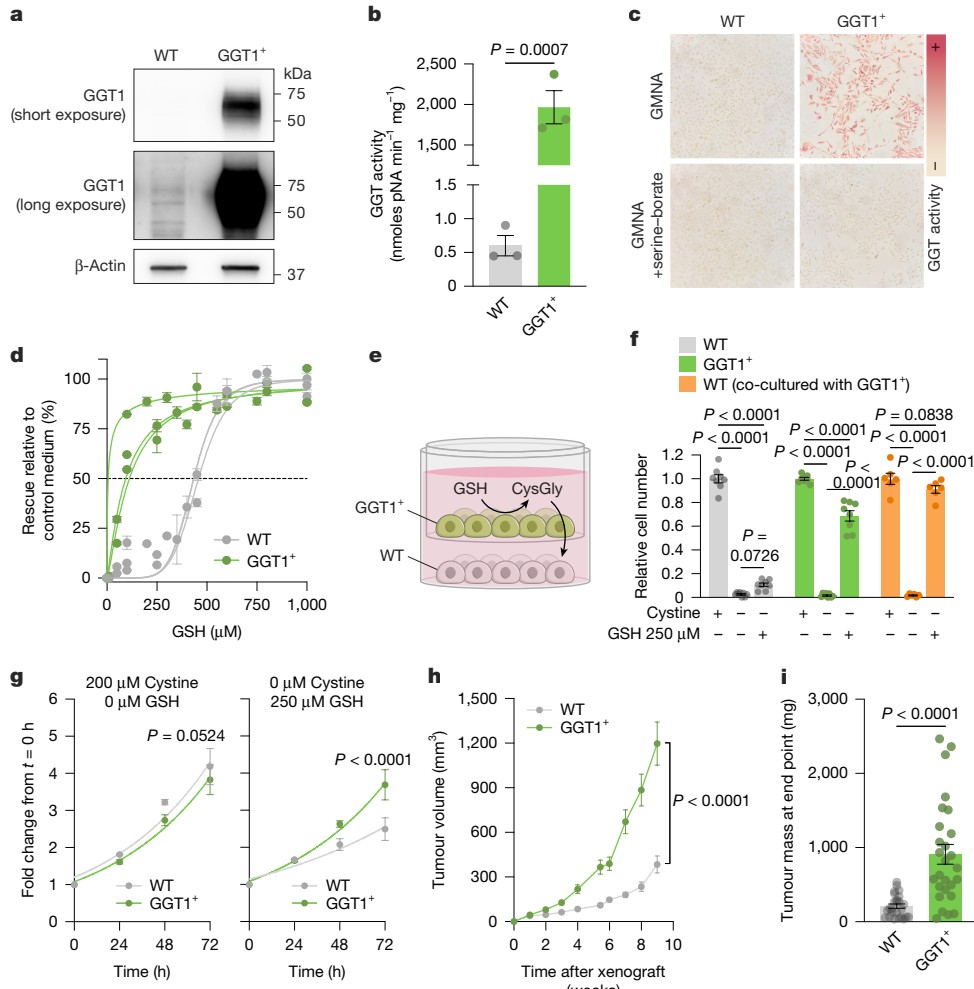

**Fig. 3 | GGT1 is sufficient to promote GSH catabolism and tumour growth.**
**a**,**b**, Immunoblot analyses of human GGT1 (**a**) and GGT activity (**b**) in WT and
GGT1+ PC3 prostate cancer cells ($n$ = 3 technical replicates from 2 independent
experiments). For gel source data, see Supplementary Fig. 1. **c**, GGT activity
measured by γ-4-methoxy-β-naphthylamide (GMNA) histochemical staining
in WT and GGT1+ cells. Serine–borate was used to competitively inhibit GGT
(negative control). Magnification of micrographs, ×10. **d**, WT and GGT1+ cells
were grown for 72 h in control medium (208 μM cystine) or in cystine-free
medium supplemented with the indicated GSH concentrations ($n$ = 6 technical
replicates from 3 independent experiments). **e**, Schematic of non-contact
co-culture experiments using 0.4 μm PET membrane Transwell inserts
in medium containing low concentrations of GSH (250 μM), which were
insufficient to rescue the growth of WT cells in cystine-depleted conditions.

**f**, Relative cell numbers of WT, GGT1+ and WT cells co-cultured with GGT1+ cells
in control, cystine-depleted or cystine-depleted with GSH-supplemented
(250 μM) conditions ($n$ = 8 technical replicates for WT and GGT1+, $n$ = 6 for
co-cultured cells, from 4 independent experiments). **g**, Relative cell numbers at
the indicated time points for WT and GGT1+ cells grown in medium containing
200 μM cystine with 0 μM GSH (left) or 0 μM cystine with 250 μM GSH (right)
($n$ = 3 technical replicates representative of 3 independent experiments).
**h**,**i**, Tumour volume over time (**h**) and mass at end point (**i**) of xenograft tumours
from WT ($n$ = 26) and GGT1+ ($n$ = 28) PC3 cells ($n$ = 2 independent experiments).
Significance was assessed by unpaired two-tailed $t$-test (**b**,**i**) or two-way
ANOVA followed by Tukey's multiple comparisons test (**f**) or Šídák's multiple
comparisons test (**g**,**h**). Data are the mean ± s.e.m.

environment. Overexpression of GGT1 (GGT1+) in cells[40] resulted in
increased GGT1 protein levels (Fig. 3a) and GGT activity (Fig. 3b,c).
Lower levels of GSH were required to rescue GGT1+ cell growth in
cystine-free conditions than for control cells (Fig. 3d), which suggests
that catabolism by GGT is a rate-limiting step in GSH-dependent res-
cue of cancer cell growth. We proposed that GSH catabolism by cells
with high GGT activity can support surrounding cells in a paracrine
fashion. To test this hypothesis, we co-cultured GGT1+ cells with WT
cells using a Transwell assay (Fig. 3e). Even though GSH levels were
below the threshold for rescuing WT cell growth in cystine-free con-
ditions, co-culturing with GGT1+ cells led to complete rescue of WT
cell growth (Fig. 3f). Moreover, GGT1+ cells grew faster in cystine-free
with GSH-supplemented conditions but not in cystine-replete con-
ditions (Fig. 3g). Finally, this growth advantage translated to faster
growth of GGT1+ cells in vivo (Fig. 3h,i). These findings demonstrate

that GGT activity is sufficient to support GSH catabolism and survival
of surrounding cells in cystine-depleted conditions. Furthermore, this
result suggests that non-tumorigenic tissues or cells in the tumour
microenvironment with high GGT activity can drive tumour growth
and progression by catabolizing GSH and supplying amino acids in a
paracrine (or endocrine) manner.

## GSH catabolism alters drug sensitivity

The metabolic environment can influence the sensitivity of cancer cells
to anticancer compounds[41,42]. To investigate the impact of shifting from
cystine-driven to GSH-driven cysteine supply in cancer cells in an unbi-
ased manner, we performed a multifunctional approach to pharma-
cological screening (MAPS) with a library of 240 metabolic inhibitors
(Fig. 4a,b). When cancer cells relied on GSH for cysteine acquisition,

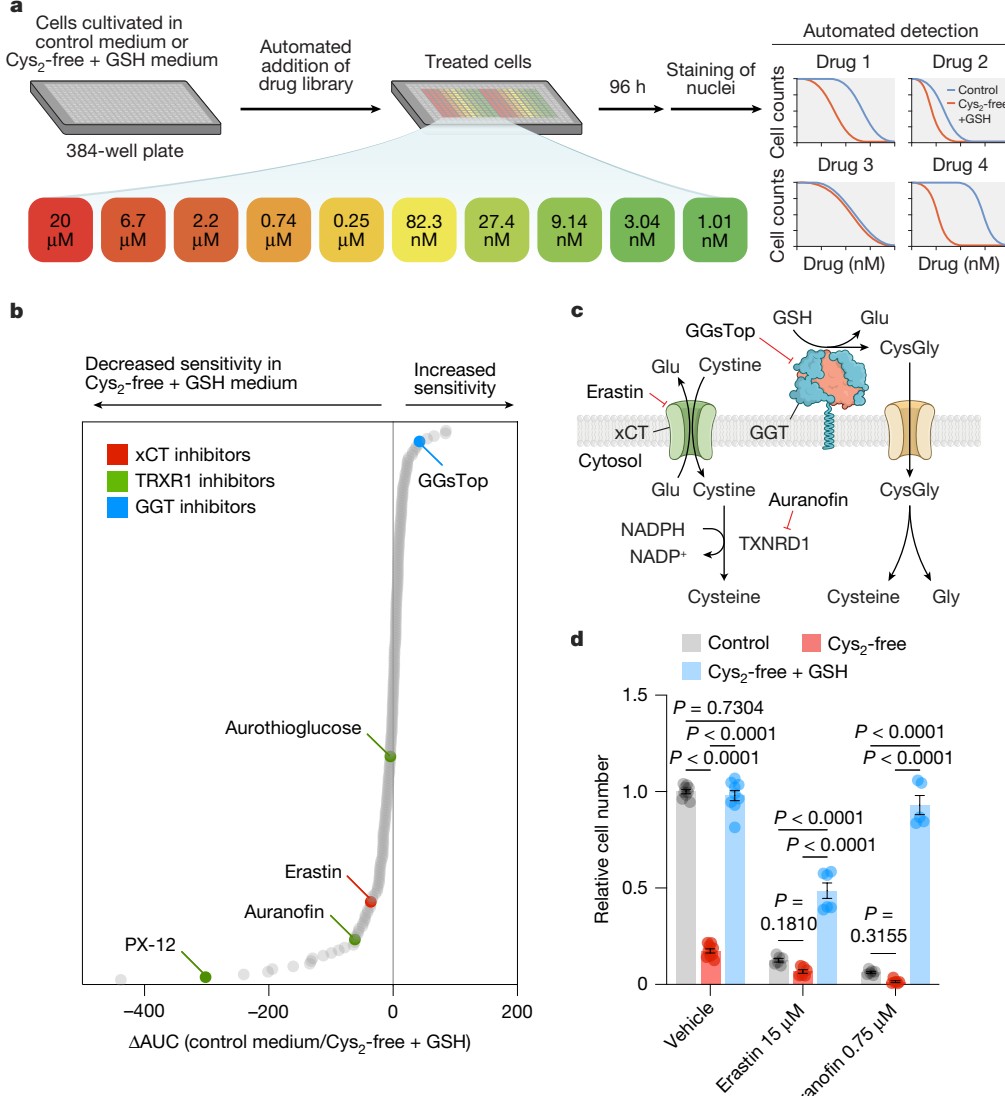

**Fig. 4 | Use of GSH as a cysteine source reduces the sensitivity of cancer cells to inhibitors of metabolic enzymes. a**, Schematic of MAPS. HCC-1806 cells grown in control (208 μM cystine) or cysteine-free with GSH supplemented (500 μM) medium were treated with libraries of drugs, each arrayed at 10 dose points (20 μM–1 nM). After 96 h, cell numbers were determined, and dose–response curves were generated for each drug. **b**, MAPS results showing each drug ranked by the difference in the area under the curve (ΔAUC) obtained

for each drug curve in the indicated conditions ($n = 2$ technical replicates). **c**, Schematic showing the targets of selected hits from **b**. **d**, HCC-1806 cells were grown in 6-well plates and treated with erastin and auranofin for 72 h in different culture media, and relative cell numbers were determined ($n = 6$ technical replicates from 3 independent experiments). Significance was evaluated by two-way ANOVA followed by Tukey's multiple comparisons test. Data are the mean ± s.e.m.

they were more sensitive to GGsTop[43], a putative inhibitor of GGT activity (Fig. 4b,c). Conversely, cancer cells were less sensitive to inhibitors of cystine uptake (for example, erastin) and the thioredoxin pathway (for example, auranofin, aurothioglucose and PX-12) (Fig. 4b–d); this pathway mediates the reduction of cystine to cysteine after its import[44]. The decreased sensitivity of cancer cells to these inhibitors may be due to several reasons, including extracellular GSH binding to and inactivating the inhibitors. Alternatively, when cancer cells use GSH as a cysteine source, they rely less on these pathways (that is, xCT and TXNRD1). Moreover, the presence of GSH in the tumour microenvironment argues against the feasibility of targeting these pathways for cancer therapy. Notably, commonly used cell culture media (for example, DMEM, RPMI and F12) contain supraphysiological levels of cystine but lack GSH and cysteinylglycine. Consequently, the contribution of xCT to cancer cell growth may be overestimated, whereas the importance of GGTs may be undervalued.

## GSH catabolism supports tumour growth

High-throughput screening of cancer cells identified a GGT inhibitor (GGsTop) with increased sensitivity when GSH was provided as a cysteine source. To further explore this effect, we examined additional GGT inhibitors (acivicin and OU749)[45] but found that GGsTop was the most potent at blocking GGT activity (Fig. 5a,b and Extended Data Fig. 10a). Cancer cells were sensitive to GGsTop when grown in cystine-free with GSH-supplemented conditions, but this increased sensitivity was reversed when cells were provided with cysteinylglycine, the product of GGT-mediated GSH catabolism (Fig. 5c,d). The kidney is suggested to have the highest levels of GGT1 (ref. 46), which was confirmed by an analysis of GGT activity in mouse tissues and expression of GGTs from public datasets (Extended Data Fig. 10b,c). Daily intraperitoneal injections of GGsTop, which has previously been shown not to impair tumour growth[47], did not abolish kidney GGT activity

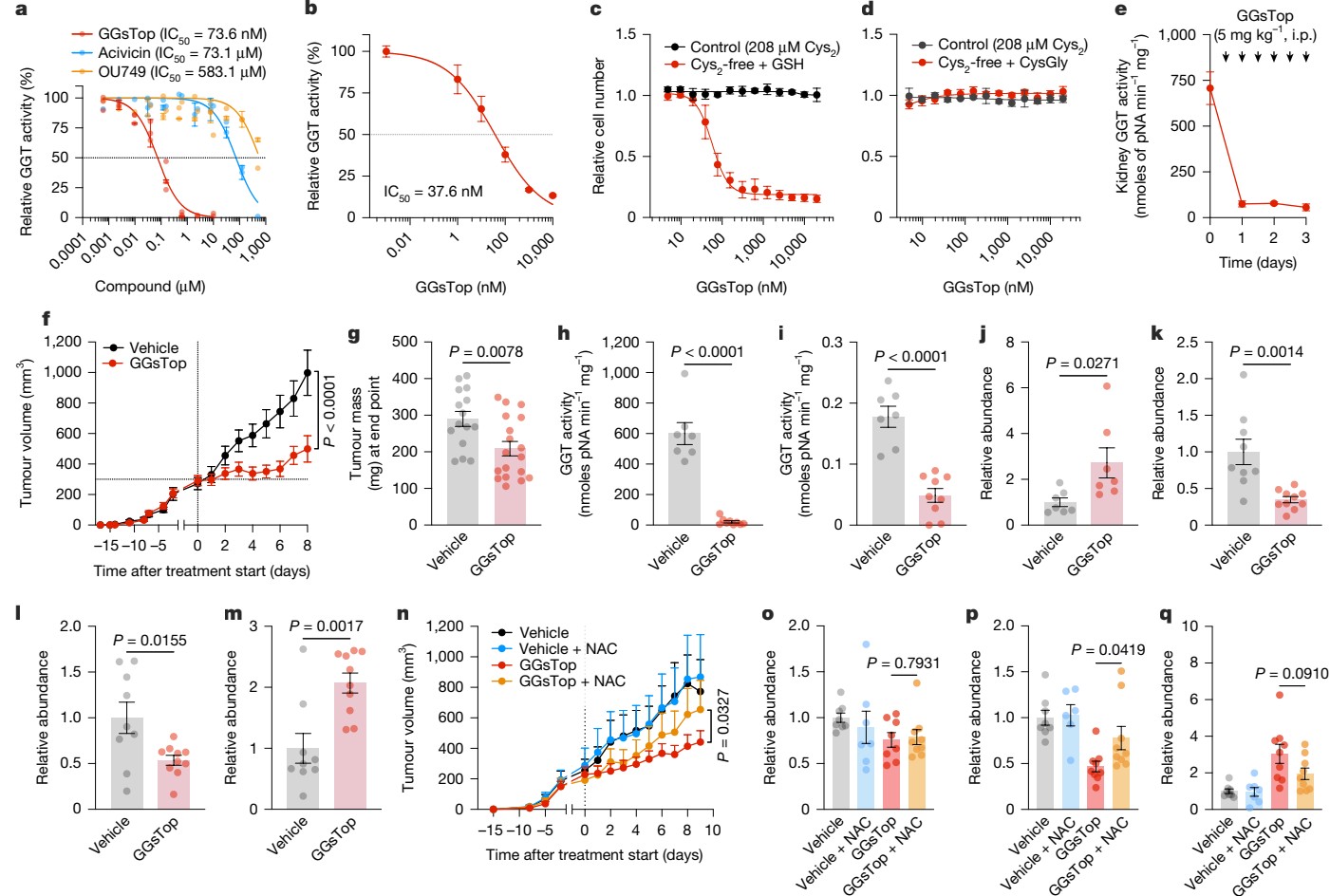

**Fig. 5 | GSH catabolism is necessary to support cysteine supply and tumour growth. a**, GGT activity in mouse kidney extracts treated with the indicated GGT inhibitors ($n = 3$ technical replicates from 3 independent experiments). **b**, GGT activity in HCC-1806 cells treated with GGsTop for 4 h ($n = 2$ technical replicates representative of 3 independent experiments). **c,d**, Relative cell numbers of HCC-1806 cells treated with GGsTop for 96 h in control or cystine-free medium supplemented with GSH (**c**) or CysGly (**d**) ($n = 4$ technical replicates from 2 independent experiments). **e**, GGT activity in kidney extracts of mice intraperitoneally (i.p.) injected with vehicle ($n = 6$ mice) or with 5 mg kg$^{-1}$ of GGsTop every 12 h for 1 ($n = 9$), 2 ($n = 9$) or 3 days ($n = 8$). **f**, Tumour volume of orthotopic HCC-1806 xenografts in mice treated with vehicle ($n = 15$) or 5 mg kg$^{-1}$ GGsTop ($n = 18$) every 12 h. Data are representative of two independent experiments. **g**, Tumour mass at the end point from **f**. **h,i**, GGT activity in the kidney (**h**) and tumours (**i**) from mice in at the end point (vehicle, $n = 7$;

GGsTop, $n = 9$). **j**, Serum GSH (*N*-ethylmaleimide (NEM)-GSH) levels from mice in **f** ($n = 7$). **k–m**, Relative abundance of cysteine (**k**), hypotaurine (**l**) and ophthalmic acid (**m**) from tumours in **f** (vehicle, $n = 9$; GGsTop, $n = 10$). **n**, Tumour volume of orthotopic HCC-1806 xenografts treated with vehicle or 5 mg kg$^{-1}$ GGsTop every 12 h alone or supplemented with NAC (30 mM) in drinking water (vehicle, $n = 5$; GGsTop, $n = 8$; vehicle with NAC, $n = 5$, GGsTop with NAC, $n = 7$, representative of 2 independent experiments). **o–q**, Relative abundance of cysteine (**o**), hypotaurine (**p**) and ophthalmic acid (**q**) from tumours in **n** (vehicle, $n = 8$; GGsTop, $n = 7$; vehicle with NAC, $n = 8$, GGsTop with NAC, $n = 9$, from 2 independent experiments). Significance was assessed by ordinary two-way ANOVA followed by Šídák's multiple comparisons test (**f**), unpaired two-tailed *t*-test (**g–m,o–q**) or ordinary two-way ANOVA (**n**). Data are the mean ± s.e.m.

(Extended Data Fig. 10d). However, intraperitoneal injections twice a day resulted in substantial inhibition of kidney GGT activity (Fig. 5e). Notably, GGsTop treatment blocked GGT activity and slowed tumour growth without causing any overt toxicity to animals (Fig. 5f–i and Extended Data Fig. 10e,f). Mechanistically, GGsTop treatment led to an accumulation of serum tGSH and a depletion of cysteine in tumours (Fig. 5j,k). Furthermore, it resulted in tumours with reduced levels of cysteine-dependent metabolites (hypotaurine) and increased levels of ophthalmic acid, which is synthesized in the absence of cysteine[32,33] (Fig. 5l,m). GGsTop treatment did not affect levels of GSH, GSSG or glutamate in tumours (Extended Data Fig. 10g–i). The impaired tumour growth induced by GGsTop treatment was rescued by supplementation with a cell-permeable source of cysteine (NAC) (Fig. 5n). Treatment of cells with both NAC and GGsTop did not rescue tumour cysteine levels but did rescue hypotaurine levels and partially suppressed ophthalmic acid levels (Fig. 5o–q and Extended Data Fig. 10j–o). GGsTop activity

remained stable in water at room temperature (Extended Data Fig. 10p), and GGsTop delivery in drinking water led to inhibition of GGT activity in animals and suppressed tumour growth (Extended Data Fig. 10q–t). Together, these findings indicate that GGT activity maintains GSH catabolism to supply cysteine and support tumour growth. Furthermore, the data showed that blocking GGT is a potential therapeutic strategy for patients with cancer.

## Discussion

A prevailing dogma is that a major fate of cysteine is its incorporation into the antioxidant GSH. Here we demonstrated that the opposite is also true: extracellular GSH is a precursor to intracellular cysteine. Moreover, we showed that the downstream fate of cysteine in tumours is not necessarily GSH. Notably, the affinity of cysteine for cysteinyl-tRNA synthetase is orders of magnitude higher than its affinity for the rate-limiting

enzyme in GSH synthesis (GCLC)[48,49]. This result highlights that rather than synthesizing GSH, cysteine has other downstream fates of greater importance, such as protein translation or cysteine polysulfidation[50].

GSH levels were enriched in the TIF compared with serum in mouse models of cancer and in human patients with cancer. Although tumours themselves contributed to GSH abundance in the TIF, additional cell types probably also contribute. For example, after activation, macrophages can efflux GSH[51]. Moreover, GSH levels and GSH to GSSG ratios can vary across mouse strains[52], and therefore could also affect cysteine supply to tissues (and tumours) in these mice. Finally, we observed lower serum GSH levels in mice with tumours than in mice without. Past studies have shown that serum GSH levels are lower in patients with colorectal cancer than in healthy individuals[53]. Further research is required to determine the utility of serum GSH measurement as a predictor of cancer incidence.

Our findings demonstrated that GSH catabolism by GGTs supports cancer cell growth and survival. A question that remains to be answered is whether GSH catabolism occurs primarily in tumours or in non-tumorigenic tissue (for example, kidney). Furthermore, less is known about whether a specific subtype of cancer relies more on GSH catabolism. Renal cell carcinomas (papillary and clear cell) express some of the highest levels of GGT1. By contrast, another subtype of renal cell carcinoma (chromophobe) contains the lowest GGT1 expression levels among tumours. Indeed, GGT1 has been studied in the context of kidney cancer, with pro-tumour and antitumour effects being attributed[54–56]. Even less is known regarding its potential role in tumour metastasis and the potential side effects of GSH catabolism in specific tissues (for example, release of glutamate in brain metastasis). Numerous additional questions remain regarding the catabolism of metabolites and their impact on tissue homeostasis in disease.

Following its release from GSH, cysteinylglycine must be further broken down into the individual amino acids cysteine and glycine. Dipeptidases are predicted to control the breakdown of dipeptides, including cysteinylglycine; however, the exact enzymes involved are poorly understood. Carnosine dipeptidase II (CNDP2) has been described to support the catabolism not only of cysteinylglycine[57] but also of other dipeptides, such as those containing glutamine[35]. Future studies involving a systematic analysis of dipeptidase–dipeptides relationships, in malignant and non-malignant tissues, are required.

Targeting the supply of amino acids, including cysteine, to tumours is an emerging area of therapeutic research. Substantial attention has been paid to blocking cystine uptake[58] or cysteine production from methionine through the transsulfuration pathway[59]. Previous studies that have measured the incorporation of cysteine into metabolites suggested that additional sources of cysteine for tumours, beyond extracellular cystine or de novo synthesized cysteine, probably exist[16]. We examined the importance of a non-intuitive source of cysteine: that is, GSH catabolism by GGTs. We showed that blocking GGT activity slows tumour growth by depriving it of intracellular cysteine. The therapeutic window for inhibiting GGT as a cancer therapy is potentially large, as sustained inhibition of GGT activity did not induce any overt toxicities. Degradation of extracellular GSH, in a similar approach to degradation of extracellular cystine or cysteine[9], is an additional potential approach. Beyond a target for therapeutic intervention, monitoring GSH catabolism holds value as a cancer diagnostic. Indeed, serum GGT activity is a routine clinical measurement and is a risk factor for cancer[60]. Further research is required to fully elucidate the therapeutic potential surrounding GSH catabolism and the supply of amino acids to cancers.

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

## Methods

### Animal studies

All animal studies were performed according to protocols approved by the University Committee on Animal Resources at the University of Rochester Medical Center. $Gclc^{f/f}$ mice[27] were crossed with the MMTV-PYMT (The Jackson Laboratory, 022974)[61] and $Rosa26^{creERT2}$ (The Jackson Laboratory, 008463)[62] mouse strains. For the tumour allograft model, $Gclc^{f/f}Rosa26^{creERT2}$ MMTV-PYMT mice and control C57BL/6 mice (The Jackson Laboratory, 000664) were used (female, at least 12 weeks old). For tumour xenograft experiments, athymic nude NU/J mice (The Jackson Laboratory, 002019) were used (female, at least 8 weeks old).

For the tumour allograft model, $Gclc^{f/f}Rosa26^{creERT2}$ MMTV-PYMT mice were euthanized and autochthonous tumours were collected, cut into 8 mm³ fragments and cryopreserved in liquid nitrogen. These fragments were later thawed on ice and implanted into the right fourth mammary fat pad of immunocompetent C57BL/6 mice. Once the grafted fragments had formed fully established tumours, they were collected, cut into 8 mm³ fragments and cryopreserved using the same procedure. Finally, these tumour fragments were thawed and grafted into recipient immunocompetent C57BL/6 mice and used for the designated experimental treatments. To activate Cre recombinase and induce $Gclc$ deletion in tumour pieces from $Gclc^{f/f}Rosa26^{creERT2}$ MMTV-PYMT mice, female mice were intraperitoneally injected with tamoxifen (Sigma-Aldrich, T5648) at 50–160 mg kg⁻¹ once daily for 5 consecutive days. For tumour xenograft models, $1 \times 10^6$ HCC-1806 cells or $5 \times 10^6$ PC3 cells were injected into the right fourth mammary fat pad or hind flank of athymic nude NU/J mice (The Jackson Laboratory, 002019) in 100 µl sterile PBS or a 1:1 mixture of cell suspension in PBS and Matrigel (Corning, 356237) using a 29-gauge 1 ml insulin syringe. In all cases, animals received only one cell injection at a single location per experiment.

Tumour volume was estimated using the formula for an oblate spheroid $v = \frac{\pi}{6} \times (l^2) \times w$, where volume ($v$) is calculated using caliper measurements of the longest ($l$) and shortest ($w$) sides of the tumour. For pharmacological interventions, treatment of mice was initiated once the average tumour volume in the groups reached approximately 300 mm³. Mice were treated with pharmacological agents by either intraperitoneal (i.p.) injection or inclusion in the drinking water. For treatment via i.p. injection, GGsTop (Tocris, 4452; WuXi, 926281-37-0; or MCE, HY-108467) was diluted in sterile saline and injected at the indicated concentrations. For treatment via drinking water, GGsTop (WuXi, 926281-37-0) was diluted to 62.5 µg ml⁻¹ (189 µM) and provided to mice ad libitum. Based on an average daily water consumption of 4 ml, the estimated daily dose of GGsTop for a 25 g mouse was 10 mg kg⁻¹ per day (equivalent to 5 mg kg⁻¹ twice a day i.p. injections). NAC (Sigma-Aldrich, A7250) was diluted in drinking water at a concentration of 30 mM, the pH was adjusted to 7.00 and it was provided to mice ad libitum.

Mice of desired strains were age-matched and assigned randomly to their treatment groups. For xenograft studies, animals were allocated into groups ensuring the mean, median and standard error of tumor size was similar across all groups. Investigators were not blinded to group allocation during experiments owing to technical limitations. Statistical methods were not used to chose sample sizes. The number of animals assigned per condition was selected to account for the variability of the examined phenotypes based on pilot experiment and past experience with the animal models (≥6 animals per experimental condition). Mice were euthanized at humane end points (when the tumour diameter exceeded a total of 20 mm in length, when ulceration occurred or when weight loss exceeded 20%). Tumours and tissues were snap-frozen on dry ice and stored at −80 °C or placed in 10% neutral buffered formalin (Fisher Scientific, 22-110-761) for 24 h and then stored in 100% methanol (Fisher Scientific, A412P-4).

### Immunohistochemistry

Formalin-fixed, paraffin-embedded tissue sections (5 µm) were used for haematoxylin and eosin staining and immunohistochemical analyses. The tissues were dewaxed and rehydrated through a series of xylene and ethanol changes. For antibody staining, antigen retrieval was performed on the slides by incubating them in a steamer for 40 min in citrate buffer (Vector Labs, H-3300-250). Tissues were permeabilized with PBS supplemented with 0.1% Tween-20 for 10 min, followed by a peroxidase block with 3% $H_2O_2$ (Sigma-Aldrich, 216763) for 20 min. The slides were washed in PBS and then blocked using 10% goat serum in PBS for 1 h at room temperature before adding primary antibodies diluted in the blocking buffer. Primary antibody (anti-GCSc 1:100 (SantaCruz, sc390811), anti-CD45 1:100 (Santa Cruz, sc1178), anti-F4/80 1:100 (Cell Signaling Technology, 70076), anti-DNA/RNA Damage 1:200 (Abcam, 62623), anti-NRF2 1:100 (Abcam, 31163) and anti-GGT1 1:200 (donated from M. H. Hannigan)[63]) incubation was carried out overnight at 4 °C. Biotinylated goat anti-mouse IgG or goat anti-rabbit IgG (Vector Labs, BA-9200 1:200 and BA-1000 1:200, respectively) secondary antibodies were added in blocking buffer and incubated for 1 h at room temperature. Stains were developed using VectaStain Elite ABC-HRP Peroxidase and DAB Substrate kits (Vector Labs, PK-7100 and SK-4100, respectively). Tissues were counterstained with haematoxylin, mounted with Permount mounting medium (Fisher Scientific, SP15500) and coverslipped for imaging. Images were taken using an Olympus VS120 virtual slide microscope and Visiopharm image analysis system.

### Immunofluorescence

Formalin-fixed, paraffin-embedded tissue sections (5 µm) were used for immunofluorescence. The tissues were dewaxed and rehydrated through a series of xylene and ethanol changes. For antibody staining, antigen retrieval was performed on the slides by incubating them in a steamer for 40 min in citrate buffer (Vector Labs, H-3300-250). The slides were washed in water and then blocked using 10% goat serum in PBS for 1 h at room temperature before adding primary antibodies diluted in the blocking buffer. Primary antibody (anti-GCSc 1:10 (SantaCruz, sc390811) and anti-CD45 1:200 (Proteintech, 31243-1-AP)) incubation was carried out overnight at 4 °C. Secondary antibodies (AlexaFluor 594 anti-mouse 1:500 (Invitrogen, A11005) and AlexaFluor 488 anti-rabbit 1:500 (Invitrogen, A11034)) were added to blocking buffer and incubated at room temperature for 1 h. For immunofluorescence, the second-to-last wash included 1 µg ml⁻¹ 4′,6′-diamidino-2-phenylindole dihydrochloride (DAPI; Sigma-Aldrich D9542) in PBS to stain cell nuclei. Immunofluorescence tissues were mounted with Immuno-Mount (Thermo Fisher, 9990402) and coverslipped for imaging. Immunofluorescence-stained slides were imaged using a Zeiss Axioimager.M2m with an AxioCam MRm camera and the AxioVision program for imaging.

### Cell culture

Cell lines were maintained in RPMI 1640 (Thermo Fisher, 11875119) with 5% FBS (Sigma-Aldrich, SH30396.03) and 1% penicillin and streptomycin (Thermo Fisher, 15070063). Cell lines were obtained from the American Type Culture Collection (ATCC), the National Cancer Institute Division of Cancer Treatment and Diagnosis (NCI-DCTD) or donated from J. Brugge, S. Mello, B. Altman and P. Rappold. PC3 wild-type and PC3 GGT1⁺ cell lines were a gift from M. Hannigan[40]. The authenticity of the HCC-1806, MDAMB231, MDAMB468, JIMT-1 and A549 cell lines was confirmed by STR profiling. The authenticity of the remaining cell lines was not tested. All cell lines tested negative for mycoplasma contamination using a MycoAlert Mycoplasma Detection kit (Lonza, LT07-418). For experiments involving alterations to the culture medium, cells were seeded in regular RPMI 1640 with 5% dialysed FBS (Gibco, 26400044) and 1% penicillin and streptomycin. RPMI 1640 medium without L-glutamine, L-cysteine, L-cystine and L-methionine was used as the base medium for different medium

conditions (MP Biomedicals, 1646454). L-glutamine (Fisher Scientific, 25030081), L-methionine (Sigma-Aldrich, M5308), L-cystine (Sigma-Aldrich, C7602), GSH (Sigma-Aldrich, G4251) or cysteinylglycine (Bachem, 4002969) was added to each condition at the indicated concentrations. GGsTop (Tocris, 4452; WuXi, 926281-37-0; or MCE, HY-108467), acivicin (MCE, HY-W016586), OU749 (Cayman, 13804), auranofin (MCE, HY-B1123), erastin (MCE, HY-15763), vincristine sulfate (MCE, HY-N0488), ferrostatin-1 (Fer-1; MCE, HY-100579), RSL3 (MCE, HY-10018A) and MK-571 (MCE, HY-19989) were diluted in dimethyl sulfoxide (DMSO; Sigma-Aldrich, 472301) at 10–20 mM and added at the indicated concentrations. DL-Buthionine-($S,R$)-sulfoximine (BSO; MCE, HY-106376) and trolox (Sigma-Aldrich, 238813) were diluted in water at 100 mM and added at the indicated concentrations. Cells were seeded at 50,000 per well in 6-well plates and at 10,400 per well in 24-well plates. After allowing the cells to attach for 24 h, the medium was aspirated, the wells were washed once with PBS and the indicated medium was added. For experiments in 384-well plates, 500 cells were seeded per well, and after 24 h, the medium was switched using a series of 4 aspiration–wash steps with the desired final medium using Multidrop Combi (Thermo). Co-culture experiments with PC3 WT and GGT1$^+$ cell lines were performed using 6-well (24 mm) PET inserts with 0.4 µm pores (VWR, 76313-902 or Corning, 354570). For these experiments, cells were seeded at the same density (12,992 cells per cm$^2$) in both the bottom well and the insert. For cell number measurements in 6-well plates, the medium was aspirated, the wells were rinsed with PBS and the cells were detached by incubating with 500 µl of 0.25% trypsin (Thermo Scientific, 25200056) for 20 min at 37 °C. Trypsin was neutralized with 500 µl complete medium, the cells were resuspended and 100 µl of the suspension was mixed with 6.9 ml Beckman Coulter Isoton II diluent (Fisher Scientific, NC2470899) for counting with a Beckman Coulter Z1 Coulter Cell Particle counter. For cell number measurements in 24-well plates, unstained plates were scanned in bright field using SparkCyto (Tecan) at 37 °C under 5% CO$_2$. For experiments in 384-well plates, wells were aspirated, rinsed with PBS and incubated with a fixative and staining solution containing 3.7% formaldehyde (Fisher Scientific, F75P-1GAL) and 5 µg ml$^{-1}$ Hoechst 33342, trihydrochloride, trihydrate (Invitrogen, H1399). After 30 min, wells were aspirated, washed with PBS and plates were sealed with adhesive foil (VWR, 60941-124). Plates were imaged on a CellInsight CX5 HCS platform (Thermo), and counts of nuclei were used as the readout.

### Generation of KO and knockdown cell lines

Optimized sgRNA oligonucleotides[64] were cloned into lentiCRISPRv2 (ref. 65) (Addgene, 52961) for CRISPR–Cas9 KO or into pLV hU6-sgRNA hUbC-dCas9-KRAB-T2a-Puro[66] (Addgene, 71236) for CRISPRi knockdown, as previously described[41]. HEK293T cells were transfected with 1 µg lentiCRISPRv2 containing the sgRNA insert, 0.67 µg psPAX2 (Addgene, 12260) and 0.33 µg pVSVg (Addgene, 8454) using 6 µg polyethylenimine (Sigma-Aldrich, 919012) in Opti-MEM (Thermo Fisher, 31985062). Supernatants containing lentiviral particles were collected and replaced with fresh medium daily for 3 consecutive days. The pooled lentiviral supernatant was filtered through 0.45 µm PVDF syringe filters (Whatman, 6746-2504), aliquoted and stored at –80 °C. For transduction, 200 µl lentiviral supernatant was thawed on ice and added to HCC-1806 cells together with 8 mg ml$^{-1}$ Polybrene (hexadimethrine bromide; Sigma-Aldrich, H9268) in 6-well plates. Cells were then allowed to recover for 24 h in fresh medium before antibiotic selection with 2 µg ml$^{-1}$ puromycin for at least 4 days, after which they were expanded or used for subsequent experiments. To isolate single-cell clones, antibiotic-selected polyclonal GGT1 KO cells were seeded at low density (50 cells per 15 cm dish) and grown for 21–28 days. Then colonies were individually collected by trypsinization using 8 mm cloning cylinders (Pyrex, 09-552-21) and transferred to individual wells for expansion. The following sgRNAs for GGT1 CRISPRi were used:

sgGGT1 1 (HL-ID: 3) forward: 5′-CACCGGTGCCTCCCACTGTCCGCCG-3′; sgGGT1 1 (HL-ID: 3) reverse: 5′-CACCGGTCTGGGCTCACCCGACGCC-3′; sgGGT1 2 (HL-ID: 5) forward: 5′-AAACCGGCGGACAGTGGGAGGCACC-3′; and sgGGT1 2 (HL-ID: 5) reverse: 5′-AAACGGCGTCGGGTGAGCCCAG ACC-3′. The following sgRNAs for GGT1 CRISPR–Cas9 KO were used: sgGGT1 1 (HL-ID: 6) forward: 5′-CACCGTGCGGGACGGTGGCTCTG-3′; sgGGT1 1 (HL-ID: 6) reverse: 5′-AAACCAGAGCCACCGTCCCGCAC-3′; sgGGT1 2 (HL-ID: 10) forward: 5′-CACCGCCCAGTGCGCCGCTCAG-3′; and sgGGT1 2 (HL-ID: 10) reverse: 5′-AAACCTGAGCGGCGCAC TGGGC-3′. The following sgRNAs for GCLC CRISPR–Cas9 KO were used: sgNTC forward: 5′-CACCGGAGGCTAAGCGTCGCAA-3′; sgNTC reverse: 5′-AAACTTGCGACGCTTAGCCTCc-3′; sgGCLC-a forward: 5′-CACCGCATACTCACCTGAAGCGA-3′; sgGCLC-a reverse: 5′-AAACT CGCTTCAGGTGAGTATGC-3′; sgGCLC-b forward: 5′-CACCGAAATATCC GACATAGGAG-3′; and sgGCLC-b reverse: 5′-AAACCTCCTATGTCGG ATATTTC-3′.

### Flow cytometry

To quantify cell proliferation, 300,000 cells were plated in 100 mm dishes and treated as specified. At the end point, attached cells were pulsed with 10 mM BrdU (Sigma-Aldrich, B9285) for 30 min in their culture medium. Labelled cells were then detached by trypsinization, pelleted, washed once with PBS, resuspended in 300 µl ice-cold PBS and fixed by adding 700 µl ice-cold absolute ethanol while gently vortexing. Fixed cells were stored at –20 °C. For labelling, the fixed cells were thawed and washed for 5 min with 1 ml of 1% BSA (Sigma-Aldrich, A7906) in PBS. The cells were then centrifuged at 10,000g for 2 min and incubated with 1 ml of 2 M HCl and 0.5% Triton X-100 (Sigma-Aldrich, 93443) for 30 min with gentle agitation. Then the cells were centrifuged and washed once with 1 ml of 0.1 M borax (Sigma-Aldrich, 71997), followed by another wash with 1% BSA in PBS. After centrifugation and thorough removal of the supernatant, the pellets were incubated for 1 h in 60 µl staining solution (1% BSA and 0.5% Tween-20 in PBS) containing 3 µl anti-BrdU–FITC monoclonal antibody (BioLegend, 364104, clone 3D4). The samples were subsequently washed with 1 ml of 1% BSA and 0.5% Tween-20 in PBS, centrifuged and the pellets were resuspended in 1 ml of PI solution (20 µg ml$^{-1}$ propidium iodide (Sigma-Aldrich, P4170), 0.5 mg ml$^{-1}$ RNase A (Sigma-Aldrich, R5503) and 0.1% Triton X-100 in PBS). The cells were incubated overnight (approximately 16 h), and then 10,000 events were analysed by flow cytometry using a BD Accuri C6 Plus flow cytometer (BD Biosciences) and FCS Express 7 Research (DeNovo Software).

For quantifying cell death, 50,000 HCC-1806 cells were seeded into 6-well plates and treated as indicated. At the end point, cells were collected by trypsinization (the culture medium and PBS washes were collected and re-added to each sample) and washed once with assay buffer (10 mM HEPES, 140 mM NaCl and 3.3 mM CaCl$_2$, pH 7.4). Then the pellets were stained for 20 min in 100 µl assay buffer containing 5 µg ml$^{-1}$ propidium iodide (Sigma-Aldrich, P4170) and 3 µl Annexin V–FITC (BioLegend, 0640906). Then cells were diluted with 0.5–1 ml assay buffer, and 10,000 events were analysed by flow cytometry using a BD Accuri C6 Plus flow cytometer (BD Biosciences) and FCS Express 7 Research (DeNovo Software).

For quantifying lipid peroxidation, 95,000 cells were seeded into 12-well plates and treated as indicated. At the end point, 2.5 µM Bodipy 581/591 C11 (Cell Signaling, 95978) was added directly to each well and incubated for 30 min. Wells were then washed with PBS and collected by trypsinization. Pellets were resuspended in 300 µl PBS, and 10,000 events were analysed by flow cytometry using a BD Accuri C6 Plus flow cytometer (BD Biosciences) and FCS Express 7 Research (DeNovo Software).

### Immunoblot assays

Cell lysates from HCC-1806, PC3 WT and PC3 GGT1$^+$ cells were obtained by placing PBS-washed cell plates on ice and scraping with RIPA buffer

(Thermo Scientific, 89900) containing Halt Protease and Phosphatase Inhibitor cocktail (Thermo Scientific, 1861280). Cell lysates were incubated for 30 min on ice and then centrifuged at 12,000$g$ for 15 min at 4 °C. The supernatant was collected and stored at −80 °C. Extracted proteins were quantified using a Pierce BCA Protein Assay kit (Thermo Scientific, 23225). Lysates were combined with 6× Laemmli SDS sample buffer (Boston BioProducts, BP-111R) containing 5% 2-mercaptoethanol (VWR Life Science, M131), then loaded onto 4–20% Criterion TGX pre-cast gels (Bio-Rad, 5671093). Each lane received 40 μg protein for HCC-1806 samples or 75 μg for PC3 samples. Separated proteins were transferred onto Immobilon-P Transfer membranes (MilliporeSigma, IPVH00010), blocked for 1 h using 5% BSA in TBS−0.5% Tween-20 and stained overnight with primary rabbit-anti-GCLC (Sigma-Aldrich, HPA036359) or affinity-purified rabbit-anti-GGT1 (GGT129, as previously described[63]) 1:1,000 in BSA 2.5% overnight at 4 °C. Stained membranes were washed three times with TBS-T and stained with HRP-linked secondary antibody donkey-anti-rabbit (Amersham, NAV934) 1:5,000 in BSA 2.5% for 1 h. Membranes were washed three more times with TBS-T, and then the antibody-stained protein signal was amplified and visualized using SuperSignal (Thermo Fisher, 34577). Blots were imaged with a ChemiDoc MP Imaging system (Bio-Rad). Membranes were stripped for 20 min at room temperature using a Blot Restore Membrane Rejuvenation kit (Sigma-Aldrich, 2520-M), then washed twice with TBS-T, re-blocked with BSA 5% for 30 min and incubated with mouse-anti-β-actin (Sigma-Aldrich, A1978) 1:3,000 in BSA 2.5% for 4 h at room temperature and then incubated with a sheep-anti-mouse (Amersham, NA931) secondary antibody at 1:5,000 in BSA 2.5% for 2 h.

## Drug screening

MAPS was performed as previously described[41]. In brief, 500 cells were seeded per well in 384-well plates, and after 24 h, the medium was switched using a series of 4 aspiration–wash steps with the desired final medium. Compound libraries were gifts from J. Brugge and the ICCB-L. Next, 100 nl of compounds were transferred using a JANUS MDT Workstation (Revvity) and a 384-well pin tool (V&P Scientific). After 96 h, wells were aspirated, washed with PBS, and a fixative/staining solution containing 3.7% formaldehyde (Fisher Scientific, F75P-1GAL) and 5 μg ml⁻¹ Hoechst 33342, trihydrochloride, trihydrate (Invitrogen, H1399) was added. After 30 min, wells were aspirated, PBS was added and plates were sealed with adhesive foil (VWR, 60941-124). Plates were imaged using a CellInsight CX5 HCS platform (Thermo), and nuclei were identified as a readout of cell counts. Data post-processing was conducted using R scripts.

## GGT activity assay

Cell cultures were washed and scraped with PBS. The use of trypsin for cell detachment was avoided to prevent proteolytic degradation of cell-surface GGT. Detached cells were collected and centrifuged at 300$g$ for 5 min at room temperature. Then pellets were resuspended in lysis buffer (Tris-HCl 100 mM and 0.5% Triton X-100, pH 8) containing Halt Protease and Phosphatase Inhibitor cocktail (Thermo Scientific, 1861280), thoroughly vortexed for 1 min and incubated for 15 min on ice. Lysates were cleared by centrifugation at 12,000$g$ for 15 min at 4 °C and the supernatants were stored at −80 °C until analysis. Tissues were lysed by transferring them to pre-filled bead mill tubes (Fisher Scientific, 15-340-154) containing lysis buffer (Tris-HCl 100 mM and 0.5% Triton X-100, pH 8) containing Halt Protease and Phosphatase Inhibitor cocktail and homogenized using a bead mill (VWR) for 10 s. Tissue lysates were then rotated for 15 min at 4 °C and centrifuged at 12,000$g$ for 15 min at 4 °C. Supernatants of cells or tissues were stored at −80 °C until analysis. Extracted proteins were quantified using a Pierce BCA Protein Assay kit. GGT reaction assays were carried out by using the GpNA method[67]. In brief, cell or tissue protein lysates were assayed in transparent 96-well plates (Greiner, 655101) in the presence of 1 mM of the GGT substrate ʟ-glutamic acid γ-p-nitroanilide (GpNA; Cayman, 36209) and 20 mM of the transpeptidation acceptor glycyl-glycine (Sigma-Aldrich, G1002) in Tris-HCl 100 mM, pH 8, buffer. GGT catalyses the breakdown of the γ-glutamyl bond in the substrate and generates p-nitroanilide (pNA), which is monitored kinetically by an increase in absorbance at 418 nm. P-nitroaniline (Sigma-Aldrich, 185310) was used to generate a standard curve (0–40 nmoles) for the calculation of GGT activity. Absorbance (418 nm) was assessed using a plate reader set at 37 °C with a total reading time of 45–60 min and 1–5 min intervals between reads. GGT activity was calculated by obtaining the slope of the linear range of each sample (Abs per min) and dividing by the slope of a 4-nitroaniline standard curve (Abs per nmoles pNA) and by the protein mass (mg) to determine the reaction rate of the sample (nmoles pNA min⁻¹ mg⁻¹). The optimal protein mass for the GGT assay was experimentally determined for each cell line or mouse tissue (HCC-1806 lysates or tumour xenografts: 200 μg; PC3 cell line: 100 μg; PC3 GGT1⁺: 1 μg; kidney: 0.5 μg; spleen: 150 μg; liver: 250 μg; seminal vesicle: 10 μg; pancreas: 2.5 μg; lungs: 2.5 μg; muscle: 200 μg; testis: 50 μg; epididymis: 10 μg; PyMT breast tumour: 50 μg; heart: 150 μg; mammary fat pad: 150 μg; brain: 80 μg; large intestine: 80 μg; brown adipose tissue: 300 μg).

## GGT histochemical assay

Histochemical staining of GGT was performed as previously described[68], with modifications. Cells were seeded into 6-well plates and cultured in complete medium. At the end point, cells were washed 3 times with saline and stained for 20 min at room temperature using a saline solution containing 0.2 mM ʟ-glutamic acid γ-(4-methoxy-β-naphthylamide) (GMNA; Sigma-Aldrich, G0141), 20 mM glycyl-glycine (Sigma-Aldrich, G1002), 25 mM Tris (Sigma-Aldrich, 93362) and 1.2 mM FastBlue (Sigma-Aldrich, F3378). For negative controls, the staining solution included 5 mM serine (Sigma-Aldrich, S4311) and 10 mM sodium borate (Sigma-Aldrich, 71997). After incubation, cells were rinsed with saline and incubated with 100 mM CuSO₄ (Sigma-Aldrich, C1297) for 2 min. Following a final saline wash, 50% glycerol was added to the wells, and bright-field images were taken using an inverted microscope.

## GSH quantification

For some experiments, the GSH concentration was determined using a GSH-Glo Glutathione Assay (Promega, V6912). In brief, 6,250 cells were seeded into 96-well white cell culture-treated plates (Falcon, 353296), treated for 24 h as indicated and labelled following the manufacturer's instructions. The luminescence of samples and of the GSH standard curve was detected using a Spark Multimode microplate reader (Tecan).

## RNA analysis

mRNA was isolated from cells or tissues using an E.Z.N.A. total RNA kit I (Omega Bio-Tek, R6834-02). For gene expression analysis, 1 μg RNA was used for cDNA synthesis using a qScript cDNA Synthesis kit (Quanta Bio, 66196756). The expression of target genes was analysed via quantitative real-time PCR with a QuantaStudio 5 qPCR machine (Applied Biosystems, Thermo Fisher Scientific).

## Mouse serum analysis

Mice were anaesthetized with isoflurane, after which blood was collected via the retro-orbital venous sinus into BD microtainer tubes (BD, 365967). Serum was isolated from the blood by centrifuging blood samples at 10,000$g$ for 5 min and stored at −80 °C. VRL Animal Health Diagnostics carried out the analysis of biomarkers of liver damage.

## Metabolite analysis

For metabolite analysis of cultured cancer cells, cells were seeded in 6-well plates and treated as specified. At the end point, cells were rapidly washed with ice-cold PBS and incubated at −80 °C for 30 min with extraction solvent at a ratio of 1 ml per 1 × 10⁶ cells. The cells were then scraped

on ice, transferred to pre-chilled microcentrifuge tubes, centrifuged at 15,000$g$ for 20 min and the supernatants were collected. Conditioned medium was extracted by taking 10 µl of the medium supernatant and mixing it with 390 µl of the extraction solvent. Control medium samples were generated by incubating the treatment medium in cell-free 6-well plates under the same conditions and duration as the cell culture samples. Extracts of the medium were incubated at –80 °C for at least 15 min before being centrifuged at 15,000$g$ for 20 min. All extracts were stored at –80 °C. Metabolite profiling of reduced thiol-containing metabolites was carried out in the presence of NEM. The data shown reflect the detection of their corresponding NEM adducts (for example, NEM-cystine, NEM-cysteinylglycine and NEM-GSH). tGSH was calculated as the sum of reduced GSH and two times the amount of oxidized glutathione (tGSH = GSH + 2 × GSSG). The extraction solvent consisted of 80% methanol and 20% aqueous solution (pH 7.00) containing 10 mM ammonium formate (Sigma-Aldrich, 70221-25G-F) and 25 mM NEM (Thermo Scientific, 040526.06). The final concentrations of ammonium formate and NEM in the extraction solvent were 2 mM and 5 mM, respectively. For inverse tracing experiments, HCC-1806 cells were seeded in T75 flasks and cultured for >7 days in cystine-free medium supplemented with heavy isotope-labelled $^{13}C_2$-cystine (Cambridge Isotope Laboratories, CLM-520-0). At the end of this >7-day labelling period ($t = 0$ h), samples were collected to verify complete incorporation of the heavy label into cellular metabolites. The remaining cells were collected by trypsinization, washed three times with PBS and seeded into 6-well plates using cystine-free medium containing unlabelled GSH. After 3 days in the presence of unlabelled GSH ($t = 72$ h), cells were extracted for metabolomic analysis as described above.

For quantification of sulfur-containing metabolites in cultured cancer cells and mouse serum, 10 µl samples were extracted with 90 µl of ice-cold extraction solvent (80% methanol and 20% $H_2O$) containing isotope-labelled internal standards (36 µM $^{13}C_2$,$^{15}N$-GSH, 0.92 µM $^{13}C_4$$^{15}N_2$-GSSG and 10 µM $D_4$-cystine). Following 30 min of incubation at 4 °C, extracts were cleared by centrifugation (17,000$g$, 20 min, 4 °C) and then supernatants were analysed by liquid chromatography and mass spectrometry (LC–MS). For the quantification of these metabolites in TIF samples, 5 µl samples were extracted in 45 µl ice-cold extraction solvent (75% acetonitrile, 25% methanol and 0.1% formic acid) containing isotope-labelled internal standards (36 µM $^{13}C_2$,$^{15}N$-GSH, 0.92 µM $^{13}C_4$,$^{15}N_2$-GSSG and 10 µM $D_4$-cystine). Extracted samples were vortexed at 4 °C for 10 min, cleared by centrifugation (17,000$g$, 10 min, 4 °C) and then supernatants were analysed by LC–MS. For chromatographic metabolite separation, a Vanquish ultra-performance liquid chromatography (UPLC) system was coupled to a Q Exactive HF (QE-HF) mass spectrometer equipped with heated electrospray ionization (HESI; Thermo Fisher Scientific). Samples were run on an Atlantis Premier BEH Z-HILIC VanGuard FIT column (2.5 µm, 2.1 mm × 150 mm, Waters). Mobile phase A was 10 mM $(NH_4)_2CO_3$ and 0.05% $NH_4OH$ in $H_2O$, whereas mobile phase B was 100% acetonitrile. The column chamber temperature was maintained at 30 °C. The mobile phase parameters were set according to the following gradient: 0–13 min of 80% to 20% of mobile phase B; 13–15 min of 20% of mobile phase B. The ESI ionization mode was negative, and the MS scan range ($m/z$) was set to 65–975. The mass resolution was 120,000 and the automatic gain control (AGC) target was 3 × 10⁶. The capillary temperature and capillary voltage were maintained at 320 °C and 3.5 kV, respectively. A volume of 5 µl of each sample was loaded for metabolite detection. The LC–MS metabolite peaks were manually integrated and identified using El-Maven (v.0.12.0) by matching with a previously established in-house library[33].

For metabolic analysis of cultured cancer cells and tumours, frozen tumour tissue was homogenized in 80% methanol containing 20 mM NEM using a Precellys cold tissue homogenizer (Bertin) at a ratio of 20 mg tissue to 1 ml solvent. Following homogenization, samples were transferred to –80 °C for 30 min and then placed on regular ice for 30 min with vortexing every 10 min. Next, samples were centrifuged at

17,000$g$ for 10 min and 800 µl supernatant was dried in a vacuum evaporator (Thermo). Samples were reconstituted in 90 µl of 50% acetonitrile (Fisher Scientific, A955) and transferred to glass vials for LC–MS analysis. For mouse serum collection, the animals were anaesthetized with isoflurane, and blood was collected via the retro-orbital venous sinus into BD microtainer tubes (BD, 365967). Serum was obtained by centrifuging blood samples at 10,000$g$ for 5 min and stored at –80 °C. Then 20 µl serum was mixed with 2 µl 200 mM NEM (Thermo Scientific, 040526.06) and extracted with 80% methanol. Next, 900 µl serum extract was dried in a vacuum evaporator (Thermo), reconstituted in 90 µl of 50% acetonitrile (A955, Fisher Scientific) and transferred to glass vials for LC–MS analysis. LC–MS analysis was carried out by the URMC Metabolomics Resource. For LC–MS analysis, metabolite extracts were analysed by high-resolution mass spectrometry with an Orbitrap Exploris 240 (Thermo) coupled to a Vanquish Flex liquid chromatography system (Thermo). A volume of 2 µl samples was injected into a Waters XBridge XP BEH Amide column (150 mm length × 2.1 mm i.d., 2.5 µm particle size) maintained at 25 °C, with a Waters XBridge XP VanGuard BEH Amide (5 mm × 2.1 mm i.d., 2.5 µm particle size) guard column. For positive-mode acquisition, mobile phase A was 100% LC–MS-grade $H_2O$ with 10 mM ammonium formate and 0.125% formic acid. Mobile phase B was 90% acetonitrile with 10 mM ammonium formate and 0.125% formic acid. For negative-mode acquisition, mobile phase A was 100% LC–MS-grade $H_2O$ with 10 mM ammonium acetate, 0.1% ammonium hydroxide and 0.1% medronic acid (Agilent). Mobile phase B was 90% acetonitrile with 10 mM ammonium acetate, 0.1% ammonium hydroxide and 0.1% medronic acid. The gradient was 0 min, 100% B; 2 min, 100% B; 3 min, 90% B; 5 min, 90% B; 6 min, 85% B; 7 min, 85% B; 8 min, 75% B; 9 min, 75% B; 10 min, 55% B; 12 min, 55% B; 13 min, 35%, 20 min, 35% B; 20.1 min, 35% B; 20.6 min, 100% B; 22.2 min, 100% B, all at a flow rate of 150 µl min⁻¹, followed by 22.7 min, 100% B; 27.9 min, 100% B at a flow rate of 300 µl min⁻¹, and finally 28 min, 100% B at flow rate of 150 µl min⁻¹, for a total length of 28 min. The H-ESI source was operated in positive mode at spray voltage 3,500 or negative mode at spray voltage 2,500 with the following parameters: sheath gas 35 a.u., aux gas 7 a.u., sweep gas 0 a.u., ion transfer tube temperature 320 °C, vaporizer temperature 275 °C, mass range 70 to 1,000 $m/z$, full scan MS1 mass resolution of 120,000 FWHM, RF lens at 70% and standard AGC. Data-dependent MS2 fragmentation for compound identification and annotation was performed via the AquireX workflow (Thermo Scientific), which comprised three deep scans with MS1 resolution at 60,000 and MS2 resolution at 15,000. LC–MS data were analysed using Compound Discover (v.3.3, Thermo Scientific) and El-Maven software[69] for peak area determination and compound annotation. Compounds were annotated by matching to LC–MS method-specific retention time values of external standards and MS² spectral matching to external standards and the mzCloud database (Thermo Scientific).

### Human biospecimens, clinical data and metabolite analysis of human samples

Human biospecimens and deidentified clinical data were provided by the Wilmot Cancer Institute Biobank Shared Resource (BSR) at the University of Rochester. All samples were collected under Institutional Review Board-approved protocols (STUDY61977 and STUDY7108), and all participants provided written informed consent. A total of 16 participants were enrolled, all of whom were women with a median age of 64 years (range of 25–88 years). Neoadjuvant treatment was administered in 4 participants (25%). The cohort included 81% white, 6% Black and 13% with race not reported. Most tumours were invasive ductal carcinoma (94%) with 6% classified as other histology. Oestrogen receptor positivity was observed in 81% of cases and progesterone receptor positivity in 62%. HER2 status was distributed as 0 in 56%, 1+ in 32%, 2+ in 6% and 3+ in 6%. Tumour staging showed 44% T1 and 56% T2 with no T3 or T4 disease. Nodal staging included 38% N0, 38% N1, 6% N2

and 18% NX. All participants were M0 at diagnosis with no metastatic disease. Tumour grade was 1 in 12%, 2 in 50% and 3 in 38%.

Peripheral whole blood was collected in the perioperative setting via a venous catheter immediately before surgery using $K_2$-EDTA-coated tubes (BD Biosciences, 367899). Samples were centrifuged at 2,500$g$ for 10 min at 4 °C to separate plasma. The plasma supernatant was aliquoted, snap-frozen in liquid nitrogen and stored at –80 °C before analysis.

Immediately following excision, tumour specimens were delivered to Surgical Pathology and reviewed by certified pathologists at the University of Rochester Medical Center, who allocated remnant malignant tissue for research. Research biospecimens were promptly transferred to the BSR laboratory, rinsed with ice-cold PBS and gently blotted dry. Tumour tissues (100–500 mg) were dissected into smaller fragments to cover the filter surface of 0.22 µm nylon-filtered microcentrifuge tubes (Corning, 8160), taking care to minimize mechanical cell lysis. Tubes were centrifuged at 300$g$ for 5 min at 4 °C to collect TIF. The isolated TIF was snap-frozen in liquid nitrogen and stored at –80 °C before analysis.

Quantitative metabolite profiling of the TIF and serum human patient samples was performed based on a protocol adapted from a previous study[28]. In brief, chemical standard libraries of 149 metabolites in seven pooled libraries were prepared and serially diluted in high-performance liquid chromatography (HPLC)-grade water from in a dilution series from 5 mM to 1 µM to generate external standard pools, which are used for calibration of isotopically labelled internal standards and to quantify concentrations of metabolites for which internal standards were not available. We then measured 0.3 µl of each sample using a 1 µl Hamilton syringe (Hamilton, 80135) and extracted these volumes in 20 µl of a 75:25:0.1 HPLC grade acetonitrile–methanol–formic acid extraction mix with the following labelled stable isotope internal standards: $^2H_9$-choline (DLM-549), $^{13}C_4$-3-hydroxybutyrate (CLM-3853), $^{13}C_6$,$^{15}N_2$-cystine (CNLM-4244), $^{13}C_3$-lactate (485926, Sigma-Aldrich), $^{13}C_6$-glucose (CLM-1396), $^{13}C_3$-serine (CLM-1574), $^{13}C_2$-glycine (CLM-1017), $^{13}C_5$-hypoxanthine (CLM-8042), $^{13}C_2$,$^{15}N$-taurine (CNLM-10253), $^{13}C_3$-glycerol (CLM-1510), $^2H_3$-creatinine (DLM-3653), $^{13}C_{10}$-kynurenic acid (80445, Sigma-Aldrich), $^{13}C_{10}$,$^{15}N_5$-adenosine (CNLM-3806), $^{13}C_8$-indole-3-carboxaldehyde (CLM-10745), $^{15}N_4$-inosine (NLM-4264), $^{13}C_{10}$-kynurenine (CLM-9884) and $^{13}C_4$-methylmalonic acid (CLM-9426). All standards were obtained from Cambridge Isotope Laboratory unless stated otherwise. Samples in extraction mix were vortexed for 10 min at 4 °C and centrifuged at 15,000 rpm for 10 min at 4 °C to pellet insoluble material. Next, 18 µl of the soluble polar metabolite supernatant was transferred to sample vials for analysis by LC–MS as previously described[4,28]. Once LC–MS analysis was performed, Skyline software was used for metabolite identification. External standard libraries were used to confirm the $m/z$ and retention time for each metabolite. For quantitative analysis, when internal standards were available, external standard libraries were used to quantify concentrations of isotopically labelled internal standards in the extraction mix. Once internal standard concentrations were obtained, the peak areas of the unlabelled metabolites in the samples of culture medium were compared with the peak area of the quantified internal standard to determine the metabolite concentration in the sample.

For metabolites for which an internal standard was not present in the extraction mix, external standard libraries were used to perform analysis of relevant metabolite concentrations. In brief, the peak area of the metabolite was normalized to the peak area of an isotopically labelled internal standard with similar elution time, both in samples from culture medium and external standard library dilutions. Using the external standard library dilutions, we created a standard curve based on the linear relationship of the normalized peak area and the concentration of the metabolite, excluding those metabolites with an $r^2 < 0.95$. This standard curve was then used to interpolate the concentration of the metabolite in the samples.

## Proteomic analysis

Cells were seeded in 10-cm dishes and cultured for 7 days in cystine-free medium supplemented with heavy isotope-labelled $^{13}C_6$,$^{15}N_2$-cystine (Cambridge Isotope Laboratories, CNLM-4244-H). At the end of this 7-day labelling period ($t = 0$ h), samples were collected to verify complete incorporation of the heavy label into cellular proteins. The remaining cells were washed three times with PBS and transferred to 6-well plates containing cystine-free medium supplemented with unlabelled GSH. After an additional 4 days ($t = 96$ h), cells were collected by trypsinization, pelleted, snap-frozen and stored at –80 °C. Cell pellets were then lysed in 50 µl of 5% SDS, 100 mM TEAB and sonicated (QSonica) for 5 cycles, with a 1 min resting period on ice after each cycle. Samples were then centrifuged at 15,000$g$ for 5 min to pellet cellular debris, and the supernatant was collected. Protein concentration was determined using a BCA kit (Thermo Scientific), after which samples were diluted to 1 mg ml$^{-1}$ in 5% SDS and 50 mM TEAB. Next, 25 µg protein from each sample was reduced with 2 mM dithiothreitol, followed by incubation at 55 °C for 60 min. Iodoacetamide was added to 10 mM and incubated in the dark at room temperature for 30 min to alkylate the proteins. Phosphoric acid was added to 1.2%, followed by six volumes of 90% methanol, 100 mM TEAB. The resulting solution was added to S-Trap micros (Protifi) and centrifuged at 4,000$g$ for 1 min. The S-Traps containing trapped protein were washed twice by centrifuging through 90% methanol and 100 mM TEAB. Then 1 µg of trypsin was brought up in 20 µl of 100 mM TEAB and added to the S-Trap, followed by an additional 20 µl of TEAB to ensure the sample did not dry out. Samples were incubated at 37 °C overnight. The next morning, the S-Trap was centrifuged at 4,000$g$ for 1 min to collect the digested peptides. Sequential additions of 0.1% trifluoroacetic acid (TFA) in acetonitrile and 0.1% TFA in 50% acetonitrile were added to the S-trap, centrifuged and pooled. Samples were frozen and dried in a Speed Vac (Labconco), then re-suspended in 0.1% TFA before MS analysis.

Peptides were injected onto a 75 µm × 2 cm trap column (Thermo Fisher) before re-focusing on an Aurora Elite 75 µm × 15 cm C18 column (IonOpticks) using a Vanquish Neo UHPLC (Thermo Fisher) connected to an Orbitrap Astral mass spectrometer (Thermo Fisher). Solvent A was 0.1% formic acid in water, whereas solvent B was 0.1% formic acid in 80% acetonitrile. Ions were introduced to the mass spectrometer using an Easy-Spray source operating at 2 kV. The gradient began at 1% B and ramped to 5% B in 0.1 min, increased to 30% B in 12.1 min, increased to 40% in 0.7 min and finally increased to 99% B in 0.1 min and was held for 2 min to wash the column for a total run time of 15 min. After each run was completed, the column was re-equilibrated with 1% B before the next injection. The Orbitrap Astral was operated in data-independent acquisition (DIA) mode, with MS1 scans acquired in the Orbitrap at a resolution of 240,000, with a maximum injection time of 5 ms over a range of 380–980 $m/z$. DIA MS2 scans were acquired in the Astral mass analyser with a 3 ms maximum injection time using a variable windowing scheme, using 2 Da windows from 380 to 680 $m/z$, 4 Da windows from 680 to 800 $m/z$ and 8 Da windows from 800 to 980 $m/z$. The HCD collision energy was set to 25%, and the normalized AGC was set to 500%. Fragment ions were collected over a scan range of 150–2,000 $m/z$. The cycle time was 0.6 s.

For data analysis, the raw data were processed with DIA-NN (v.1.9.2; https://github.com/vdemichev/DIANN)[70]. For all experiments, data analysis was carried out using library-free analysis mode in DIA-NN. To annotate the library, the *Homo sapiens* UniProt 'one protein sequence per gene' database (UP0000005640_9606, downloaded 7 April 2021) was used, with 'deep learning-based spectra and RT prediction' enabled. For precursor ion generation, the maximum number of missed cleavages was set to 1, the maximum number of variable modifications to 1 for Ox(M), the peptide length range to 7–30, the precursor charge range to 2–4, the precursor $m/z$ range to 380–980 and the fragment $m/z$ range to 150–2,000. The quantification was set to 'Robust LC (high precision)'

mode with normalization disabled, MBR enabled, protein inferences set to 'Genes' and 'Heuristic protein inference' turned off. MS1 and MS2 mass tolerances, along with the scan window size, were automatically set by the software. To quantify light and heavy cysteine-containing peptides, the following parameters were included in the 'Additional Options' pane in the DIA-NN software: fixed-mod Carb, 57.021464, C, label; lib-fixed-mod Carb; channels Carb,L,C,0; Carb,H,C,4.007099; original-mods. Precursors were subsequently filtered at library precursor $q$ value (1%), library protein group $q$ value (1%) and posterior error probability (50%). Protein quantification was carried out using the MaxLFQ algorithm as implemented in the DIA-NN R package (https://github.com/vdemichev/diann-rpackage), and the number of peptides quantified in each protein group was counted as implemented in the DiannReportGenerator package (https://github.com/URMC-MSRL/DiannReportGenerator)[71].

## Statistical analysis

All statistical analysis was completed using either R or GraphPad Prism (v.10.6.1).

## Reporting summary

Further information on research design is available in the Nature Portfolio Reporting Summary linked to this article.

## Data availability

Uncropped western blot images are provided in Supplementary Fig. 1. Source data are provided with this paper.

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

**Acknowledgements** We thank S. McBrayer for feedback and discussions; M. Hanigan for assistance with protocols and the donation of key reagents; staff at the Mass Spectrometry Resource Laboratory in the Center for Advanced Research Technologies (CART) and the Histology, Biochemistry, and Molecular Imaging (HBMI) Core at the Center for Musculoskeletal Research (CMSR) at URMC, the Proteomics/Metabolomics Core at Moffitt Cancer Center, which is funded in part by Moffitt's Cancer Center Support Grant (P30CA076292), and University of Chicago Metabolomics Platform (RRID: SCR_022932); and J.Brugge and staff the Ludwig Center at Harvard Medical School for their support. The work described in this publication benefited from the support of the Metabolomics Shared Resource and of the Biobank Shared Resources (BSR) at the Wilmot Cancer Institute, supported in part by a University of Rochester Wilmot Cancer Institute Support Grant (P30CA272302). The content is solely the responsibility of the authors and does not necessarily represent the official views of the National Institutes of Health. This work was supported by the Wilmot Cancer Institute Predoctoral Fellowship (to G.A.), the American Association for Cancer Research and Breast Cancer Research Foundation (20-20-26-HARR to I.S.H.), the Breast Cancer Coalition of Rochester (to I.S.H.), the American Cancer Society (RSG-23-971782-01-TBE to I.S.H.), the Ludwig Center for Metastasis Research (to A.M.), the University of Chicago Comprehensive Cancer Center (to A.M.), the Cancer Research Foundation (to A.M.), the V Foundation (to A.M.), the Pancreatic Cancer Action Network (to G.C.), the Lung Cancer Research Foundation (to J.C.), and NIH grants R01CA269813 (to I.S.H.), R01CA269813-S1 (to M.Z.), R37CA230042 (to G.M.D.), R24AA022057 (to V.V.), R01AA028859 (to Y.C.), R01CA276461 (to A.M.), T32CA009594 and F31CA278362 (to C.S.), and AI150698 (to J.M.).

**Author contributions** F.H., M.Z. and I.S.H. initiated the study, conceived the project, designed experiments, interpreted results and wrote the manuscript. F.H. and M.Z. performed the experiments with assistance from E.T.T., J.C., Z.G.S., F.A., L.D.M., D.T., G.A., A.A.K., V.C.G., Z.Z., Y.G., S.S.M., B.J.A. and S.K.B.-N. N.P.W., Y.P.K. and G.M.D. performed metabolite analyses for the in vitro experiments. B.S., J.M., C.S., G.C. and A.M. performed metabolite analysis for the in vivo experiments. K.A.W. and S.G. performed proteomics analysis for the in vitro experiments. J.J.Z., M.E.O., H.Z. and J.L.C. assisted with the orthotopic allograft tumour experiments. J.J.T. and B.N.M. assisted with the acquisition of patient samples. Y.C., V.V. and B.M.T. provided reagents and expert comments.

**Competing interests** The authors declare no competing interests.

**Additional information**
**Correspondence and requests for materials** should be addressed to Fabio Hecht or Isaac S. Harris.

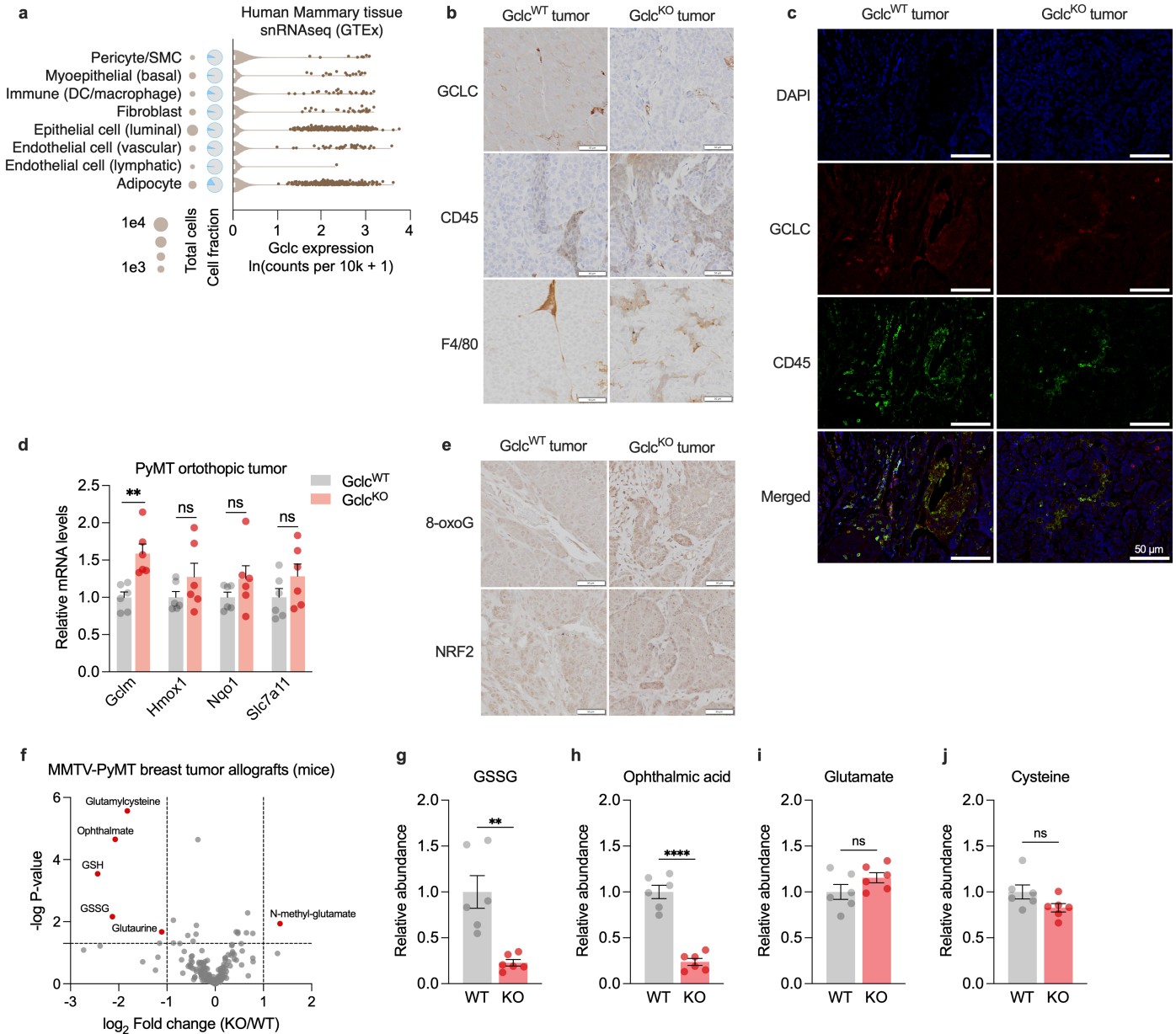

**Extended Data Fig. 1 | Gclc is expressed in non-epithelial cells, and its loss in tumors does cause redox or metabolic alterations. a**, Single-nucleus RNAseq (snRNAseq) of Gclc transcripts from the indicated cells present in human mammary tissue, from previously published data[72]. **b-e**, Autochthonous tumors from MMTV-PyMT Gclc[f/f] Rosa26-CreERT2 mice were excised and orthotopically transplanted into mammary fat pads of wild-type C57BL/6 mice, which were treated with vehicle (corn oil; WT) or 50 mg/kg tamoxifen for 5 days (KO). Representative immunohistochemistry (**b**) and immunofluorescence (**c**) staining of indicated proteins in Gclc WT and Gclc KO mammary tumors

(Scale bar = 50 μm). Relative mRNA expression of *Gclm* (p = 0.0026), *Hmox1*, *Nqo1*, and *Slc7a11* (**d**) and representative immunohistochemistry of indicated proteins (**e**) in *Gclc* WT and *Gclc* KO tumors (n = 6 representative from 3 independent experiments). **f-j**, Volcano plot of metabolic differences (**f**) and levels of GSSG (**g**), ophthalmic acid (**h**), glutamate (**i**), cysteine (NEM-cysteine) (**j**) from *Gclc* WT and *Gclc* KO tumors. Statistical significance was assessed by unpaired two-tailed t-test in **d, f, g-j**. Data represented as mean ± s.e.m., *p-value < 0.05; **p-value < 0.01; ***p-value < 0.001; ****p-value < 0.0001; ns, not significant.

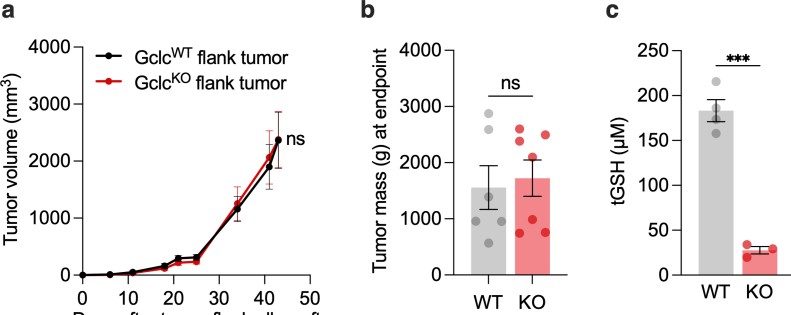

**Extended Data Fig. 2 | Intracellular GSH production is dispensable for tumor growth in the mouse flank. a-c**, Autochthonous tumors from MMTV-PyMT Gclc^{f/f} Rosa26-CreERT2 mice were excised and subcutaneously transplanted into flanks of wild-type C57BL/6 mice, which were treated with vehicle (corn oil; WT) or 50 mg/kg tamoxifen for 5 days (KO). Tumor volume over time (WT, n = 7; KO, n = 8 representative from 2 independent experiments, p = 0.9527) **(a)**, total tumor mass (WT, n = 6; KO, n = 7 representative from 2 independent experiments, p = 0.7441) **(b)**, tGSH levels (p = 0.001, WT, n = 4; KO, n = 3 representative from 2 independent experiments, p = 0.0001) **(c)**. Statistical significance was assessed by by ordinary two-way ANOVA in **a**, and by unpaired two-tailed t-test in **b,c**. Data represented as mean ± s.e.m., ***p-value < 0.001; ns, not significant.

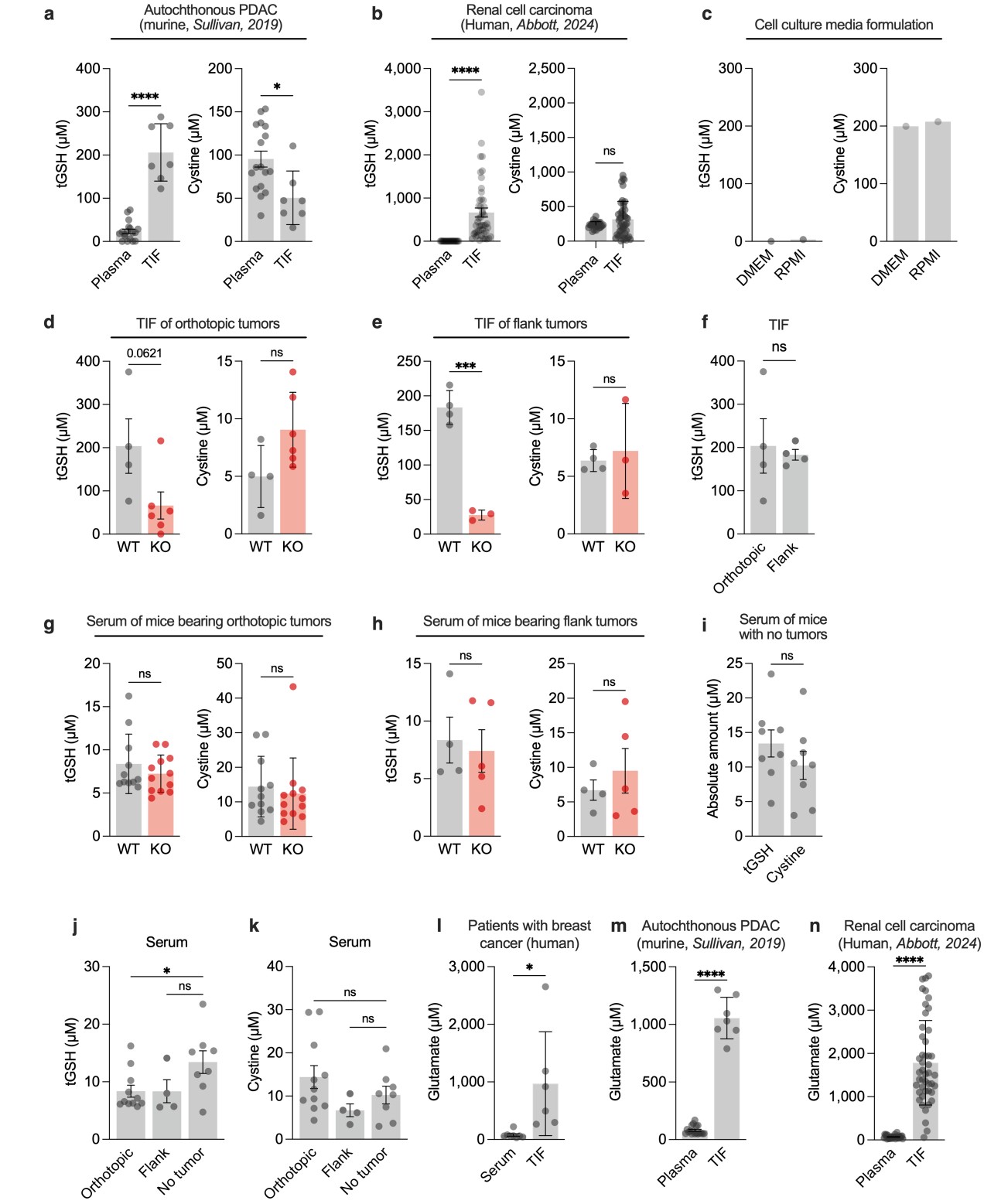

**Extended Data Fig. 3 |** See next page for caption.

**Extended Data Fig. 3 | Total GSH (tGSH) is enriched in the TIF compared to serum across multiple cancer models. a-c**, Concentration of tGSH and cystine in plasma (n = 17) and tumor interstitial fluid (n = 7) from the LSL-KrasG12D/+ Trp53f/f Pdx-1-Cre (KP[-/-]C) mice[28] bearing pancreatic ductal adenocarcinoma (PDAC) (GSH: ****p < 0.0001, Cys: *p = 0.0114) **(a)**, in the plasma (n = 27) and tumor interstitial fluid (n = 46) from renal cell carcinoma (RCC) human patients[29] (GSH: ****p < 0.0001, Cys: p = 0.1078) **(b)**, in cell culture media formulations **(c)**. **d-i**, Autochthonous tumors from MMTV-PyMT Gclc[f/f] Rosa26-CreERT2 mice were excised and orthotopically transplanted into the mammary gland or subcutaneously transplanted into the flank of wild-type C57BL/6 mice, which were treated with vehicle (corn oil; WT) or 50 mg/kg tamoxifen for 5 days (KO). tGSH and cystine levels in TIF from orthotopic (WT, n = 4; KO, n = 6; GSH: p = 0.0621, Cys: p = 0.0735) **(d)** or flank (WT, n = 4; KO, n = 3; GSH: p = 0.0001, Cys: p = 0.7050) **(e)** tumors. tGSH in TIF from orthotopic and flank Gclc WT MMTV-PyMT tumors (n = 4; p = 0.7607) **(f)**. tGSH and cystine levels in serum of mice with orthotopic (WT, n = 11; KO, n = 12; GSH: p = 0.3512, Cys: p = 0.6212) **(g)** or flank (WT, n = 4; KO, n = 5; GSH: p = 0.7398, Cys: p = 0.4937) **(h)** tumors. Serum tGSH and cystine in serum of mice without Gclc WT MMTV-PyMT tumors (n = 8; p = 0.2805) **(i)**. **j-k**, Serum tGSH (no tumor vs. orthotopic: *p = 0.0412; no tumor vs. flank: p = 0.1310) **(j)** and cystine (no tumor vs. orthotopic: p = 0.3725; no tumor vs. flank: p = 0.6482) **(k)** levels in mice bearing orthotopic (n = 11), flank (n = 4), or no Gclc WT tumors (n = 8). **l-n**, Glutamate levels in plasma and TIF from Fig. 1k (*p = 0.0153) **(l)**, from **a** (p < 0.0001) **(m)**, and from **b** (p < 0.0001) **(n)**. Total GSH (tGSH) was calculated as the sum of reduced glutathione (GSH) and twice the amount of oxidized glutathione (GSSG). Statistical significance was assessed by an unpaired two-tailed t-test in **a-i**, **l-n**, and by Ordinary One-Way ANOVA followed by Dunnett's multiple comparisons test in **j**, **k**. Data represented as mean ± s.e.m., *p-value < 0.05; **p-value < 0.01; ***p-value < 0.001; ****p-value < 0.0001; ns, not significant.

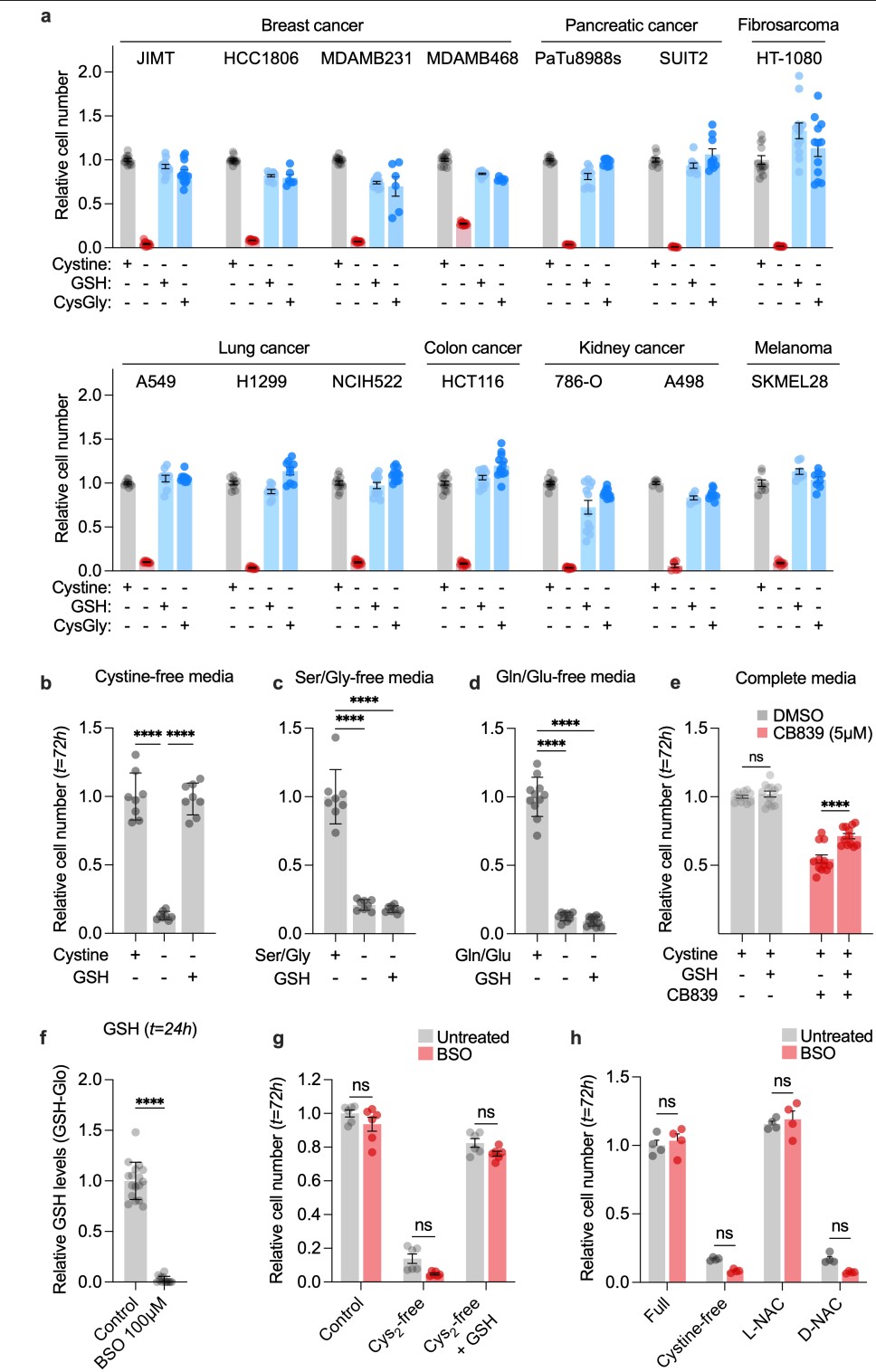

**Extended Data Fig. 4** | See next page for caption.

**Extended Data Fig. 4 | Supplementation with GSH or cysteinylglycine rescues cancer cells in cystine-free conditions. a**, Cancer cell lines were grown in control (208 µM cystine; $Cys_2$), cystine-free ($Cys_2$-free), cystine-free/GSH-supplemented (750 µM) or cystine-free/CysGly-supplemented (750 µM) medium, and relative cell numbers were determined. For JIMT-1, HT-1080, NCIH522, HCT-116, 786-O, and SK-MEL-28, n of technical replicates are: Control = 12; Cystine-free = 12; Cystine-free/GSH-supplemented = 12; Cystine-free/CysGly-supplemented = 12, from 4 independent experiments. For HCC-1806, MDAMB231, and MDAMB468, n of technical replicates are: Control = 12; Cystine-free = 12; Cystine-free/GSH-supplemented = 12; Cystine-free/CysGly-supplemented = 9, from 4 independent experiments. For PaTu8988s, SUIT2, A549, and H1299, n of technical replicates are: Control = 9; Cystine-free = 9; Cystine-free/GSH-supplemented = 9; Cystine-free/CysGly-supplemented = 9, from 3 independent experiments). For A498, n of technical replicates are: Control = 6; Cystine-free = 6; Cystine-free/GSH-supplemented = 6; Cystine-free/CysGly-supplemented = 6, from 2 independent experiments). **b-d**, HCC-1806 cells were grown in control media or cystine-free (n = 8 technical replicates from 2 independent experiments; ****p < 0.0001) **(b)**, serine- and glycine-free (n = 8 technical replicates from 2 independent experiments; ****p < 0.0001) **(c)**, glutamine- and glutamate-free (n = 11 technical replicates from 3 independent experiments; ****p < 0.0001) **(d)** media alone or supplemented with GSH (750 µM) for 72 h, and relative cell numbers were determined. **e**, HCC-1806 cells were grown in control media alone or supplemented with GSH (750 µM) and treated with DMSO or CB-839 (5 µM) for 72 h, and relative cell numbers were determined (n = 12 technical replicates from 3 independent experiments; ****p < 0.0001). **f**, HCC-1806 cells were treated with BSO (100 µM) for 24 h, and relative GSH levels were determined. Statistical significance was assessed by an unpaired two-tailed t-test (Control, n = 17; BSO-treated, n = 13 technical replicates from 3 independent experiments, ****p < 0.0001). **g**, HCC-1806 cells were grown in control (208 µM cystine; $Cys_2$), cystine-free ($Cys_2$-free), or cystine-free/GSH-supplemented (750 µM) medium, untreated or treated with BSO (100 µM) for 72 h, and relative cell numbers were determined. Statistical significance was assessed by two-way ANOVA followed by Šídák's multiple comparisons (n = 6 technical replicates). **h**, HCC-1806 cells were grown in control (208 µM cystine; $Cys_2$), cystine-free ($Cys_2$-free), cystine-free/L-NAC-supplemented (400 µM), cystine-free/D-NAC-supplemented (400 µM) medium, untreated or treated with BSO (100 µM) for 72 h, and relative cell numbers were determined (n = 4 technical replicates). Statistical significance was assessed by ordinary One-Way ANOVA followed by Dunnett's multiple comparisons test in **b**, **c**, **d**; by two-way ANOVA followed by Šídák's multiple comparisons in **e**, **g**, **h**; and by unpaired two-tailed t-test in **f**. Data represented as mean ± s.e.m., ****p-value < 0.0001; ns, not significant.

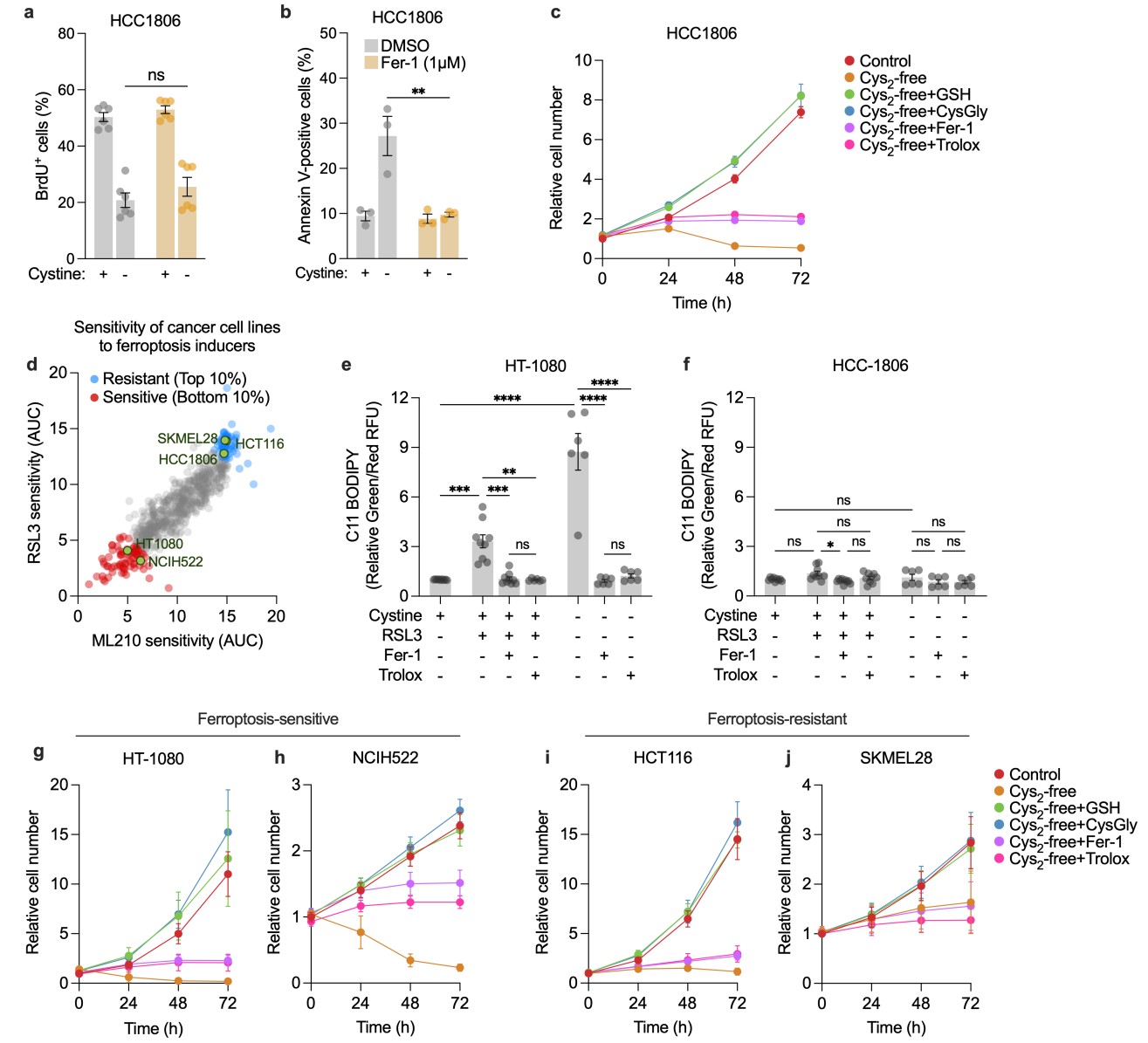

**Extended Data Fig. 5 | Rescue of cancer cells by supplementation of GSH or cysteinylglycine in cystine-free conditions is distinct from rescue provided by ferroptosis inhibitors. a-b**, HCC1806 cells were grown in control (208 μM cystine; Cys₂) and cystine-free (Cys₂-free) medium with vehicle (DMSO) or ferrostatin-1 (Fer-1, 1 μM). Percentages of proliferative (BrdU⁺) (n = 6 technical replicates representative of 2 independent experiments) (**a**) and apoptotic (Annexin V⁺) (n = 3 technical replicates representative of 3 independent experiments, **p = 0.0042) (**b**) cells were determined. **c**, HCC1806 cells were grown in control (208 μM cystine; Cys₂), cystine-free (Cys₂-free), or cystine-free supplemented with GSH (750 μM), CysGly (750 μM), Fer-1 (10 μM), or Trolox (100 μM), and relative cell numbers were determined (n = 8 technical replicates from 2 independent experiments). **d**, Sensitivity (area under the curve, AUC) of n = 760 cancer cell lines to GPX4 inhibitors RSL-3 and ML-210. Ferroptosis-resistant (top 10%, blue) and ferroptosis-sensitive (bottom 10%, red) indicated. Sensitivity values determined from DepMap and CCLE databases. **e-f**, Ferroptosis-sensitive HT-1080 cell line (**e**) and ferroptosis-resistant

HCC-1806 cell line (**f**) were grown in control (208 μM cystine; Cys₂) and cystine-free (Cys₂-free) medium with vehicle (DMSO) or RSL3 (250 nM), alone or with Fer-1 (10 μM) or Trolox (100 μM). After 24 h, cells were stained with C11-BODIPY, and relative green and red fluorescence was determined (n = 6 technical replicates from 2 independent experiments; in **e**, **p = 0.0027, ***p = 0.0006 (Control *vs.* RSL3), ***p = 0.0009 (RSL3 *vs.* RSL3 + Trolox), and ****p < 0.0001; in **f**, *p = 0.0207). **g-j**, Ferroptosis-sensitive cell lines HT-1080 (**g**) and NCIH522 (**h**), and ferroptosis-resistant cell lines HCT116 (**i**) and SKMEL28 (**j**) were grown in control (208 μM cystine; Cys₂), cystine-free (Cys₂-free), or cystine-free supplemented with GSH (750 μM), CysGly (750 μM), Fer-1 (10 μM) or Trolox (100 μM), and relative cell numbers were determined (n = 12 technical replicates from 3 independent experiments). Statistical significance was analyzed by ordinary two-way ANOVA followed by Šídák's multiple comparisons in **a**, **b**; and by ordinary one-way ANOVA followed by Tukey's multiple comparisons in **e**, **f**. Data represented as mean ± s.e.m., ns, not significant.

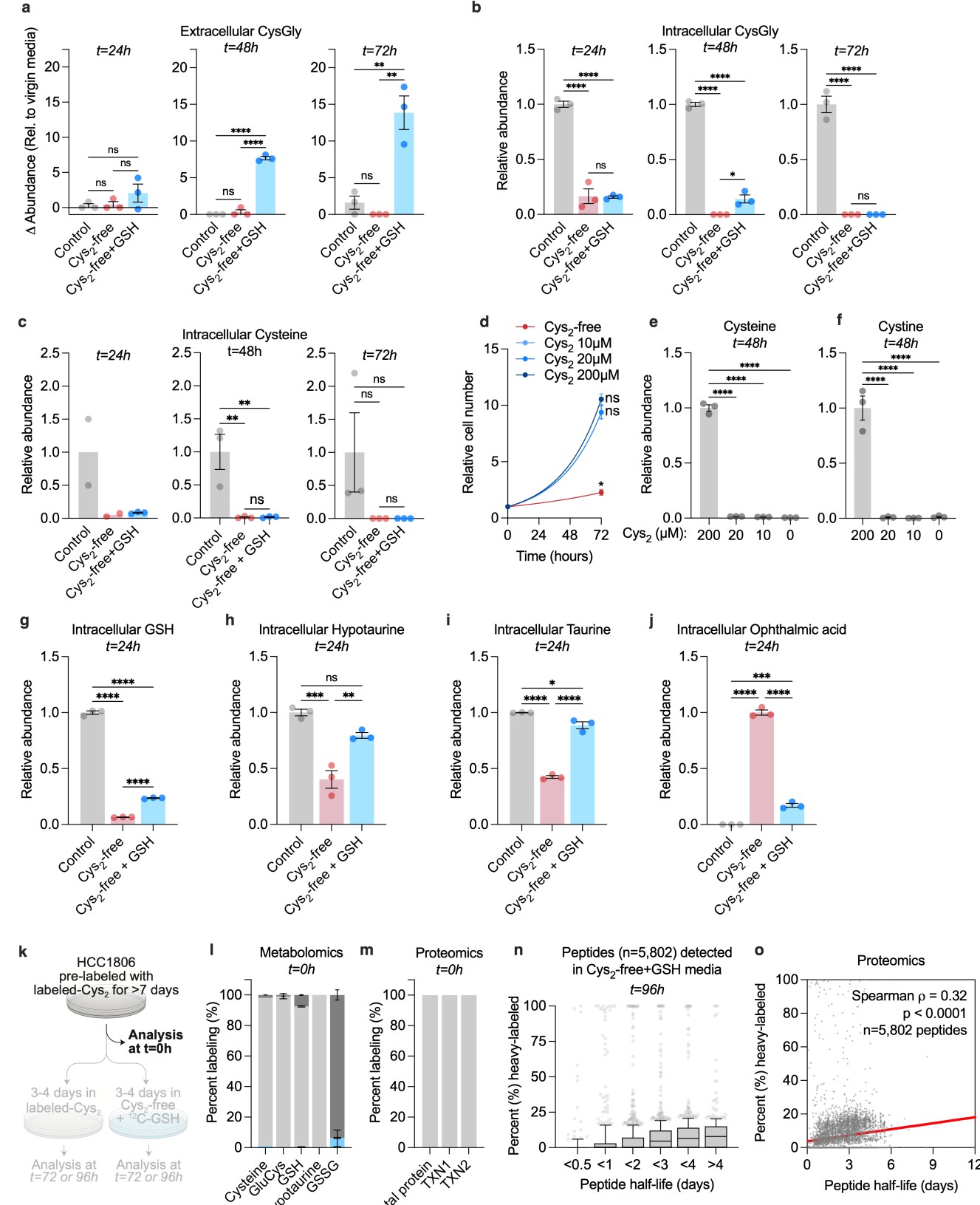

**Extended Data Fig. 6** | See next page for caption.

**Extended Data Fig. 6 | Supplementation of GSH in cystine-free conditions rescues downstream products of cysteine metabolism. a-c**, Difference in abundance in the extracellular cysteinylglycine in medium **(a)** and relative abundance of intracellular cysteinylglycine **(b)** and intracellular cysteine **(c)** for HCC-1806 cells grown in the indicated conditions and time points (n = 3 technical replicates representative from 2 independent experiments, except for **c** at 24 h, in which n = 2 technical replicates from 2 independent experiments); in **a**, ****p < 0.0001, **p = 0.0021 (Control *vs.* Cystine-free+GSH), **p = 0.0011 (Cystine-free *vs.* Cystine-free+GSH); in **b**, ****p < 0.0001; and in **c**, ****p < 0.0001, **p = 0.0090 (Control *vs.* Cystine-free), **p = 0.0092 (Control *vs.* Cystine-free+GSH). **d-f**, Relative cell number (*p = 0.0276) **(d)** and relative abundance of intracellular NEM-cysteine (****p < 0.0001) **(e)** and intracellular cystine (****p < 0.0001) **(f)** in HCC-1806 cells grown in the indicated cystine concentrations and time points (n = 3 technical replicates from one experiment. **g-j**, Relative abundance of intracellular NEM-GSH (****p < 0.0001) **(g)**, hypotaurine (**p = 0.0037, ***p < 0.0004) **(h)**, taurine (*p = 0.0146, ****p < 0.0001 **(i)**, and ophthalmic acid (***p = 0.0009, ****p < 0.0001) **(j)**. Δ Abundance represents the metabolite level difference between conditioned and virgin media (n = 3 technical replicates representative of 2 independent experiments). **k**, Schematic of cystine stable-isotope labeling approach for metabolomics and proteomics. **l-m**, Percent labeling from metabolomics **(l)** and proteomics **(m)** at 0 h, demonstrating complete cystine labeling prior to introducing unlabeled GSH (n = 3 technical replicates from one experiment). **n-o**, Relationship between peptide half-life[36] and percent residual $^{13}$C heavy-labeling in cystine-free/$^{12}$C-GSH-supplemented medium at 96 h. Correlation was calculated by nonparametric two-tailed Spearman test (****p < 0.0001; ρ = 0.3225). Statistical significance assessed by one-way ANOVA with Tukey's multiple comparisons test in **a-c**, **e-j**. Statistical significance between each condition versus the control (200 μM Cystine) was analyzed by Kruskal-Wallis test with Dunn's multiple comparisons in **d**. Data represented as mean ± s.e.m., *p-value < 0.05; **p-value < 0.01; ***p-value < 0.001; ****p-value < 0.0001; ns, not significant.

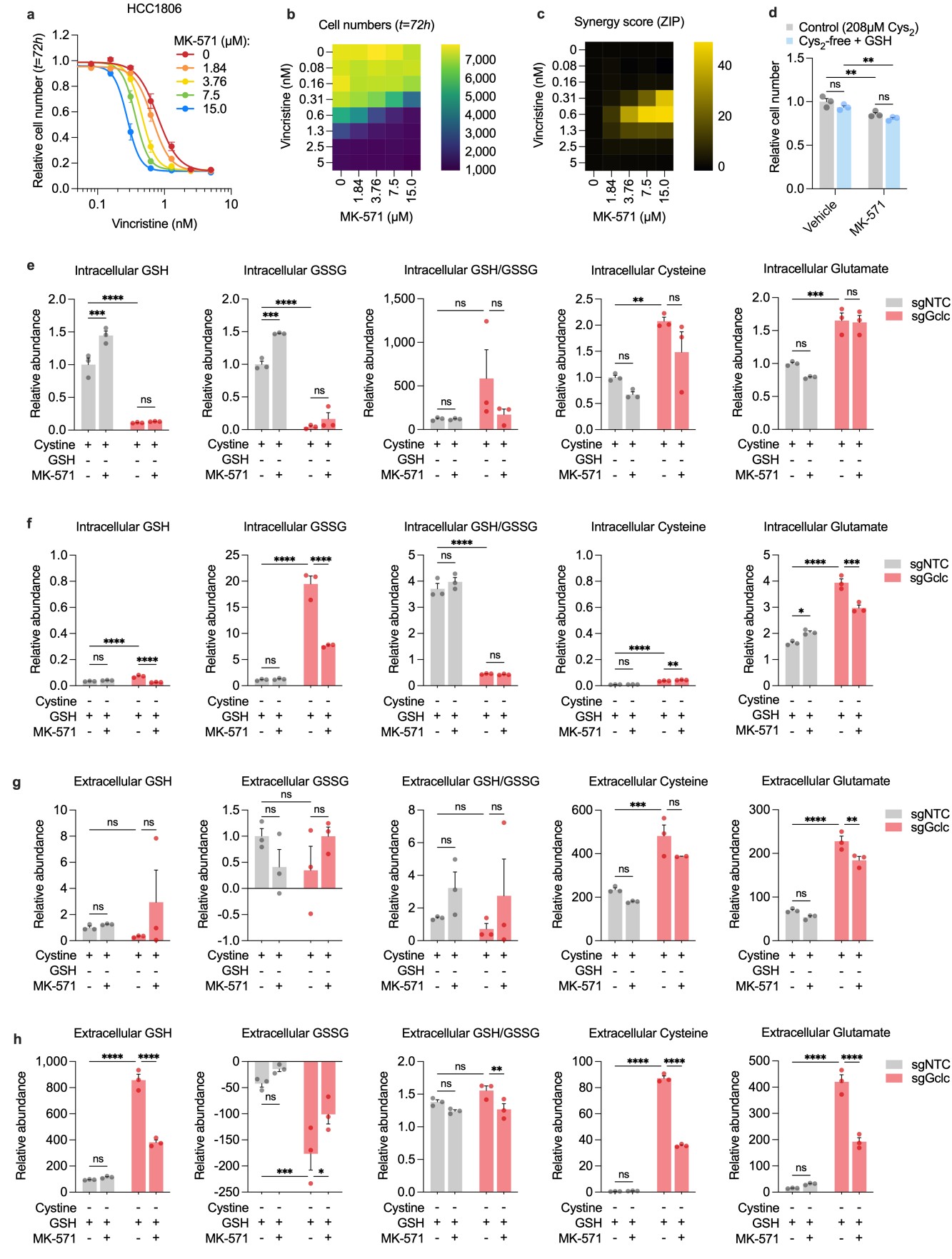

**Extended Data Fig. 7** | See next page for caption.

**Extended Data Fig. 7 | Blocking the export of intracellular GSH does not impair the ability of GSH to rescue in cystine-free conditions. a**, Dose-response curves showing relative cell number after 72 h of treatment of HCC-1806 cells with vincristine (0-5 nM) in the presence of the MRP1 inhibitor MK-571 (0-15 μM) (n = 8 technical replicates from one experiment). **b**, Heatmap of absolute cell numbers of data shown in **(a)**. **c**, Drug synergy analysis of data shown in **(a)** using the ZIP model. **d**, Relative cell numbers after 72 h of treatment of HCC-1806 cells without or with MK-571 (15 μM) in control (208 μM cystine; $Cys_2$) (**p = 0.0026) or cystine-free/GSH-supplemented (750 μM) medium (**p = 0.0045). n = 3 technical replicates from one experiment. **e-h**, HCC-1806 cells were grown without or with MK-571 (15 μM) in control (208 μM cystine; $Cys_2$) or cystine-free/GSH-supplemented (750 μM) medium for 48 h, and the relative abundance of indicated intracellular **(e-g)** and extracellular **(f-g)** metabolites was determined. Δ Abundance represents the metabolite level difference between conditioned and virgin media (n = 3 technical replicates from one experiment). In **e**, Intracellular GSH: ***p = 0.0009, ****p < 0.0001; intracellular GSSG: ***p = 0.0003, ****p < 0.0001; intracellular cysteine: **p = 0.0053, *p = 0.0212; intracellular glutamate: ***p = 0.0004, ****p < 0.0001. In **f**, Intracellular GSH: **p = 0.0096, ****p < 0.0001; intracellular GSSG: ***p = 0.0004, ****p < 0.0001; intracellular GSH/GSSG: ****p < 0.0001; intracellular cysteine: **p = 0.0011, ****p < 0.0001; intracellular glutamate: ***p = 0.0002, *p = 0.0196. In **g**, intracellular cysteine: **p = 0.0020, ***p = 0.0004, intracellular glutamate: ***p = 0.0039, ****p < 0.0001. In **h**, Intracellular GSH: ****p < 0.0001; intracellular GSSG: *p = 0.0105 (sgNTC *vs.* sgGclc), *p = 0203, ***p = 0.0009; intracellular GSH/GSSG: **p = 0.0087; intracellular cysteine: ****p < 0.0001, intracellular glutamate: ****p < 0.0001. Statistical significance was assessed by two-way ANOVA followed by uncorrected Fisher's Least Significant Difference (LSD) test in **d-h**. Data represented as mean ± s.e.m., ns, not significant.

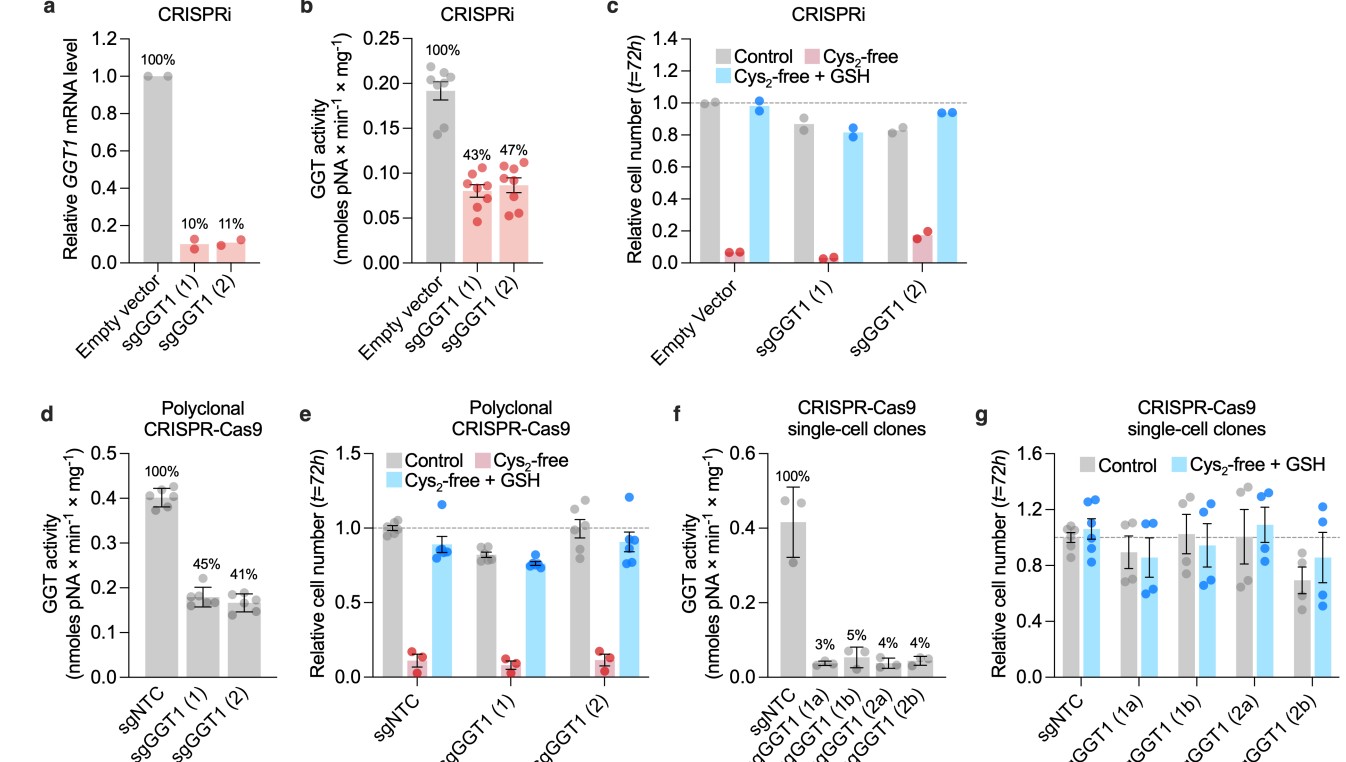

**Extended Data Fig. 8 | Reduction or deletion of GGT1 expression does not impair GSH-mediated rescue of cells in cystine-free conditions. a-b**, HCC-1806 cells were transduced with CRISPRi lentiviral vector containing sgRNA guides targeting the GGT1 promoter. GGT1 mRNA **(a)** (n = 2 technical replicates representative of one experiment) and GGT activity **(b)** (n = 8 technical replicates from 3 independent experiments) in control (208 μM cystine) medium were determined. **c**, GGT1 knockdown cells from **(a-b)** were grown in control (208 μM cystine; Cys$_2$), cystine-free (Cys$_2$-free), or cystine-free/GSH-supplemented (750 μM) medium for the indicated timepoints, and cell numbers were determined (n = 2 technical replicates representative of one experiment). **d-e**, HCC-1806 cells were transduced with CRISPR-Cas9 lentiviral vector containing sgRNA guides targeting GGT1 and maintained as a polyclonal

population. GGT activity **(d)** (n = 6 technical replicates from 3 independent experiments) and cell numbers **(e)** (n = 3 technical replicates for Cys$_2$-free or n = 6 technical replicates for Control or Cys$_2$-free + GSH from 3 independent experiments) for cells grown in control (208 μM cystine; Cys$_2$), cystine-free (Cys$_2$-free), or cystine-free/GSH-supplemented (750 μM) medium for the indicated time points were determined. **f-g**, Single-cell clones were established from the polyclonal population, and GGT activity **(f)** (n = 3 technical replicates from 2 independent experiments) and cell numbers **(g)** (n = 6 technical replicates for Control or n = 4 technical replicates for Cys$_2$-free + GSH from 2 independent experiments) for cells grown in control (208 μM cystine; Cys$_2$), cystine-free (Cys$_2$-free), or cystine-free/GSH-supplemented (750 μM) medium for the indicated timepoints were determined. Data represented as mean ± s.e.m.

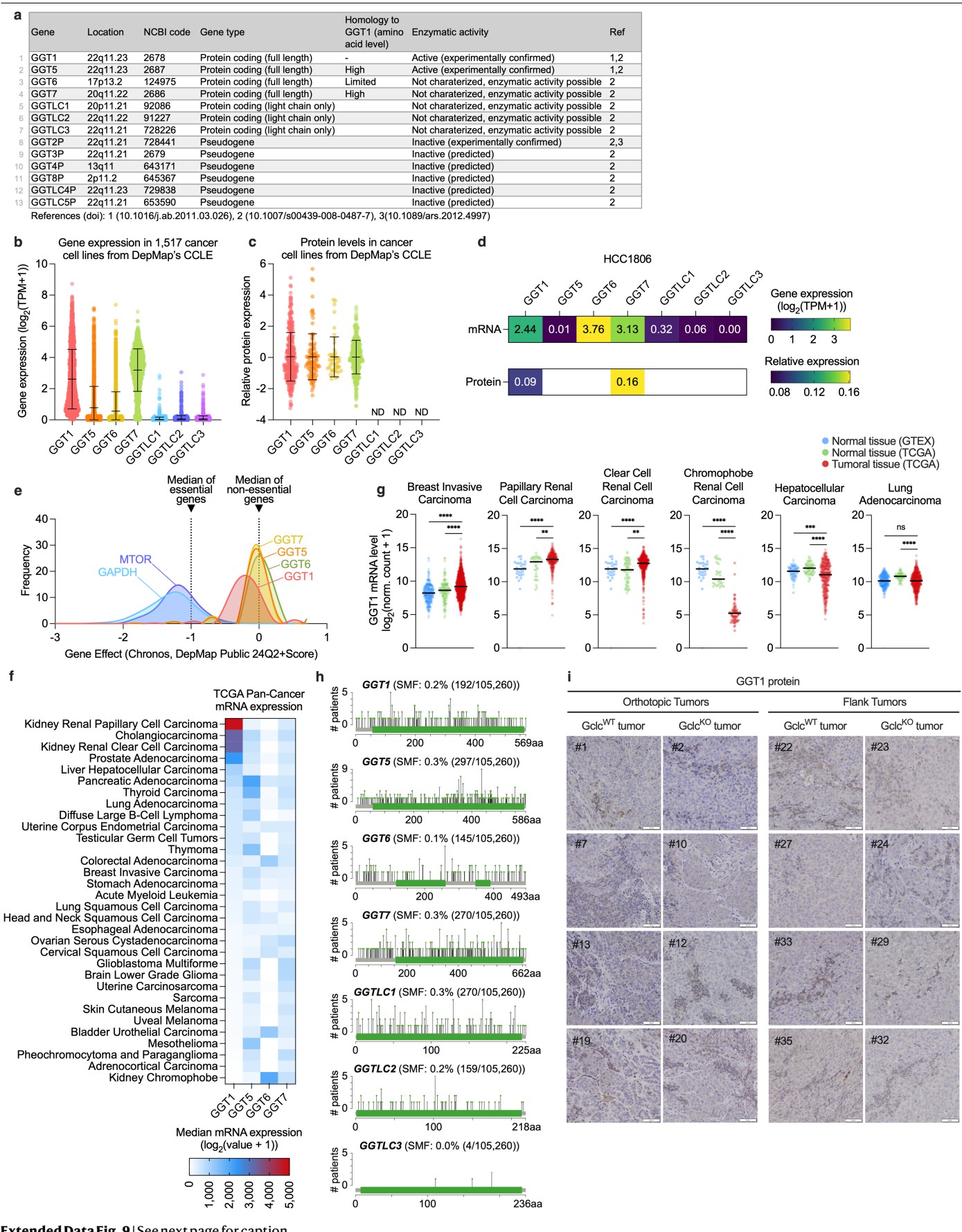

**Extended Data Fig. 9 |** See next page for caption.

**Extended Data Fig. 9 | Multiple GGT isoforms are expressed in cancer cell lines, but none score as essential in genome-wide genetic screens. a**, GGT gene family table showing all 13 GGT-related genes in the human genome. GGT1, 5, 6, and 7 are predicted to have enzymatic activity, but only two (GGT1,5) have been functionally characterized. **b**, mRNA expression (Expression Public 24Q2) of cancer cell lines (n = 1,517) from DepMap's Cancer Cell Line Encyclopedia (CCLE). **c**, Protein levels from cancer cell lines from DepMap's CCLE (Proteomics GGT1: P19440 (n = 339), GGT5: P36269 (n = 108), GGT6: Q6P531 (n = 45), GGT7: Q9UJ14 (n = 375)). **d**, mRNA and protein levels of GGT isoforms in the cell line HCC-1806, using data from **(b)** and **(c)**. **e**, Histogram (splined curves) of gene dependency of cancer cell lines from Dependency Map (DepMap) (Public 24Q2+Score Chronos). A lower score suggests a gene is more likely to be essential in a cell line. A score of 0 indicates a non-essential gene, while -1 reflects the median for essential genes. *MTOR* and *GAPDH* are shown as examples of common essential genes. **f**, mRNA expression of GGT enzymes was analyzed in human tumors from 32 different TCGA projects. **g**, GGT1 mRNA expression levels in human tumoral and non-tumoral tissues from the Cancer Genome Atlas Program (TCGA) and the Genotype-Tissue Expression (GTEx) databases. Black lines represent median values. Statistical significance was analyzed by two-tailed unpaired t-test. ns, not significant; ****p-value < 0.0001; **p = 0.0029 (papillary renal cell carcinoma); **p = 0.0038 (clear cell renal cell carcinoma); ***p = 0.0005 (hepatocellular carcinoma). **h**, The mutational landscape and somatic mutation frequence (SMF) of each GGT gene was analyzed in 105,260 cancer patient samples across 226 non-redundant studies using the cBioPortal platform revealing a low mutation frequency (mean = 0.15%). **i**, Autochthonous tumors from MMTV-PyMT Gclcf/f Rosa26-CreERT2 mice were excised and orthotopically transplanted into the mammary gland or subcutaneously transplanted into the flank of wild-type C57BL/6 mice, which were treated with vehicle (corn oil; WT) or 50 mg/kg tamoxifen for 5 days (KO). Immunohistochemistry of GGT1 protein expression in *Gclc* WT and KO orthotopic and flank tumors (n = 4 animals representative of 2 independent experiments). Numbers displayed on the top left of each panel represent the animal identification number. Scale bar = 50 μm.

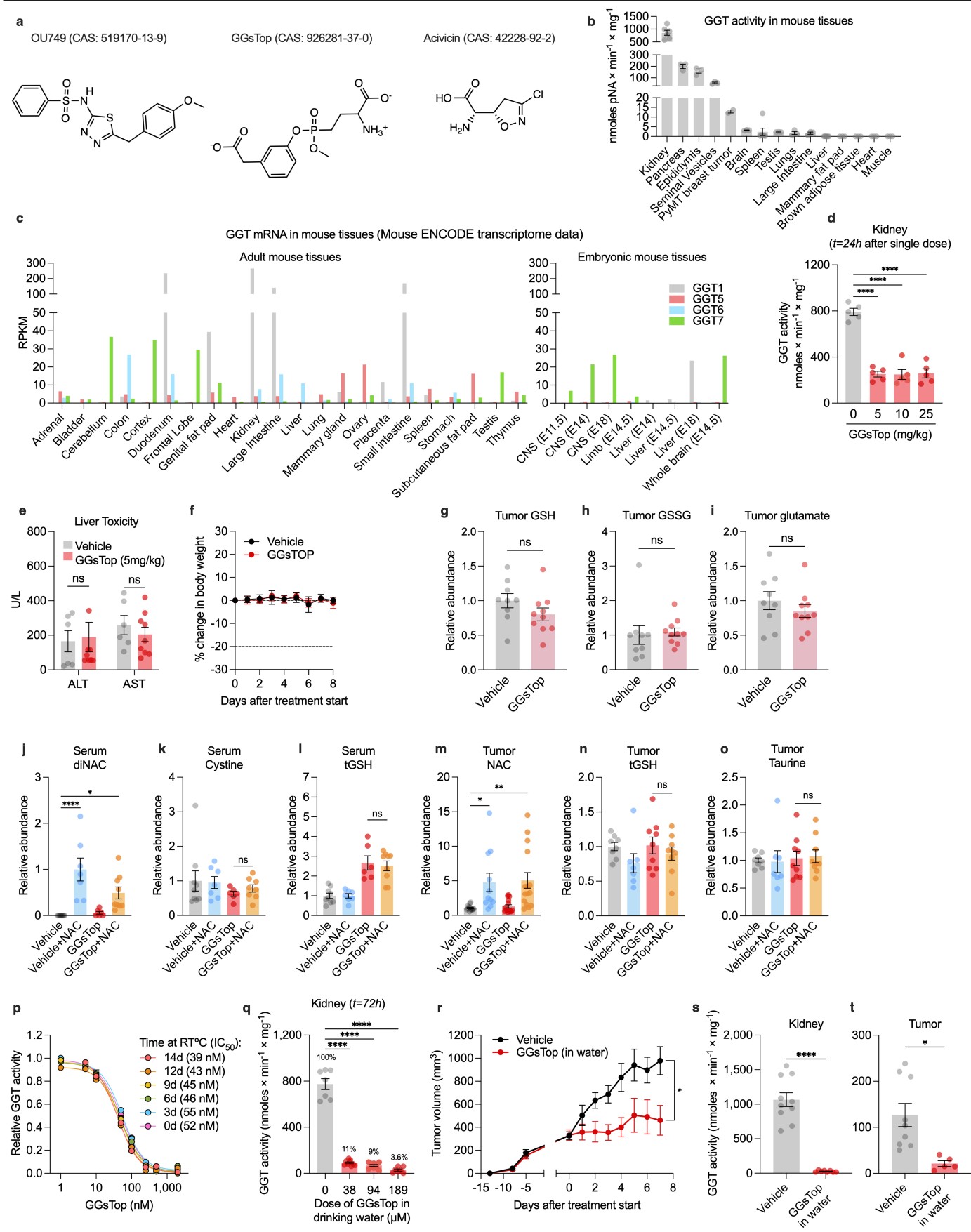

**Extended Data Fig. 10** | See next page for caption.

**Extended Data Fig. 10 | GGsTop efficiently inhibits GGT activity in animals without causing toxicity. a**, Structure of putative inhibitors of GGT. **b**, GGT activity in untreated C57BL/6 mouse tissues (kidney, spleen, and liver; n = 6 animals analyzed in 2 independent experiments. pancreas, epididymis, seminal vesicles, brain, testis, lungs, large intestine, brown adipose tissue, heart, and muscle; n = 3 animals analyzed in one experiment. Mammary fat pad; n = 4 animals analyzed in 2 independent experiments. PyMT breast tumors; n = 2 animals analyzed in one experiment). **c**, mRNA expression of GGT isoforms in 22 adult and 8 embryonic mouse tissues from publicly available datasets[73]. **d**, GGT activity in the kidney of mice 24 h after a single i.p. injection of GGsTop at the indicated concentrations (n = 5 animals from one experiment; ****p < 0.0001). **e-f**, ALT and AST liver damage serum markers (n = 6 for Control, n = 9 for GGsTop) **(e)** and percent change in body weight (n = 10 per group) **(f)** in mice treated bi-daily with vehicle (saline) or 5 mg/kg i.p. GGsTop for 8 days. **g-i**, Tumor metabolites from mice with orthotopically implanted HCC-1806 cells treated with bi-daily i.p. injections of vehicle or 5 mg/kg GGsTop (n = 9 for Control, n = 10 for GGsTop). **j-o**, Serum and tumor metabolites from mice with orthotopically implanted HCC-1806 cells treated with the vehicle, n-acetylcysteine (30 mM in the drinking water), GGsTop (189 μM in the drinking water), or NAC and GGsTop (Tumors; n = 8 for vehicle, n = 7 for vehicle+NAC, n = 9 for GGsTop, n = 9 for GGsTop+NAC. Serum; n = 9 for vehicle, n = 7 for vehicle+NAC, n = 6 for GGsTop, n = 9 for GGsTop+NAC). In **j**, *p = 0.020, ****p < 0.0001; in **m**, *p = 0.0208, **p = 0.0080. **p**, GGsTop aliquots (25 mg/mL in $H_2O$) were kept at room temperature for the indicated number of days, after which their inhibitory potency was assessed using an in vitro activity assay with mouse kidney lysates (n = 3 technical replicates). **q**, GGT activity in the kidney of mice 72 h after delivery of GGsTop in the drinking water at the indicated concentrations (n = 7 for 0 and 189 μM, n = 9 for 38 μM, n = 10 for 94 μM, ****p < 0.0001). **r-t**, Tumor volume over time (*p = 0.0254) **(r)**, and GGT activity in the kidney (****p < 0.0001) **(s)** and tumor (*p = 0.0101) **(t)** from mice with orthotopically implanted HCC-1806 cells treated with the vehicle (n = 10) or GGsTop (189 μM in the drinking water, n = 8). Statistical significance was assessed by one-way ANOVA followed with Dunnet's multiple comparisons test in **d, q**, by ordinary two-way ANOVA followed by Šídák's multiple comparisons test in **e, r**, and by ordinary two-way ANOVA followed Tukey's multiple comparisons test in **j-o**. Statistical significance was assessed by two-tailed unpaired t-test in **g-l, s, t**. Data represented as mean ± s.e.m., *p-value < 0.05; **p-value < 0.01; ***p-value < 0.001; ****p-value < 0.0001; ns, not significant.

# Reporting Summary

## Statistics

For all statistical analyses, confirm that the following items are present in the figure legend, table legend, main text, or Methods section.

| n/a | Confirmed | |
|---|---|---|
| ☐ | ☒ | The exact sample size (*n*) for each experimental group/condition, given as a discrete number and unit of measurement |
| ☐ | ☒ | A statement on whether measurements were taken from distinct samples or whether the same sample was measured repeatedly |
| ☐ | ☒ | The statistical test(s) used AND whether they are one- or two-sided<br>*Only common tests should be described solely by name; describe more complex techniques in the Methods section.* |
| ☒ | ☐ | A description of all covariates tested |
| ☒ | ☐ | A description of any assumptions or corrections, such as tests of normality and adjustment for multiple comparisons |
| ☐ | ☒ | A full description of the statistical parameters including central tendency (e.g. means) or other basic estimates (e.g. regression coefficient) AND variation (e.g. standard deviation) or associated estimates of uncertainty (e.g. confidence intervals) |
| ☐ | ☒ | For null hypothesis testing, the test statistic (e.g. *F*, *t*, *r*) with confidence intervals, effect sizes, degrees of freedom and *P* value noted<br>*Give P values as exact values whenever suitable.* |
| ☒ | ☐ | For Bayesian analysis, information on the choice of priors and Markov chain Monte Carlo settings |
| ☒ | ☐ | For hierarchical and complex designs, identification of the appropriate level for tests and full reporting of outcomes |
| ☒ | ☐ | Estimates of effect sizes (e.g. Cohen's *d*, Pearson's *r*), indicating how they were calculated |

*Our web collection on statistics for biologists contains articles on many of the points above.*

## Software and code

Policy information about availability of computer code

| Data collection | • Microscopy images of cell lines in multi-well plates were acquired using either the CellInsight CX5 HCS instrument (Thermo) with HCS Navigator Software (Thermo, version 6.6.2, build 8533) or the SparkCyto instrument (Tecan) with SparkControl software (Tecan, version 4.0). |
|---|---|
| Data analysis | • Statistical analysis was performed using GraphPad Prism (version 10.6.1).<br>• Illustrations (Fig 1a, 2a,h,j, 3e, 4a,c) were created using Adobe Illustrator (version 30.1).<br>• MAPS data was processed using R (version 4.1.1).<br>• Flow cytometry analysis was performed using FCS Express 7 Research (DeNovo Software, version 7.24.0030).<br>• LC-MS metabolite peaks identification was conducted using Compound Discover (v3.3, Thermo Scientific), Skyline software (Version 25.1), and El-Maven software (Version 0.12.0).<br>• Proteomics data was processed with DIA-NN software (version 1.9.2, https://github.com/vdemichev/DIANN).<br>• Cell counts of images acquired by CellInsight CX5 HCS platform (Thermo) were determined using HCS Navigator Software (Thermo, version 6.6.2, build 8533).<br>• Cell counts of images acquired by SparkCyto (Tecan) were determined using SparkControl (Tecan, version 4.0). |

For manuscripts utilizing custom algorithms or software that are central to the research but not yet described in published literature, software must be made available to editors and reviewers. We strongly encourage code deposition in a community repository (e.g. GitHub). See the Nature Portfolio guidelines for submitting code & software for further information.

# Data

Policy information about <u>availability of data</u>
All manuscripts must include a <u>data availability statement</u>. This statement should provide the following information, where applicable:
- Accession codes, unique identifiers, or web links for publicly available datasets
- A description of any restrictions on data availability
- For clinical datasets or third party data, please ensure that the statement adheres to our <u>policy</u>

Source data for all figures are provided with this paper. Uncropped western blot images are provided in Supplementary Figure 1.

Publicly available datasets used in this study:
• GTEX: https://gtexportal.org/home/ (Extended Data Fig 1a, 9g)
• TCGA, accessed through cBioPortal platform: https://www.cbioportal.org/datasets (Extended Data Fig 9g,f,h)
• Cancer Cell Line Encyclopedia (CCLE): https://depmap.org/portal/ccle/ (Extended Data Fig 9b,c,d)
• DepMap: https://depmap.org/portal/ (Extended Data Fig 9e)
• ENCODE database, accessed through NCBI-Gene portal: https://www.ncbi.nlm.nih.gov/gene/ (BioProject: PRJNA66167) (Figure 10c)

# Research involving human participants, their data, or biological material

Policy information about studies with <u>human participants or human data</u>. See also policy information about <u>sex, gender (identity/presentation), and sexual orientation</u> and <u>race, ethnicity and racism</u>.

| | |
|---|---|
| Reporting on sex and gender | The biological sex of all 16 deidentified biospecimens was categorized as female. |
| Reporting on race, ethnicity, or other socially relevant groupings | The cohort included 81% white participants, 6% black, and 13% with race not reported. |
| Population characteristics | A total of 16 subjects were enrolled, all of whom were female with a median age of 64 years (range 25 to 88). Neoadjuvant treatment was administered in 4 subjects (25%). The cohort included 81% White, 6% Black, and 13% with race not reported. Most tumors were invasive ductal carcinoma (94%) with 6% classified as other histology. Estrogen receptor positivity was observed in 81% of cases and progesterone receptor positivity in 62%. HER2 status was distributed as 0 in 56%, 1+ in 32%, 2+ in 6%, and 3+ in 6%. Tumor staging showed 44% T1 and 56% T2 with no T3 or T4 disease. Nodal staging included 38% N0, 38% N1, 6% N2, and 18% NX. All subjects were M0 at diagnosis with no metastatic disease. Tumor grade was 1 in 12%, 2 in 50%, and 3 in 38%. |
| Recruitment | Human biospecimens and deidentified clinical data were provided by the Wilmot Cancer Institute Biobank Shared Resource (BSR) at the University of Rochester. |
| Ethics oversight | Human biospecimens and deidentified clinical data were provided by the Wilmot Cancer Institute Biobank Shared Resource (BSR) at the University of Rochester. All samples were collected under Institutional Review Board-approved protocols (STUDY61977 and STUDY7108), and all subjects provided written informed consent. |

Note that full information on the approval of the study protocol must also be provided in the manuscript.

# Field-specific reporting

Please select the one below that is the best fit for your research. If you are not sure, read the appropriate sections before making your selection.

☒ Life sciences ☐ Behavioural & social sciences ☐ Ecological, evolutionary & environmental sciences

For a reference copy of the document with all sections, see nature.com/documents/nr-reporting-summary-flat.pdf

# Life sciences study design

All studies must disclose on these points even when the disclosure is negative.

| | |
|---|---|
| Sample size | For cell culture experiments, sample sizes were not chosen based on statistical methods. All cell culture experiments were repeated at least 2 independent times, each with n≥3 technical replicates (i.e., independent wells) per experimental condition, unless stated otherwise. This number was selected based on pilot experiments and past experience showing that ≥2 independent experiments consistently captures biological variability and is sufficient to detect reproducible and statistically significant differences in cell growth and metabolite abundance between experimental conditions.<br><br>For animal studies, sample sizes were not chosen based on statistical methods. The number of animals assigned per condition was selected to account for the variability of the examined phenotypes based on pilot experiment and past experience with the animal models (≥6 animals per experimental condition).<br><br>For human studies, sample sizes were not chosen based on statistical methods. All available human samples were collected within the study |

| | |
|---|---|
| time frame, yielding a total of 16 deidentified plasma and breast cancer tissue specimens for analysis. | |

Data exclusions | For in vitro and in vivo studies, the existance of outliers was tested using the Robust Regression and Outlier Removal (ROUT) method (Q=1%) in GraphPad Prism 10.6.1 software. For animal xenograft studies, injections that resulted in a tumor of less than 50 mm3 within the first 14 days were excluded (prior to animal group allocation).

Replication | All in vitro data reported in this study (including metabolomics) were repeated at least twice (independent experiments), each including a minimum of two technical replicates (i.e., independent wells). All in vitro experiments (including metabolomics) were similar between replicate independent experiments. For proteomics, additional repeats were not conducted due to the magnitude of the observed effect and logistics required for the experiment. All animal experiments were conducted ≥2 independent times, each with with at least 6 animals per condition. All in vivo experiments were similar between independent replicate experiments.

Randomization | Mice of desired strains were age-matched and assigned randomly to their treatment groups. For xenograft studies, animals were allocated into groups ensuring the mean, median and standard error of tumor size was similar across all groups. For in vitro cancer cell culture experiments, cells at similar passage number (passage <25) were seeded at equal densities across all conditions and plate form factors to minimize inter- and intra-experimental variability. Human biospecimens were not assigned to different experimental groups, so randomization was not required.

Blinding | Investigators were not blinded to group allocation during experiments due to technical limitations. All treatments and measurements were performed under the same conditions for all animals. Further, all samples were analyzed together and subjected to the same data processing.

# Reporting for specific materials, systems and methods

We require information from authors about some types of materials, experimental systems and methods used in many studies. Here, indicate whether each material, system or method listed is relevant to your study. If you are not sure if a list item applies to your research, read the appropriate section before selecting a response.

## Materials & experimental systems

| n/a | Involved in the study |
|---|---|
| ☐ | ☒ Antibodies |
| ☐ | ☒ Eukaryotic cell lines |
| ☒ | ☐ Palaeontology and archaeology |
| ☐ | ☒ Animals and other organisms |
| ☒ | ☐ Clinical data |
| ☒ | ☐ Dual use research of concern |
| ☒ | ☐ Plants |

## Methods

| n/a | Involved in the study |
|---|---|
| ☒ | ☐ ChIP-seq |
| ☒ | ☐ Flow cytometry |
| ☒ | ☐ MRI-based neuroimaging |

## Antibodies

Antibodies used | Primary antibodies used in Western Blotting:
• rabbit-anti-GGT1 (affinity-purified, #GGT129). This antibody was produced by Dr. Marie Hanigan (University of Oklahoma) and it is not commercially available. Reference: PMID 8813074
• rabbit-anti-GCLC (Sigma-Aldrich HPA036359)
• mouse-anti-beta actin (Sigma-Aldrich A1978, Clone AC-15)

Secondary antibodies used in Western Blotting:
• donkey-anti-rabbit IgG-HRP (Amersham/ECL, NA934)
• sheep-anti-mouse IgG-HRP (Amersham/ECL NA931)

Primary anitbodies used for immunohistochemistry (IHC):
• anti-GCSc (SantaCruz sc390811)
• anti-CD45 (Santa Cruz sc1178)
• anti-F4/80 (Cell Signaling Technology 70076)
• anti-DNA/RNA Damage (Abcam 62623)
• anti-Nrf2 (Abcam 31163)

Secondary anitbodies used for immunohistochemistry (IHC):
• Biotinylated goat anti-mouse IgG  (Vector Labs BA-9200)
• Biotinylated goat anti-rabbit IgG (Vector Labs BA-1000)

Primary antibodies used for immunofluorescence:
• anti-GCSc (SantaCruz sc390811)
• anti-CD45 (Proteintech 31243-1-AP)

Secondary antibodies used for immunofluorescence:
• Goat anti-Mouse-Alexa Fluor 594 (Invitrogen A11005)
• Goat-anti-rabbit-AlexaFluor 488 (Invitrogen A11034)

| | |
|---|---|
| | Antibody used for BrdU staining:<br>• anti-BrdU-FITC (Biolegend 364104, clone 3D4) |
| Validation | Primary antibodies used in Western Blotting:<br>• anti-GGT1: validation performed by Dr. Marie Hanigan (University of Oklahoma). Western Blotting detected bands of the expected size (Reference: PMID 8813074). Further, tissue IHC staining intensity (PMID 8813074) correlates with mRNA expression levels (ENCODE database).<br>• anti-GCLC: (Sigma Aldrich HPA036359): Validated by IHC, Western Blotting (via Orthogonal RNAseq and Capture MS), and Protein array by the Human Protein Atlas (HPA) project. Manufacture's recommended usage for WB: 0.04-0.4 µg/mL. Manufacture's recommended usage for IHC: 1:50-1:200. Reference: https://www.proteinatlas.org/ENSG00000001084-GCLC/summary/antibody<br>• anti-Beta-actin (Sigma Aldrich A1978, Clone AC-15): validated by vendor for western blotting with human samples. Manufacture's recommended usage for WB: 0.5-1 µg/mL using cell extract of human foreskin fibroblasts or chicken fibroblasts. Reference: https://www.sigmaaldrich.com/US/en/product/sigma/a1978<br><br>Primary anitbodies used for immunohistochemistry (IHC):<br>• anti-GCSc (SantaCruz sc390811): according to the manufacturer, sc390811 (H-5) is recommended for detection of g-GCSc of mouse, rat and human origin by immunohistochemistry, including paraffin-embedded sections. Starting dilution 1:50, dilution range 1:50-1:500. Reference: https://datasheets.scbt.com/sc-390811.pdf or https://www.scbt.com/p/gamma-gcsc-antibody-h-5<br>• anti-CD45 (Santa Cruz sc1178): according to the manufacturer, sc1178 is recommended for detection of CD45 of mouse, rat and human origin by immunohistochemistry (including paraffin-embedded sections). Starting dilution 1:50, dilution range 1:50-1:500. Refence: https://datasheets.scbt.com/sc-1178.pdf or https://www.scbt.com/p/cd45-antibody-35-z6<br>• anti-F4/80 (D2S9R) (Cell Signaling Technology 70076): according to the manufacturer, 70076 is recommended for detection of F4/80 of mouse origin by immunohistochemistry (including paraffin-embedded sections). Recommended dilution: 1:125 - 1:500. Reference: https://www.cellsignal.com/products/primary-antibodies/f4-80-d2s9r-rabbit-monoclonal-antibody/70076<br>• anti-DNA/RNA Damage (Clone 15A3) (Abcam 62623): according to the manufacturer, 62623 recognizes 8-hydroxy-2'-deoxyguanosine, 8-hydroxyguanine and 8-hydroxyguanosine by immunohistochemistry. Recommended usage dilution: 1 mg/mL. Reference: https://www.abcam.com/en-us/products/primary-antibodies/dna-rna-damage-antibody-15a3-ab62623<br>• anti-Nrf2 (Abcam 31163): according to the manufacturer, 31163 is recommended for detection of NRF2 of human, mouse (predicted), rat (predicted), chicken (predicted), and cow (predicted) by immunohistochemistry (including paraffin-embedded sections). Recommended dilution: 1:100. Reference: https://doc.abcam.com/datasheets/inactive/ab31163/en-us/nrf2-antibody-ab31163.pdf<br><br>Primary antibodies used for immunofluorescence:<br>• anti-GCSc (SantaCruz sc390811): according to the manufacturer, sc390811 is recommended for detection of g-GCSc of mouse, rat and human origin by immunofluorescence. Starting dilution 1:50, dilution range 1:50-1:500. Reference: https://datasheets.scbt.com/sc-390811.pdf or https://www.scbt.com/p/gamma-gcsc-antibody-h-5<br>• anti-CD45 (Proteintech 31243-1-AP): according to the manufacturer, 31243-1-AP is recommended for detection of CD45 of mouse, rat and human origin by immunofluorescence. Recommended dilution: 1:50-1:500. Reference: https://www.ptglab.com/products/CD45-Antibody-31243-1-AP.htm<br><br>Antibody used for BrdU staining:<br>• anti-BrdU-FITC (Biolegend 364104, clone 3D4): validated by vendor for Intracellular Staining for Flow Cytometry (ICFC). Manufacture's recommended usage for flow cytometric staining: 5µL/10^6 cells in 100 µL staining volume. Reference: https://www.biolegend.com/fr-lu/products/fitc-anti-brdu-antibody-10623 |

# Eukaryotic cell lines

Policy information about cell lines and Sex and Gender in Research

| | |
|---|---|
| Cell line source(s) | • HT-1080 (CCL-121, Lot: 70048591), NCI-H522 (CRL-5810, Lot: 70070052), and SKMEL28 (HTB-72, Lot: 70056504) were purchased from the American Type Culture Collection (ATCC).<br>• HCT-116 was purchased from the National Cancer Institute Division of Cancer Treatment and Diagnosis (NCI-DCTD) (Vial designation: 0507662, Lot: 0507660-0507663).<br>• HCC1806 (ATCC, CRL-2335), MDAMB231 (ATCC, HTB-26), MDAMB468 (ATCC, HTB-132), and JIMT-1 (DSMZ; ACC-589) were donated by Dr. Joan Brugge (Harvard University).<br>• PC3 (ATCC, CRL-1435) was donated by Dr. Marie Hanigan (University of Oklahoma).<br>• PaTu8988s (DSMZ, ACC-204) and SUIT2 (JCRB, 1094) were donated by Dr. Stephano Mello (University of Rochester).<br>• H1299 (ATCC, CRL-5803) and A549 were donated by Dr. Brian Altman (University of Rochester).<br>• 786-O (ATCC, CRL-1932) and A498 (ATCC, HTB-44) were donated by Dr. Phillip Rappold (University of Rochester). |
| Authentication | The authenticity of HCC1806, MDAMB231, MDAMB468, JIMT-1, and A549 was confirmed by STR profiling. The authenticity of the remaining cell lines was not tested. |
| Mycoplasma contamination | All cell lines tested negative for mycoplasma contamination using MycoAlert Mycoplasma Detection Kit (Lonza LT07-418). |
| Commonly misidentified lines<br>(See ICLAC register) | None of the 15 cell lines used in this study appear on ICLAC's Register of Misidentified Cell Lines (Version 13, released 26 April 2024). |

# Animals and other research organisms

Policy information about studies involving animals; ARRIVE guidelines recommended for reporting animal research, and Sex and Gender in Research

| | |
|---|---|
| Laboratory animals | • 12-18 weeks female C57BL/6 Gclc f/f mice were crossed with the MMTV-PYMT (Jackson Lab, #022974) and Rosa26-CreERT2 (Jackson Labs, #008463) mouse strains.<br>• 8-12 weeks female athymic nude NU/J (Jackson Labs, #002019) were used for orthotopic tumor allografts and xenografts.<br>• Mice were housed in standard individually ventilated cages connected to a filtered air circulation system under pathogen-free conditions. Animals were maintained on a 12 h light/12 h dark cycle, with lights on during the daytime. Housing rooms were kept at controlled temperature (18-26°C) and relative humidity (30–70%). Food and water were provided ad libitum. |
| Wild animals | No wild animals were used in this study. |
| Reporting on sex | Since the major animal models of this study were models of breast cancer, all animals used in this study were females. Sex of mice were determined at 14-28 days using protocols approved by the University Committee on Animal Resources at the University of Rochester Medical Center. |
| Field-collected samples | Study did not involve samples collected from the field. |
| Ethics oversight | All animal studies were performed according to protocols approved by the University Committee on Animal Resources at the University of Rochester Medical Center. |

Note that full information on the approval of the study protocol must also be provided in the manuscript.

# Plants

| | |
|---|---|
| Seed stocks | N/A |
| Novel plant genotypes | N/A |
| Authentication | N/A |

