## [Peer Review File · Nature]

Catabolism of extracellular glutathione supplies cysteine to support tumors

Corresponding Author: Dr Isaac Harris

Version 0:

Reviewer comments:

Referee #1

(Remarks to the Author)

GSH and cysteine are required for tumor growth. Here, Hecht and colleagues demonstrate that high levels of extracellular GSH function as a source of cysteine within cancer cells. They go on to show that inhibiting GGT activity and subsequent GSH catabolism partially suppress tumor growth. I enjoyed reading this manuscript, which builds on an older discovery from Harry Eagle. The authors key premise is that the major role of GSH in tumorigenesis is supply of the amino acid cysteine rather than suppressing ferroptosis or as a broad based anti-oxidant– this is highly interesting. At its current state however, further evidence is needed demonstrate the generality of these findings.

Major Comments

1. The GCLC deletion in tumors decreases GSH levels dramatically but not to zero (its hard to know exact concentrations given the 'relative' scale bar). Given millimolar levels of GSH in cells, even a small remaining fraction could support growth. Can the authors fully deplete GSH in cells and examine its role in proliferation/tumor growth?
2. The low levels of cystine in TIF/serum relative to GSH are very cool. Because this manuscript suggests this finding is the norm in cancer, I believe that measuring GSH/cystine in the TME from patient tumors is critical as a mechanism to support their findings.
3. Where is GSH going in tumors? The authors make claims about its use for metabolism, protein synthesis. They should conduct tracing experiments to figure this out. What fraction of cysteine from GSH is going to cysteine in proteins vs metabolic pathways?
4. How do serum levels of cystine/GSH compare to WT mice not harboring tumors?
5. The ferroptosis experiments are key and should be repeated in a panel of cell lines that are both sensitive and resistant to ferroptosis.
6. In Figure 2h, supplementing GSH did not increase intracellular cystine levels, which is puzzling since the manuscript emphasizes that extracellular GSH is an important source of intracellular cystine.
7. Line 179, how is GSH driving drug resistance? What is meant by this title? The authors should also comment that GSH is used to detoxify drugs.
8. Beyond GGT, is it possible that other enzymes involved in extracellular GSH catabolism or uptake could contribute to tumor growth regulation? Is GGT commonly over-expressed or mutated in cancer.
9. The authors should use a genetic knockdown of GGT in their tumors of interest to complement their pharmacological studies.
10. As mentioned by the authors in the discussion section, it remains unclear which types of cancer are more dependent on GSH catabolism. The authors primarily used breast cancer cells as their model. What is the evidence suggesting that breast cancer is particularly reliant on GSH catabolism. The authors should measure GSH levels in different cancer cell lines or analyze the expression of GGT to investigate whether breast cancer or other types of cancer are especially dependent on GSH catabolism.

Minor comments

- 1) Citation missing for line 104

Referee #2

(Remarks to the Author)

Hecht and Zocchi et al use mass spectrometry metabolomics in cell culture and murine models to demonstrate the role of extracellular glutathione (GSH) in supporting tumor cell cysteine levels and proliferation. In their first murine experiment, the authors show that deletion of *Gclc*, the enzyme involved in the rate-limiting step in GSH synthesis, is not necessary for to tumor growth in *Gclc* replete animals. Intrigued by high levels of GSH in the plasma and TIF, the authors investigated and found that GSH supports cell proliferation in cystine-deprived conditions by rescuing cysteine fates in the cell, such as GSH and (hypo)taurine. Although the authors find GGT activity is difficult to genetically remove, presumably due to redundancy in function between several isoforms, GGT1 overexpressing cells can support the growth of neighboring, co-cultured cells, demonstrating the sufficiency of GGT activity to support GSH catabolism. Hecht and Zocchi et al then perform a drug screen that reveals increased sensitivity to a GGT inhibitor (GGsTop) in cystine-deprived, GSH-replete media. Indeed, treating cells and mice with GGsTop reduces cell proliferation and tumor growth, while also increasing serum GSH and decreasing cysteine and cystine in the serum and tumor. The finding that NAC is able to ameliorate the effect of GGsTop on tumor growth indicates that this effect is due to cysteine starvation. Overall, this is an interesting and well written study that clearly shows the capability of extracellular glutathione to contribute to cysteine metabolism. However, a few important discrepancies need to be addressed.

Major Comments:

My biggest issue relates to the physiological balance of glutathione in their model. The authors show two seemingly contradictory results 1) Glutathione is highly abundant in the tumor interstitial fluid (~30x higher abundance than the plasma) 2) Intratumoral glutathione breakdown provides cysteine to tumor cells. If the tumors are net consumers of glutathione, one would expect a reverse of this concentration gradient, as cancer cell glutathione catabolism depletes glutathione in the tumor, which could be replenished by perfusion from the blood. However, this opposite concentration relationship would suggest that tumor cells are instead net exporters of glutathione, which would mean that the increased glutathione in the TIF is a net cysteine loss for the cancer cells. How do the authors reconcile this discrepancy?

To this end, do GCLC KO tumors have comparably high glutathione in the TIF compared to GCLC WT tumors (which would inform whether the high intratumoral glutathione derives from cancer cells)? Do tumors injected into different tumor sites have different TIF glutathione levels (perhaps suggesting a role for different non-cancer cells to provide glutathione locally in the tumor microenvironment)? One would imagine that GGT levels in each tumor microenvironment could be an important factor here as very high levels would deplete glutathione while very low levels would insufficiently provide glutathione breakdown products to cancer cells. Overall, this seems to be a major missing piece in the study and needs to be addressed.

Figure 5m is an important functional experiment to validate their mechanism, but it is surprisingly devoid of corroborative data. To tie up their mechanism, the authors should conduct metabolomics on serum and tumors from these animals to uncover how NAC treatment affects systemic (NAC, cystine, and glutathione) and intratumoral (cysteine, glutathione, (hypo)taurine) cysteine metabolism.

While much of the data focuses on glutathione catabolism delivering cysteine to cells, the authors are careful to leave open the possibility that it may also deliver the other constituent parts of glutathione. Can the authors test the relative capability of glutathione to fulfill limitations of glycine (potentially using glycine or serine/glycine deficient media) or glutamate (potentially using a glutaminase inhibitor)?

Minor Comments:

Line 77 "after its release..." The peptidase step is somewhat mysterious throughout the paper. Can the authors elaborate on what is known about these peptidase enzymes? Perhaps some citations would be useful here.

Referee #3

(Remarks to the Author)

Hecht et al. discover that GSH catabolism promotes tumor growth in a gamma-glutamyltransferase dependent manner, thus providing novel evidence that extracellular glutathione in the tumor microenvironment is capable of acting as an amino acid reservoir. To test this, the authors use the Rosa26-CreERT2 *Gclcf* mouse they generated and implanted formed tumors into C57BL/6 mice, in which they activated Cre-recombinase. The authors show GGT activity is necessary and sufficient to promote cancer cell survival, thus highlighting a new actionable pathway of nutrient acquisition in cancer. They show that treatment with GGsTOP, a GGT inhibitor, in vivo, increased serum GSH and impaired tumor growth that was rescued by NAC (a cysteine donor).

The key novel insight from this work is that it highlights a potential reason why targeting GSH via *Gclc* to induce ferroptosis may be challenging in vivo. This work also offers evidence to support the importance of distinguishing between in vitro and in vivo contexts, where metabolite availability significantly differs, as related to targeting of ferroptosis. However, the use of multiple model systems and cancer types, without consistently presenting data across all models obscures the cohesiveness of the proposed mechanism. Critically, while the in vitro mechanism clearly demonstrates that GGT activity is

sufficient to support GSH catabolism and promote the survival of surrounding cells under cystine-depleted conditions, its relevance to the *in vivo* findings shown in Fig. 1 remains uncertain. Bridging this gap will require further experiments to clearly elucidate the extent to which this mechanism operates *in vivo*. Specifically, several alternative possibilities (described below) need to be experimentally investigated to clarify the contribution of the observed extracellular GSH catabolism phenotype to tumor growth *in vivo* and it will be important to understand the extent to which this mechanism occurs in the setting of Gclc inhibition (in Gclc-deficient tumor cells, or with BSO treatments).

Major comments

1. The provided rationale for exploring that extracellular GSH may be critical to tumor growth comes from *in vivo* experiments demonstrating Gclc deletion in tumors failed to perturb tumor growth. However, given that in Fig 1c there are significant decreases in tumor GSH, and yet no differences in tumor growth does this not suggest that GSH is not required for these tumor cells to survive, and that already in the absence of *de novo* GSH synthesis the cancer cells have adequate levels of upstream metabolites (i.e., cysteine) to sustain growth? Please clarify this rationale based on the data shown in Fig. 1.
2. Are there phenotypic, metabolic, or redox features (are basal ROS levels different?) in the Gclc^{-/-} versus WT tumors that the authors can demonstrate that would indicate extracellular GSH catabolism would confer an increased tumor survival benefit, thus strengthening the premise of their hypothesis?
3. What are the intracellular cysteine levels of the tumors shown in Fig. 1c? It is possible cysteine levels in the Gclc^{-/-} tumors are higher because cysteine is not being synthesized into GSH, which would thus decrease the cellular requirement for extracellular cystine import.
4. In Figure 1f, the authors move into a 4T1 orthotopic model, and while this reviewer greatly appreciates the use of multiple breast cancer models as well as the PDAC model, it creates difficulty in data interpretation as not all of the data is shown for the different lines. Generating the GSH measurements in plasma and TIF from the mice in Fig. 1a-e would be a way to connect these two independent datasets in the same model as it is unclear if the observed phenotypes in Fig. 1f can also be applied to the data shown in Fig 1a-e. The difficulty comparing data in the different models also occurs later in the manuscript as the authors shift the HCC1806 and the phenotypes shown in Figure 1 are uncertain to be occurring in this model.
5. Another interpretation of the data shown in Fig. 1f is that the tumor cells are exporting glutathione (such as via MRPs/ABC transporters or GSH solute exchangers), thus increasing its presence in TIF. Please provide experiments to account for this possibility, as this could change the interpretation of the data. Although it may not mimic the *in vivo* conditions, demonstrating *in vitro* in WT or Gclc^{-/-} cells and conducting intracellular and extracellular GSH, GSSG, GSH/GSSG, cysteine, cystine, glutamate +/- exogenous GSH and +/- export transporter inhibition may be one way to provide insight into this possibility and would be sufficient exploration of this alternate explanation.
6. Please include non-tumor bearing controls in Fig. 1f for the plasma measurements (this reviewer appreciates that fluid from non-tumor bearing mammary fat pad conditions is not feasible). Plasma levels of mice bearing 4T1 WT and 4T1 Gclc-deficient tumors are also needed to connect the data in Fig 1-e and Fig 1f between the different models, as it could change the data interpretation if TIF from 4T1 Gclc-deficient tumors contains the same GSH, cystine, and glutamate levels as TIF from WT tumors.
7. Different mouse strains are known to have significantly different GSH and GSH/GSSG profiles (PMID: 24613380) and this should be taken into consideration when comparing data from different mouse strains (such as C57BL/6 and Balb/c). In Fig. 1f, the interpretation of the data with respect to these different strains would benefit from the inclusion of GSH measurements of plasma from all strains used in this manuscript.
8. The authors show that extracellular GSH can be used as a cysteine source *in vitro*, but have not yet demonstrated if extracellular GSH can be used as a cysteine source *in vivo*. *In vivo* tracing experiments, such as tracing extracellular cystine, such as using a ¹³C6-cystine infusion, and tracking glutathione labeling within the tumors *in vivo* in WT compared to Gclc^{-/-} tumors may be one way to show the extent to which tumor cells rely on cysteine uptake *in vivo* and strengthen the connections between the mechanism observed *in vitro* and the *in vivo* phenotype.
9. One major outstanding question is that although the *in vitro* findings show GSH can be broken down to fuel intracellular metabolism of cysteine *in vitro* in the absence of *de novo* GSH metabolism, this mechanism is not yet fully supported *in vivo*. Based on the authors hypothesis, Gclc^{-/-} tumors, or BSO combinatorial treatments with GGsTOP, should have increased sensitivity to GGsTop because of the increased dependency on GSH catabolism. Is this the case?

Minor comments

1. Where possible please show GSH, GSSG, and GSH/GSSG ratios throughout where GSH measurements are provided (as in Fig. 1c, Fig. 1f)
2. It is unclear if the *in vivo* experiments in Figure 1 represent only 1 independent experiment, please clarify and justify as to why a second independent replication is not provided.
3. Please also show the levels of glutamate for both Figure 1f and ED Data Fig 1

4. Please conduct the experiments shown in ED Fig. 2e also using NAC to distinguish between gclc-associated affects versus cysteine replenishment effects to support the interpretation of the data with differences in cell survival versus proliferation with ferrostatin-1.
5. Please describe the data and the data interpretation more extensively for the panels in Extended Data 3 and 4 in the manuscript text (currently one sentence description in the text). There are a few additional panels (such as in Fig. 5, among others) that are not directly described in the manuscript text.
6. Take care to ensure appropriate replications of experiments are completed (as in Extended Data Fig. 6b, which appears to be 1 replication)
7. In Fig. 3f, the statistical bars are obscuring the data points
8. Please show tumor GSH, GSSG, GSH/GSSG, glutamate levels in Figure 5

Version 1:

Reviewer comments:

Referee #1

(Remarks to the Author)

The authors have answered my queries. I commend them for the amount of work they conducted.
liron bar-peled

Referee #2

(Remarks to the Author)

The authors have done a commendable job of addressing my critiques. Overall, I find the results compelling and highly interesting. I support publication.

Referee #3

(Remarks to the Author)

The authors have sufficiently addressed the raised concerns with the inclusion of new experiments and interpretations, which contribute to the improved strength of the manuscript's conclusions.

Pertaining to major comment #9, the outcomes of the BSO in vivo experiments in ED Fig. 10 involve multiple unresolved and untested variables, including whether the phenotype is recapitulated in gclc^{-/-} lines or is a BSO-specific effect, and whether the modest increase in BSO-induced GGT activity explains the observed functional differences. As the authors appropriately note, these data cannot be interpreted conclusively at this stage. Accordingly, inclusion of this experiment is left to the authors' discretion as conclusions from these findings remain speculative and thus neither strengthen nor undermine the manuscripts' central conclusions.

Within the set of new in vivo experiments generated to address related comments, the authors show that GGT1 expression in tumors does not appear to depend on GCLC status (new ED Fig. 9i). This result provides adequate in vivo context to address the original question of major comment #9 regarding GSH catabolism capacity in the absence of de novo GSH metabolism in vivo, although quantification or representation of the IHC staining across multiple independent tumors could support this conclusion more rigorously.

We appreciate the Reviewers' comments and suggestions. The revisions have significantly strengthened the evidence supporting the manuscript's conclusions. **In total, we have added 91 new data panels and 3 additional Extended Data Figures.**

Specific responses to Reviewer comments:

Reviewer comments – Blue

Author responses – Black

Text from the manuscript – Bolded in Black

Referee #1 (Remarks to the Author):

GSH and cysteine are required for tumor growth. Here, Hecht and colleagues demonstrate that high levels of extracellular GSH function as a source of cysteine within cancer cells. They go on to show that inhibiting GGT activity and subsequent GSH catabolism partially suppress tumor growth. I enjoyed reading this manuscript, which builds on an older discovery from Harry Eagle. The authors key premise is that the major role of GSH in tumorigenesis is supply of the amino acid cysteine rather than suppressing ferroptosis or as a broad based anti-oxidant– this is highly interesting. At its current state however, further evidence is needed demonstrate the generality of these findings.

We appreciate the kind words and interest from Reviewer #1. We hope that the revised manuscript provides further evidence to demonstrate the generality of the findings.

Major Comments

1. The GCLC deletion in tumors decreases GSH levels dramatically but not to zero (its hard to know exact concentrations given the 'relative' scale bar). Given millimolar levels of GSH in cells, even a small remaining fraction could support growth. Can the authors fully deplete GSH in cells and examine its role in proliferation/tumor growth?

We thank Reviewer #1 for these thoughtful comments and suggestions. To fully deplete GSH in cells, we have used CRISPR-Cas9 approaches to generate Gclc WT and Gclc KO cells in the HCC-1806 breast cancer cell line. We characterized two different Gclc KO lines, with varying degrees of residual GSH (**Fig. 1f-1g**). We find that loss of GSH synthesis in breast cancer cells does not slow tumor volume over time (**Fig. 1h**) or decrease tumor mass at endpoint (**Fig. 1i**).

We further examined why GSH levels were not completely depleted in Gclc KO tumors from our orthotopic breast tumor model, which consists of orthotopically transplanting MMTV-PyMT Gclc^{ff} Rosa26-CreERT2 tumor chunks into immunocompetent wild-type C57/B6 mice. The tumors generated in mice consisted of Gclc KO tumor cells, but also cells from the wild-type host mouse. We hypothesized that GSH levels do not reach zero in whole-tumor tissue extracts, in part, due to the presence of wild-type host cells. Indeed, analysis of single-nuclei RNAseq data from non-tumoral breast tissue revealed that several cell types in mammary tissue, including immune cells, express high levels of Gclc (**Extended Data Fig. 1a**). We identified immune cells (CD45⁺ and F4/80⁺ cells) in Gclc KO tumors that retained GCLC expression, potentially contributing to the remaining GSH levels in Gclc KO tumors (**Extended Data Fig. 1b-1c**). **Fig. 1 and Extended Data Fig. 1**, and their legend are shown below.

Figure 1
Figure 1. Intracellular production of GSH is dispensable for tumor growth. **a**, Schematic of the tumor-specific *Gclc* knockout mouse model. Autochthonous tumors from MMTV-PyMT *Gclc*^{fl/fl} Rosa26-CreERT2 mice were excised and orthotopically transplanted into mammary fat pads of wild-type C57BL/6 mice. C57BL/6 mice were treated with vehicle (corn oil; WT) or 50 mg/kg tamoxifen for 5 days (KO). **b**, Relative *Gclc* mRNA levels of WT and KO tumors. Statistical significance assessed by an unpaired two-tailed t-test ($n = 5$ representative of $n = 3$ independent experiments). **c**, GSH levels of WT and KO tumors. Statistical significance assessed by an unpaired two-tailed t-test ($n = 6$ representative of $n = 3$ independent experiments). **d-e**, Volume over time (**d**) and mass at endpoint (**e**) of WT tumors ($n = 11$) and KO tumors ($n = 12$) (representative of $n = 3$ independent experiments). Statistical significance was assessed by two-way ANOVA followed by Šidák's multiple comparisons test for tumor volume and by an unpaired two-tailed t-test for tumor mass. **f-g**, Immunoblot of *Gclc* protein levels (**f**) and GSH levels (**g**) in HCC-1806 human breast cancer cells infected with lentiCRISPR v2 containing non-targeting guides (sgNTC) and guides against GCLC (sgGclc 1 and sgGclc 2). Statistical significance was assessed by ordinary one-way ANOVA with Tukey's multiple comparisons test. **h-i**, Volume over time (**h**) and mass at endpoint (**i**) of orthotopically implanted HCC-1806 sgNTC, sgGclc 1, and sgGclc 2 breast cancer cells. Statistical significance was assessed by ordinary one-way ANOVA with Tukey's multiple comparisons test for volume and by two-way ANOVA followed by Šidák's multiple comparisons test for tumor mass ($n = 7$ representative of $n = 2$ independent experiments). **j**, Concentration of total GSH (tGSH; GSH + 2×GSSG) in serum ($n = 11$) and tumor interstitial fluid ($n = 4$) from *Gclc* WT MMTV-PyMT autochthonous tumors from (a). Statistical significance was assessed by an unpaired two-tailed t-test. **k**, Concentration of tGSH in serum ($n = 8$) and tumor interstitial fluid ($n = 6$) from patients with breast cancer. Statistical significance was assessed by an unpaired two-tailed t-test. Data represented as mean \pm s.e.m., ns, not significant; * p -value < 0.05; ** p -value < 0.01; *** p -value < 0.001; **** p -value < 0.0001.

Extended Data Figure 1

Extended Data Figure 1. Gclc is expressed in non-epithelial cells, and its loss in tumors does cause redox or metabolic alterations. **a**, Single-nucleus RNAseq (snRNAseq) of *Gclc* transcripts from the indicated cells present in human mammary tissue. **b-e**, Autochthonous tumors from MMTV-PyMT *Gclc*^{fl/fl} *Rosa26-CreERT2* mice were excised and orthotopically transplanted into mammary fat pads of wild-type C57BL/6 mice, which were treated with vehicle (corn oil; WT) or 50 mg/kg tamoxifen for 5 days (KO). Representative immunohistochemistry (**b**) and immunofluorescence (**c**) staining of indicated proteins in *Gclc* WT and *Gclc* KO mammary tumors. Relative mRNA expression of indicated genes (**d**) and representative immunohistochemistry of indicated proteins (**e**) in *Gclc* WT and *Gclc* KO tumors. Statistical significance was assessed by an unpaired two-tailed t-test in (**d**) ($n = 6$ representative of $n = 3$ independent experiments). **f-j**, Volcano plot of metabolic differences (**f**) and levels of GSSG (**g**), ophthalmic acid (**h**), glutamate (**i**), NEM-cysteine (**j**) from *Gclc* WT and *Gclc* KO tumors. Statistical significance was assessed by an unpaired two-tailed t-test in (**g-j**) ($n = 6$ representative of $n = 3$ independent experiments). Data represented as mean \pm s.e.m., * p -value <0.05 ; ** p -value <0.01 ; *** p -value <0.001 ; **** p -value <0.0001 ; ns, not significant.

2. The low levels of cystine in TIF/serum relative to GSH are very cool. Because this manuscript suggests this finding is the norm in cancer, I believe that measuring GSH/cystine in the TME from patient tumors is critical as a mechanism to support their findings.

We thank Reviewer #1 for their kind words and suggestion to measure levels in TIF/serum from patient tumors. We have obtained human samples and we have conducted quantitative LC-MS metabolomics on TIF and serum and found that, similar to murine tumor models, total GSH levels (tGSH; GSH + 2×GSSG), but not cystine levels, are enriched in the TIF compared to the serum of patients with breast cancer (**Fig. 1k and Fig. 2c**). We have also analyzed publicly available datasets of metabolite concentrations in interstitial fluid from patients with cancer¹ (Abbott KL, eLife, 2024, PMID: 38787918). We found that, similar to patients with breast cancer, GSH levels, but not cystine levels, are enriched in the TIF compared to the serum of patients with renal cell carcinoma (**Extended Data Fig. 3c**). **Figs. 1, 2, and Extended Data Fig. 3**, and their legends are shown below.

Figure 1

Figure 1. Intracellular production of GSH is dispensable for tumor growth. **a**, Schematic of the tumor-specific *Gclc* knockout mouse model. Autochthonous tumors from MMTV-PyMT *Gclc*^{ff} Rosa26-CreERT2 mice were excised and orthotopically transplanted into mammary fat pads of wild-type C57BL/6 mice. C57BL/6 mice were treated with vehicle (corn oil; WT) or 50 mg/kg tamoxifen for 5 days (KO). **b**, Relative *Gclc* mRNA levels of WT and KO tumors. Statistical significance assessed by an unpaired two-tailed t-test ($n = 5$ representative of $n = 3$ independent experiments). **c**, GSH levels of WT and KO tumors. Statistical significance assessed by an unpaired two-tailed t-test ($n = 6$ representative of $n = 3$ independent experiments). **d-e**, Volume over time (**d**) and mass at endpoint (**e**) of WT tumors ($n = 11$) and KO tumors ($n = 12$) (representative of $n = 3$ independent experiments). Statistical significance was assessed by two-way ANOVA followed by Šídák's multiple.

comparisons test for tumor volume and by an unpaired two-tailed t-test for tumor mass. **f-g**, Immunoblot of Gclc protein levels (**f**) and GSH levels (**g**) in HCC-1806 human breast cancer cells infected with lentiCRISPR v2 containing non-targeting guides (sgNTC) and guides against GCLC (sgGclc 1 and sgGclc 2). Statistical significance was assessed by ordinary one-way ANOVA with Tukey's multiple comparisons test. **h-i**, Volume over time (**h**) and mass at endpoint (**i**) of orthotopically implanted HCC-1806 sgNTC, sgGclc 1, and sgGclc 2 breast cancer cells. Statistical significance was assessed by ordinary one-way ANOVA with Tukey's multiple comparisons test for volume and by two-way ANOVA followed by Šídák's multiple comparisons test for tumor mass (n = 7 representative of n = 2 independent experiments). **j**, Concentration of total GSH (tGSH; GSH + 2×GSSG) in serum (n = 11) and tumor interstitial fluid (n = 4) from Gclc WT MMTV-PyMT autochthonous tumors from (**a**). Statistical significance was assessed by an unpaired two-tailed t-test. **k**, Concentration of tGSH in serum (n = 8) and tumor interstitial fluid (n = 6) from patients with breast cancer. Statistical significance was assessed by an unpaired two-tailed t-test. Data represented as mean ± s.e.m., ns, not significant; *p-value<0.05; **p-value<0.01; ***p-value<0.001; ****p-value<0.0001.

Figure 2

Figure 2. Extracellular GSH supplies amino acids to promote cancer cell growth and survival in cysteine-free environments. **a**, Schematic of the different mechanisms of cysteine acquisition and utilization. **b**, Concentration of cystine in serum (n = 11) and tumor interstitial fluid (n=4) from Gclc WT MMTV-PyMT autochthonous tumors from **Fig. 1a**. Statistical significance was assessed by an unpaired two-tailed t-test. **c**, Concentration of cystine in serum (n = 8) and tumor interstitial fluid (n = 6) from patients with breast cancer. Statistical significance was assessed by an unpaired two-tailed t-test. **d-f**, HCC-1806 breast cancer cells were grown in control (208 µM cystine; Cys₂), cystine-free (Cys₂-free), cystine-free/GSH-supplemented (750 µM) or

cystine-free/CysGly-supplemented (750 μ M) medium, and cell numbers (**d**), percentages of proliferative (BrdU⁺) (**e**) and apoptotic (Annexin V⁺) (**f**) cells were determined at the indicated timepoints. Statistical significance was analyzed by two-way ANOVA followed by Tukey's multiple comparisons test (n = 3 technical replicates representative of n = 4 independent experiments) for (**d**), two-way ANOVA followed by Šídák's multiple comparisons test (n = 6 technical replicates from n = 3 independent experiments for **e**; n = 9 technical replicates from n = 3 independent experiments for **f**). **g**, Levels of extracellular CysGly in media in HCC-1806 breast cancer cells grown in the indicated media and at 72 hours. One-way ANOVA followed by Tukey's multiple comparisons test (n = 3 technical replicates). **h**, Schematic of ¹³C-cystine stable-isotope labeling approach and metabolomics. **i**, Percent labeling of ¹³C-cystine stable-isotope in indicated species at 72 hours (n = 5 technical replicates). **j**, Schematic of ¹⁵N¹³C-cystine stable-isotope labeling approach and proteomics. **k**, Percent labeling of ¹⁵N¹³C-cystine stable-isotope in indicated species at 96 hours (average of n = 3 technical replicates). Data is representative as mean \pm s.e.m., *p-value<0.05; **p-value<0.01; ***p-value<0.001; ****p-value<0.0001; ns, not significant.

Extended Data Figure 3

Extended Data Figure 3. Total GSH (tGSH) is enriched in the TIF compared to serum across multiple cancer models. **a-c**, Concentration of tGSH and cystine in plasma ($n = 17$) and tumor interstitial fluid ($n = 7$) from the LSL-KrasG12D/+ Trp53f/f Pdx-1-Cre (KP-/-C) mice² bearing pancreatic ductal adenocarcinoma (PDAC) (**a**), in the plasma ($n = 27$) and tumor interstitial fluid ($n = 46$) from renal cell carcinoma (RCC) human patients¹ (**b**), in cell culture media formulations (**c**). **d-i**, Autochthonous tumors from MMTV-PyMT Gclc^{fl/fl} Rosa26-CreERT2 mice

were excised and orthotopically transplanted into the mammary gland or subcutaneously transplanted into the flank of wild-type C57BL/6 mice, which were treated with vehicle (corn oil; WT) or 50 mg/kg tamoxifen for 5 days (KO). tGSH and cystine levels in TIF from orthotopic (WT, n = 4; KO, n = 6) **(d)** or flank (WT, n = 4; KO, n = 3) **(e)** tumors. tGSH in TIF from orthotopic and flank Gclc WT MMTV-PyMT tumors (n = 4) **(f)**. tGSH and cystine levels in serum of mice with orthotopic (WT, n = 11; KO, n = 12) **(g)** or flank (WT, n = 4; KO, n = 5) **(h)** tumors. Serum tGSH and cystine in serum of mice without Gclc WT MMTV-PyMT tumors (n = 8) **(i)**. **j-k**, Serum tGSH **(j)** and cystine **(k)** levels in mice bearing orthotopic (n = 11), flank (n = 4), or no Gclc WT tumors (n = 8). **l-n**, Glutamate levels in plasma and TIF from **Fig. 1k (l)** and from **a-b (m-n)**. Total GSH (tGSH) was calculated as the sum of reduced glutathione (GSH) and twice the amount of oxidized glutathione (GSSG). Statistical significance was assessed by an unpaired two-tailed t-test. Data represented as mean \pm s.e.m., *p-value<0.05; **p-value<0.01; ***p-value<0.001; ****p-value<0.0001; ns, not significant.

3. Where is GSH going in tumors? The authors make claims about its use for metabolism, protein synthesis. They should conduct tracing experiments to figure this out. What fraction of cysteine from GSH is going to cysteine in proteins vs metabolic pathways?

We greatly appreciate this important question from Reviewer #1. First, we examined the metabolic fate of GSH-derived cysteine in cancer cells. GSH with a stable-isotope labeled on the cysteine residue was unavailable; thus, we performed “inverse” stable-isotope labeling, as previously described^{3,4} (Liebergesell TCE, Analytical Chemistry, 2024, PMID: 39367814; Guzelsoy G and Elorza SD, Nature, 2025, PMID: 39972131). We cultured cancer cells in stable-isotope labeled ¹³C-cystine (L-Cystine, 3,3'-¹³C₂; CLM-520-PK) and found cysteine-related metabolites were fully labeled, including glutamyl-cysteine, GSH, GSSG, and hypotaurine (**Fig. 2h-2i and Extended Data Fig. 6k-6l**). After switching cancer cells to a medium deprived of cystine and containing unlabeled GSH, the cysteine-related metabolites became unlabeled (**Fig. 2i**). We employed a similar approach to measure incorporation of GSH-derived cysteine into proteins, using stable-isotope labeled ¹³C¹⁵N-cystine (L-cystine, ¹³C₆¹⁵N₂, CNLM-4244). We found that proteins were fully labeled when cultured in medium containing ¹³C¹⁵N-cystine and proteins became unlabeled when cancer cells were switched to a cystine-free medium containing unlabeled GSH (**Fig. 2j-2k and Extended Data Fig. 6k and 6m**). This pattern held for antioxidant-related enzymes, such as thioredoxin 1 (TXN1) and 2 (TXN2). Interestingly, the small number of proteins that retained labeling from labeled-cystine strongly correlated with their turnover times (**Extended Data Fig. 6n and 6o**). This suggests that potentially, over time, these proteins would incorporate GSH-derived cysteine. Overall, these data demonstrate the incorporation of GSH-derived cysteine into metabolites and proteins in cancer cells. **Fig. 2 and Extended Data Fig. 6** and their legends are shown below.

Figure 2

Figure 2. Extracellular GSH supplies amino acids to promote cancer cell growth and survival in cystine-free environments. **a**, Schematic of the different mechanisms of cysteine acquisition and utilization. **b**, Concentration of total glutathione in serum (n=11) and tumor interstitial fluid (n=4) from autochthonous tumors from MMTV-PyMT *Gclc^{fl/fl} Rosa26-CreERT2* mice orthotopically transplanted into mammary fat pads of wild-type C57BL/6 mice. Statistical significance was assessed by an unpaired two-tailed t-test. **c**, Concentration of total glutathione in serum (n=8) and tumor interstitial fluid (n=6) from patients with breast cancer. Statistical significance was assessed by an unpaired two-tailed t-test. **d-f**, HCC-1806 breast cancer cells were grown in control (208 µM cystine; Cys₂), cystine-free (Cys₂-free), cystine-free/GSH-supplemented (750 µM) or cystine-free/CysGly-supplemented (750 µM) medium, and cell numbers (**d**), percentages of proliferative (BrdU⁺) (**e**) and apoptotic (Annexin V⁺) (**f**) cells were determined at the indicated timepoints. Statistical significance was analyzed by two-way ANOVA followed by Tukey's multiple comparisons test (n=4 independent experiments) for (**d**), two-way ANOVA followed by Šidák's multiple comparisons test (n=3 independent experiments) for (**e-f**). **g**, Levels of extracellular CysGly in media in HCC-1806 breast cancer cells grown in indicated media and at 72 hours. One-way ANOVA followed by Tukey's multiple comparisons test (n=3 technical replicates). **h**, Schematic of ¹³C-cysteine stable-isotope labeling approach and metabolomics. **i**, Percent labeling of ¹³C-cysteine stable-isotope in indicated species at 72 hours. **j**, Schematic of ¹⁵N¹³C-cysteine stable-isotope labeling approach and proteomics. **k**, Percent labeling of ¹⁵N¹³C-cysteine stable-isotope in indicated species at 96 hours. Data is representative as mean ± s.e.m., *p-value<0.05; **p-value<0.01; ***p-value<0.001; ****p-value<0.0001; ns, not significant.

Extended Data Figure 6

Extended Data Figure 6. Supplementation of GSH in cystine-free conditions rescues downstream products of cysteine metabolism. **a-c**, Difference in abundance in the extracellular cysteinylglycine in medium (**a**) and relative abundance of intracellular cysteinylglycine (**b**) and intracellular cysteine (**c**) for HCC-1806 cells grown in the indicated conditions and time points. Statistical significance assessed by one-way ANOVA with

Tukey's multiple comparisons test (n = 2-3 technical replicates). **d-f**, Relative cell number (**d**) and relative abundance of NEM-cysteine (**e**) and cystine (**f**) for HCC-1806 cells grown in the indicated cystine concentrations and time points. Statistical significance assessed by one-way ANOVA with Tukey's multiple comparisons test (n = 3 technical replicates). **g-j**, Relative abundance of intracellular NEM-GSH (**g**), hypotaurine (**h**), taurine (**i**), and ophthalmic acid (**j**). Statistical significance assessed by one-way ANOVA with Tukey's multiple comparisons test (n = 3 technical replicates). Δ Abundance represents the metabolite level difference between conditioned and virgin media. **k**, Schematic of cystine stable-isotope labeling approach for metabolomics and proteomics. **l-m**, Percent labeling from metabolomics (**l**) and proteomics (**m**) at 0 hours, demonstrating complete cystine labeling prior to introducing unlabeled GSH (n = 3 technical replicates). **n-o**, Relationship between peptide half-life⁵ and percent residual heavy-labeling and in cystine-free/GSH-supplemented medium at 96 hours. Data represented as mean \pm s.e.m., *p-value<0.05; **p-value<0.01; ***p-value<0.001; ****p-value<0.0001; ns, not significant.

4. How do serum levels of cystine/GSH compare to WT mice not harboring tumors?

This is a very interesting point. We compared levels of tGSH and cystine in WT mice not harboring tumors and found no difference (**Extended Data Fig. 3i**). Interestingly, we found a decrease in serum tGSH levels between mice with tumors compared with mice without tumors (**Extended Data Fig. 3j**). Previous studies have found that serum GSH levels are lower in patients with colorectal cancer than in healthy volunteers⁶ (Baltruskeviciene E, Tumori Journal, 2018, PMID: 28777429). Overall, this suggests that tumors potentially contribute to GSH breakdown in the serum. **Extended Data Fig. 3** and its legend are shown below.

Extended Data Figure 3

Extended Data Figure 3. Total GSH (tGSH) is enriched in the TIF compared to serum across multiple cancer models. **a-c**, Concentration of tGSH and cystine in plasma ($n = 17$) and tumor interstitial fluid ($n = 7$) from the LSL-KrasG12D/+ Trp53f/f Pdx-1-Cre (KP-/-C) mice² bearing pancreatic ductal adenocarcinoma (PDAC) (**a**), in the plasma ($n = 27$) and tumor interstitial fluid ($n = 46$) from renal cell carcinoma (RCC) human patients¹ (**b**), in cell culture media formulations (**c**). **d-i**, Autochthonous tumors from MMTV-PyMT Gclc^{fl/fl} Rosa26-CreERT2 mice were excised and orthotopically transplanted into the mammary gland or subcutaneously transplanted into the

flank of wild-type C57BL/6 mice, which were treated with vehicle (corn oil; WT) or 50 mg/kg tamoxifen for 5 days (KO). tGSH and cystine levels in TIF from orthotopic (WT, n = 4; KO, n = 6) **(d)** or flank (WT, n = 4; KO, n = 3) **(e)** tumors. tGSH in TIF from orthotopic and flank Gclc WT MMTV-PyMT tumors (n = 4) **(f)**. tGSH and cystine levels in serum of mice with orthotopic (WT, n = 11; KO, n = 12) **(g)** or flank (WT, n = 4; KO, n = 5) **(h)** tumors. Serum tGSH and cystine in serum of mice without Gclc WT MMTV-PyMT tumors (n = 8) **(i)**. **j-k**, Serum tGSH **(j)** and cystine **(k)** levels in mice bearing orthotopic (n = 11), flank (n = 4), or no Gclc WT tumors (n = 8). **l-n**, Glutamate levels in plasma and TIF from **Fig. 1k (l)** and from **a-b (m-n)**. Total GSH (tGSH) was calculated as the sum of reduced glutathione (GSH) and twice the amount of oxidized glutathione (GSSG). Statistical significance was assessed by an unpaired two-tailed t-test. Data represented as mean \pm s.e.m., *p-value<0.05; **p-value<0.01; ***p-value<0.001; ****p-value<0.0001; ns, not significant.

5. The ferroptosis experiments are key and should be repeated in a panel of cell lines that are both sensitive and resistant to ferroptosis.

We appreciate this suggestion from Reviewer #1. Using DepMap and CCLE databases, we obtained sensitivity values to the ferroptosis inducers RSL3 and ML210. Plotting these data, we identified cancer cell lines that were sensitive (HT-1080, NCIH522) or resistant (HCC-1806, SKMEL28, HCT116) to ferroptosis (**Extended Data Fig. 5d**). We confirmed these sensitivities by measuring lipid peroxidation following treatment of cells with RSL3 or upon cystine deprivation (**Extended Data Fig. 5e-5f**). We find that in ferroptosis-sensitive and -resistant cancer cell lines, GSH and cysteinylglycine completely rescue growth and survival upon cystine deprivation (**Extended Data Fig. 5g-5j**). Further, we find that in ferroptosis-sensitive and -resistant cancer cell lines, while antioxidants Ferrostatin-1 and Trolox prevent accumulation of lipid peroxidation (**Extended Data Fig. 5e-5f**), they do not provide a complete rescue of growth and survival upon cystine deprivation (**Extended Data Fig. 5g-5j**). **Extended Data Fig. 5** and its legend are shown below.

Extended Data Figure 5

Extended Data Figure 5. Rescue of cancer cells by supplementation of GSH or cysteinylglycine in cystine-free conditions is distinct from rescue provided by ferroptosis inhibitors. **a-b**, HCC-1806 cells were grown in control (208 μ M cystine; Cys₂) and cystine-free (Cys₂-free) medium with vehicle (DMSO) or ferrostatin-1 (Fer-1, 1 μ M). Percentages of proliferative (BrdU⁺) (**a**) and apoptotic (Annexin V⁺) (**b**) cells were determined. Statistical significance was analyzed by two-way ANOVA followed by Šídák's multiple comparisons ($n = 6$ technical replicates). **c**, HCC-1806 cells were grown in control (208 μ M cystine; Cys₂), cystine-free (Cys₂-free), or cystine-free supplemented with GSH (750 μ M), CysGly (750 μ M), Fer-1 (10 μ M), or Trolox (100 μ M), and relative cell numbers were determined ($n = 8$ technical replicates from $n = 2$ independent experiments). **d**, Sensitivity (area under the curve, AUC) of cancer cell lines to GPX4 inhibitors RSL-3 and ML-210. Ferroptosis-resistant (top 10%, blue) and ferroptosis-sensitive (bottom 10%, red) indicated. Sensitivity values determined from DepMap and CCLE databases. **e-f**, Ferroptosis-sensitive HT-1080 cell line (**e**) and ferroptosis-resistant HCC-1806 cell line (**f**) were grown in control (208 μ M cystine; Cys₂) and cystine-free (Cys₂-free) medium with vehicle (DMSO) or RSL3 (250 nM), alone or with Fer-1 (10 μ M) or Trolox (100 μ M). After 24 hours, cells were stained with C11-BODIPY, and relative green and red fluorescence was determined. Statistical significance was assessed by one-way ANOVA followed by Šídák's multiple comparisons ($n = 6-9$ technical replicates from $n = 2-3$ independent experiments). (**g-j**) Ferroptosis-sensitive cell lines HT-1080 (**g**) and NCIH522 (**h**), and ferroptosis-resistant cell lines HCT116 (**i**) and SKMEL28 (**j**) were grown in control (208 μ M cystine; Cys₂), cystine-

free (Cys₂-free), or cystine-free supplemented with GSH (750 μM), CysGly (750 μM), Fer-1 (10 μM), or Trolox (100 μM), and relative cell numbers were determined (n = 8-12 technical replicates from n = 2-3 independent experiments). Data represented as mean ± s.e.m., *p-value<0.05; **p-value<0.01; ***p-value<0.001; ****p-value<0.0001; ns, not significant.

6. In Figure 2h, supplementing GSH did not increase intracellular cystine levels, which is puzzling since the manuscript emphasizes that extracellular GSH is an important source of intracellular cystine.

This is a good point by Reviewer #1, as we also found this puzzling. We hypothesized that GSH did not increase intracellular cystine levels compared to 200 μM cystine because GSH provided sufficient cysteine for cellular processes but not enough to sustain the surplus intracellular cysteine. We hypothesized that culturing cancer cells in GSH would be similar to culturing them at lower cystine levels. Indeed, we found that, similar to culturing cancer cells with GSH, culturing cancer cells in lower cystine levels (10-20 μM) permitted growth but abolished intracellular cysteine and cystine levels (**Extended Data Fig. 6d-6f**). **Extended Data Fig. 6** and its legend are shown below.

Extended Data Figure 6

Extended Data Figure 6. Supplementation of GSH in cystine-free conditions rescues downstream products of cysteine metabolism. **a-c**, Difference in abundance in the extracellular cysteinylglycine in medium (**a**) and relative abundance of intracellular cysteinylglycine (**b**) and intracellular cysteine (**c**) for HCC-1806 cells grown in the indicated conditions and time points. Statistical significance assessed by one-way ANOVA with

Tukey's multiple comparisons test (n = 2-3 technical replicates). **d-f**, Relative cell number (**d**) and relative abundance of NEM-cysteine (**e**) and cystine (**f**) for HCC-1806 cells grown in the indicated cystine concentrations and time points. Statistical significance assessed by one-way ANOVA with Tukey's multiple comparisons test (n = 3 technical replicates). **g-j**, Relative abundance of intracellular NEM-GSH (**g**), hypotaurine (**h**), taurine (**i**), and ophthalmic acid (**j**). Statistical significance assessed by one-way ANOVA with Tukey's multiple comparisons test (n = 3 technical replicates). Δ Abundance represents the metabolite level difference between conditioned and virgin media. **k**, Schematic of cystine stable-isotope labeling approach for metabolomics and proteomics. **l-m**, Percent labeling from metabolomics (**l**) and proteomics (**m**) at 0 hours, demonstrating complete cystine labeling prior to introducing unlabeled GSH (n = 3 technical replicates). **n-o**, Relationship between peptide half-life⁵ and percent residual heavy-labeling and in cystine-free/GSH-supplemented medium at 96 hours. Data represented as mean \pm s.e.m., *p-value<0.05; **p-value<0.01; ***p-value<0.001; ****p-value<0.0001; ns, not significant.

7. Line 179, how is GSH driving drug resistance? What is meant by this title? The authors should also comment that GSH is used to detoxify drugs.

We thank Reviewer #1 for this point. In this section, we report data from **Fig. 4**, which shows that cancer cells grown in GSH are less sensitive to inhibitors of cystine uptake (i.e., erastin) and the thioredoxin pathway (i.e., auranofin, aurothioglucose, and PX-12). We have changed the title to **"Using GSH as a cysteine source lowers cancer cells' sensitivity to inhibitors of cystine uptake and reduction"**. We also agree that we should comment that GSH is used to detoxify drugs. The updated manuscript now reads:

"We found that when cancer cells rely on GSH for cysteine acquisition, they are more sensitive to GGsTop⁷, a putative inhibitor of GGT activity (Fig. 4b-4c). Conversely, cancer cells were less sensitive to inhibitors of cystine uptake (i.e., erastin) and thioredoxin pathway (i.e., auranofin, aurothioglucose, and PX-12) (Fig. 4b-4d), which mediates the reduction of cystine to cysteine following its import^{8,9}. The decreased sensitivity of cancer to these inhibitors is potentially due to several reasons, including extracellular GSH binding to and inactivating the inhibitors. Alternatively, it could suggest that when cancer cells use GSH as a cysteine source, they rely less on these pathways (i.e., xCT and TXNRD1). In addition, the presence of GSH in the tumor microenvironment argues against the feasibility of targeting these pathways for cancer therapy. Importantly, commonly used cell culture media (e.g., DMEM, RPMI, F12) contain supraphysiological levels of cystine but lack GSH and cysteinylglycine, potentially overestimating the contribution of xCT to cancer cells while undervaluing the importance of GGTs.

8. Beyond GGT, is it possible that other enzymes involved in extracellular GSH catabolism or uptake could contribute to tumor growth regulation? Is GGT commonly over-expressed or mutated in cancer.

The members of the GGT family are the only proteins reported to catabolize extracellular GSH in mammals (**Extended Data Fig. 9a**)¹⁰. We analyzed transcriptomic data of patient samples from The Cancer Genome Atlas (TCGA) and the Genotype-Tissue Expression (GTEx). We observe that GGT enzymes are expressed in virtually all cancer types at various levels, with GGT1 being the most overexpressed isoform in tumor tissues compared to non-tumoral tissues (**Extended Data Fig. 9f-9g**). Interestingly, in certain cancers, such as chromophobe renal cell carcinoma, GGT is under expressed compared to normal tissue (**Extended Data Fig. 9g**). Finally, we analyzed genomic data using the cBioPortal platform and observed that somatic mutations in GGT isoforms are rare in cancer, with an average frequency of approximately 0.15% (**Extended Data Fig. 9h**). **Extended Data Fig. 9** and its legend are shown below.

Extended Data Figure 9

Extended Data Figure 9. Multiple GGT isoforms are expressed in cancer cell lines, but none score as essential in genome-wide genetic screens. **a**, GGT gene family table showing all 13 GGT-related genes in the human genome. GGT1, 5, 6, and 7 are predicted to have enzymatic activity, but only two (GGT1,5) have been functionally characterized. **b**, mRNA expression (Expression Public 24Q2) of cancer cell lines (n=1,517) from DepMap's Cancer Cell Line Encyclopedia (CCLE). **c**, Protein levels from cancer cell lines from DepMap's CCLE (Proteomics GGT1: P19440 (n=339), GGT5: P36269 (n=108), GGT6: Q6P531 (n=45), GGT7: Q9UJ14 (n=375)). **d**, mRNA and protein levels of GGT isoforms in the cell line HCC1806, using data from **(b)** and **(c)**. **e**, Histogram (splined curves) of gene dependency of cancer cell lines from Dependency Map (DepMap) (Public 24Q2+Score Chronos). A lower score suggests a gene is more likely to be essential in a cell line. A score of 0 indicates a non-essential gene, while -1 reflects the median for essential genes. MTOR and GAPDH are shown as examples of common essential genes. **f**, mRNA expression of GGT enzymes was analyzed in human tumors from 32 different TCGA projects. **g**, GGT1 mRNA expression levels in human tumoral and non-tumoral tissues from the Cancer Genome Atlas Program (TCGA) and the Genotype-Tissue Expression (GTEx) databases. Black lines represent median values. **h**, The mutational landscape and somatic mutation frequency (SMF) of each GGT gene was analyzed in 105,260 cancer patient samples across 226 non-redundant studies using the cBioPortal platform revealing a low mutation frequency (mean = 0.15%). **i**, Autochthonous tumors from MMTV-PyMT Gclc/f Rosa26-CreERT2 mice were excised and orthotopically transplanted into the mammary gland or subcutaneously transplanted into the flank of wild-type C57BL/6 mice, which were treated with vehicle (corn oil; WT) or 50 mg/kg tamoxifen for 5 days (KO). Immunohistochemistry of GGT1 protein expression in Gclc WT and KO orthotopic and flank tumors. Statistical significance in **(g)** was analyzed by an unpaired t-test. **p-value<0.01; ***p-value<0.001; ****p-value<0.0001; ns, not significant.

9. The authors should use a genetic knockdown of GGT in their tumors of interest to complement their pharmacological studies.

We appreciate this suggestion from the Reviewer. We took three separate approaches to examine the necessity of GGT1 in sustaining GSH catabolism and cysteine supply to cancer cells. We generated GGT1 knockdown cell lines using CRISPRi approaches and GGT1 knockout cell lines (polyclonal and single-cell clones) using CRISPR-Cas9 approaches. GGT1 knockdown cell lines had minimal GGT1 mRNA (**Extended Data Fig. 8a**) but still retained significant GGT activity (**Extended Data Fig. 8b**), and these cells were still rescued by GSH in cysteine-depleted conditions (**Extended Data Fig. 8c**). GGT1 knockout cell lines showed decreased GGT activity (especially in GGT1 KO single-cell clones); however, in both models, we observed a rescue of cancer cells under cysteine-free/GSH-supplemented conditions (**Extended Data Fig. 8d-8g**). Overall, these data suggest that remaining GGT activity, supplied by other GGT family members or uncharacterized proteins, is sufficient to catabolize GSH and provide a cysteine supply to cancer cells.

We decided to extend our findings involving GGT1-overexpressing cells and explore the impact of increased GGT1 expression on tumor growth (**Fig. 3**). We hypothesized that, due to the abundance of GSH in the TIF, GGT1-overexpressing cells would grow faster in vivo. Indeed, while GGT1-overexpressing cells in vitro do not grow faster in cell culture medium containing cystine (**Fig. 3g**), GGT1-overexpressing cells in vivo showed increased tumor volume over time and increased tumor mass at endpoint (**Fig. 3h-3i**). **Extended Data Fig. 8 and Fig. 3** and their legends are shown below. Also, in the Results, it now states:

“Several enzymes possess gamma-glutamyl transferase activity¹⁰, with GGT1 being the most catalytically active isoform¹¹. We took three separate approaches to examine the necessity of GGT1 in sustaining GSH catabolism and cysteine supply to cancer cells. We generated GGT1 knockdown cell lines using CRISPRi approaches and GGT1 knockout cell lines (polyclonal and single-cell clones) using CRISPR-Cas9 approaches. GGT1 knockdown cell lines had minimal GGT1 mRNA (**Extended Data Fig. 8a**) but still retained significant GGT activity (**Extended Data Fig. 8b**) and grew similarly to control cell lines in cysteine-replete and cysteine-depleted/GSH-supplemented conditions (**Extended Data Fig. 8c-8f**). GGT1 knockout cell lines showed decreased GGT activity (especially in GGT1 KO single-cell clones); however, in both models, we observed a rescue of cancer cells under cysteine-free/GSH-supplemented conditions (**Extended Data Fig. 8g-8i**). Cancer cells express multiple GGT isoforms (**Extended Data Fig. 9a-9d**), but none of these isoforms display dependency in cancer cell line genetic screens (**Extended Data Fig. 9e**). Further, expression of GGT1 in tumors was not dependent on GCLC expression or site of tumor growth (**Extended Data Fig. 9f**). Overall, these data suggest that remaining GGT activity, supplied by other GGT

family members or uncharacterized proteins, is sufficient to catabolize GSH and provide a cysteine supply to cancer cells. Separately, we hypothesized that GGT1 was potentially sufficient to support the breakdown of GSH in the extracellular environment. Overexpression of GGT1 in cells (i.e., GGT1⁺)¹² resulted in higher GGT1 protein levels (Fig. 3a) and GGT activity (Fig. 3b-3c). Lower levels of GSH were required to rescue GGT1⁺ cells in cystine-free conditions compared to control cells (Fig. 3d), suggesting that catabolism by GGT is potentially a rate-limiting step in GSH-dependent rescue of cancer cells. We hypothesized that GSH catabolism by cells with high GGT activity could support surrounding cells in a paracrine fashion. To test this, we co-cultured GGT1⁺ cells with wild-type (WT) cells using a transwell assay (Fig. 3e). Even though GSH levels were below the threshold for rescuing WT cells in cystine-free conditions, co-culturing with GGT1⁺ cells permitted a complete rescue of WT cells (Fig. 3f). Next, we found that GGT⁺ cells grew faster but only in cystine-free/GSH-supplemented conditions (Fig. 3g). Finally, this translated to faster growth of GGT⁺ cells in vivo (Fig. 3h-3i). These findings demonstrate that GGT activity is sufficient to support GSH catabolism and survival of surrounding cells in cystine-depleted conditions. Furthermore, this suggests that non-tumorigenic tissues or cells in the tumor microenvironment with high GGT activity could potentially drive tumor growth and progression by catabolizing GSH and supplying amino acids in a paracrine (or endocrine) manner.”

Extended Data Figure 8

Extended Data Figure 8. Reduction or deletion of GGT1 expression does not impair GSH-mediated rescue of cells in cystine-free conditions. **a-b**, HCC-1806 cells were transduced with CRISPRi lentiviral vector containing sgRNA guides targeting the GGT1 promoter. GGT1 mRNA (**a**) (n = 2 technical replicates) and GGT activity (**b**) (n = 8 technical replicates from n = 3 independent experiments) in control (208 μ M cystine) medium were determined. **c**, GGT1 knockdown cells from (**a-b**) were grown in control (208 μ M cystine; Cys₂), cystine-free (Cys₂-free), or cystine-free/GSH-supplemented (750 μ M) medium for the indicated timepoints, and cell numbers were determined. **d-e**, HCC-1806 cells were transduced with CRISPR-Cas9 lentiviral vector containing sgRNA guides targeting GGT1 and maintained as a polyclonal population. GGT activity (**d**) (n = 6 technical replicates from n = 3 independent experiments) and cell numbers (**e**) (n = 3-6 technical replicates from n = 3 independent experiments) for cells grown in control (208 μ M cystine; Cys₂), cystine-free (Cys₂-free), or cystine-free/GSH-supplemented (750 μ M) medium for the indicated time points were determined. (**f-g**) Single-cell clones were established from the polyclonal population, and GGT activity (**f**) (n = 3 technical replicates from n = 2 independent experiments) and cell numbers (**g**) (n = 4-6 technical replicates from n = 2 independent experiments)

for cells grown in control (208 μM cystine; Cys₂), cystine-free (Cys₂-free), or cystine-free/GSH-supplemented (750 μM) medium for the indicated timepoints were determined.

Figure 3

Figure 3. GGT1 is sufficient to promote GSH catabolism and tumor growth. **a-b**, Immunoblot analysis of human GGT1 (**a**) and GGT activity (**b**) in wild-type (WT) and GGT1-overexpressing (GGT1⁺) PC3 prostate cancer cells. Statistical significance was assessed in (**b**) by an unpaired two-tailed t-test ($n = 3$ technical replicates from $n = 2$ independent experiments). **c**, GGT histochemical stain (GMNA method) of wild-type (WT) and GGT1-overexpressing (GGT1⁺) PC3 prostate cancer. Serine-Borate was used to competitively inhibit GGT (negative control). **d**, WT and GGT1⁺ PC3 cells were grown in control media (208 μM cystine) or cystine-free media supplemented with indicated GSH concentrations. After 72 hours, cell numbers were quantified, and the percentage of rescue ((cell numbers in indicated media – cell numbers in cystine-free media)/(cell numbers in control media)) was determined ($n = 6$ technical replicates from $n = 3$ independent experiments). **e**, Schematic of non-contact co-culture experiments using 0.4 μm PET membrane transwell inserts in media containing low concentrations of GSH (250 μM), which were insufficient to rescue the growth of WT cells in cystine-depleted conditions. **f**, Relative cell numbers of WT, GGT1⁺, or WT cells co-cultured with GGT1⁺ cells in control, cystine-depleted, or cystine-depleted/GSH-supplemented (250 μM) conditions. Statistical significance was assessed by two-way ANOVA with Tukey's multiple comparisons test ($n = 8$ technical replicates from $n = 4$ independent experiments). **g**, Fold change in cell numbers at indicated time points for WT and GGT1⁺ cells grown in medium containing 200 μM cystine/0 μM GSH (left) or 0 μM cystine/250 μM GSH (right). Statistical significance was assessed by two-way ANOVA ($n = 7$ technical replicates from $n = 3$ independent experiments). **h-i**, Volume over time (**h**) and mass at endpoint (**i**) of xenograft tumors from WT ($n = 26$) and GGT1⁺ ($n = 28$) PC3 cells. Statistical

significance was assessed by two-way ANOVA for **(h)** and an unpaired two-tailed t-test for **(i)** (n = 2 independent experiments). Data represented as mean ± s.e.m., ***p-value<0.001; ****p-value<0.0001; ns, not significant.

10. As mentioned by the authors in the discussion section, it remains unclear which types of cancer are more dependent on GSH catabolism. The authors primarily used breast cancer cells as their model. What is the evidence suggesting that breast cancer is particularly reliant on GSH catabolism. The authors should measure GSH levels in different cancer cell lines or analyze the expression of GGT to investigate whether breast cancer or other types of cancer are especially dependent on GSH catabolism.

This is a very good point. We have analyzed the expression of GGTs and found that tumors express GGTs at varying levels (**Extended Data Fig. 9f**). Notably, GGT1 is higher in breast cancers compared to normal breast tissues (**Extended Data 9g**). This phenotype is observed across several cancer subtypes, with papillary and clear cell renal cell carcinomas expressing the highest levels of GGT1. The exception to this trend was chromophobe renal cell carcinomas, which had lower levels of GGT1 compared to normal kidney tissue. **Extended Data Fig. 9** and its legend are shown below. Also, in the Discussion, it now states:

“Further, less is known about whether a specific subtype of cancer relies more on GSH catabolism. Renal cell carcinomas (papillary and clear cell) express some of the highest levels of GGT. Surprisingly, another subtype of renal cell carcinoma (chromophobe) contains the lowest GGT1 expression levels amongst tumors. Indeed, GGT1 has been studied in the context of kidney cancer, with pro- and anti-tumor effects being attributed¹³⁻¹⁵.”

Extended Data Figure 9

Extended Data Figure 9. Multiple GGT isoforms are expressed in cancer cell lines, but none score as essential in genome-wide genetic screens. **a**, GGT gene family table showing all 13 GGT-related genes in the human genome. GGT1, 5, 6, and 7 are predicted to have enzymatic activity, but only two (GGT1,5) have been functionally characterized. **b**, mRNA expression (Expression Public 24Q2) of cancer cell lines (n=1,517) from DepMap's Cancer Cell Line Encyclopedia (CCLE). **c**, Protein levels from cancer cell lines from DepMap's CCLE (Proteomics GGT1: P19440 (n=339), GGT5: P36269 (n=108), GGT6: Q6P531 (n=45), GGT7: Q9UJ14 (n=375)). **d**, mRNA and protein levels of GGT isoforms in the cell line HCC1806, using data from **(b)** and **(c)**. **e**, Histogram (splined curves) of gene dependency of cancer cell lines from Dependency Map (DepMap) (Public 24Q2+Score Chronos). A lower score suggests a gene is more likely to be essential in a cell line. A score of 0 indicates a non-essential gene, while -1 reflects the median for essential genes. MTOR and GAPDH are shown as examples of common essential genes. **f**, mRNA expression of GGT enzymes was analyzed in human tumors from 32 different TCGA projects. **g**, GGT1 mRNA expression levels in human tumoral and non-tumoral tissues from the Cancer Genome Atlas Program (TCGA) and the Genotype-Tissue Expression (GTEx) databases. Black lines represent median values. **h**, The mutational landscape and somatic mutation frequency (SMF) of each GGT gene was analyzed in 105,260 cancer patient samples across 226 non-redundant studies using the cBioPortal platform revealing a low mutation frequency (mean = 0.15%). **i**, Autochthonous tumors from MMTV-PyMT *Gclc/f Rosa26-CreERT2* mice were excised and orthotopically transplanted into the mammary gland or subcutaneously transplanted into the flank of wild-type C57BL/6 mice, which were treated with vehicle (corn oil; WT) or 50 mg/kg tamoxifen for 5 days (KO). Immunohistochemistry of GGT1 protein expression in *Gclc* WT and KO orthotopic and flank tumors. Statistical significance in **(g)** was analyzed by an unpaired t-test. **p-value<0.01; ***p-value<0.001; ****p-value<0.0001; ns, not significant.

Minor comments

1) Citation missing for line 104

Thank you for pointing this out. We have included the following references for line 104:

Harris, I. S. *et al.* Glutathione and thioredoxin antioxidant pathways synergize to drive cancer initiation and progression. *Cancer Cell* **27**, 211-222 (2015). <https://doi.org/10.1016/j.ccell.2014.11.019>

Ogiwara, H. *et al.* Targeting the Vulnerability of Glutathione Metabolism in ARID1A-Deficient Cancers. *Cancer Cell* **35**, 177-190 e178 (2019). <https://doi.org/10.1016/j.ccell.2018.12.009>

McGuirk, S. *et al.* Resistance to different anthracycline chemotherapeutics elicits distinct and actionable primary metabolic dependencies in breast cancer. *Elife* **10** (2021). <https://doi.org/10.7554/eLife.65150>

Referee #2 (Remarks to the Author):

Hecht and Zocchi et al use mass spectrometry metabolomics in cell culture and murine models to demonstrate the role of extracellular glutathione (GSH) in supporting tumor cell cysteine levels and proliferation. In their first murine experiment, the authors show that deletion of *Gclc*, the enzyme involved in the rate-limiting step in GSH synthesis, is not necessary for tumor growth in *Gclc* replete animals. Intrigued by high levels of GSH in the plasma and TIF, the authors investigated and found that GSH supports cell proliferation in cystine-deprived conditions by rescuing cysteine fates in the cell, such as GSH and (hypo)taurine. Although the authors find GGT activity is difficult to genetically remove, presumably due to redundancy in function between several isoforms, GGT1 overexpressing cells can support the growth of neighboring, co-cultured cells, demonstrating the sufficiency of GGT activity to support GSH catabolism. Hecht and Zocchi et al then perform a drug screen that reveals increased sensitivity to a GGT inhibitor (GGsTop) in cystine-deprived, GSH-replete media. Indeed, treating cells and mice with GGsTop reduces cell proliferation and tumor growth, while also increasing serum GSH and decreasing cysteine and cystine in the serum and tumor. The finding that NAC is able to ameliorate the effect of GGsTop on tumor growth indicates that this effect is due to cysteine starvation. Overall, this is an interesting and well written study that clearly shows the capability of extracellular glutathione to contribute to cysteine metabolism. However, a few important discrepancies need to be addressed.

We appreciate the kind words and interest from Reviewer #2. We hope that the discrepancies in our study have been addressed in the revised manuscript.

Major Comments:

My biggest issue relates to the physiological balance of glutathione in their model. The authors show two seemingly contradictory results 1) Glutathione is highly abundant in the tumor interstitial fluid (~30x higher abundance than the plasma) 2) Intratumoral glutathione breakdown provides cysteine to tumor cells. If the tumors are net consumers of glutathione, one would expect a reverse of this concentration gradient, as cancer cell glutathione catabolism depletes glutathione in the tumor, which could be replenished by perfusion from the blood. However, this opposite concentration relationship would suggest that tumor cells are instead net exporters of glutathione, which would mean that the increased glutathione in the TIF is a net cysteine loss for the cancer cells. How do the authors reconcile this discrepancy?

To this end, do GCLC KO tumors have comparably high glutathione in the TIF compared to GCLC WT tumors (which would inform whether the high intratumoral glutathione derives from cancer cells)? Do tumors injected into different tumor sites have different TIF glutathione levels (perhaps suggesting a role for different non-cancer cells to provide glutathione locally in the tumor microenvironment)? One would imagine that GGT levels in each tumor microenvironment could be an important factor here as very high levels would deplete glutathione while very low levels would insufficiently provide glutathione breakdown products to cancer cells. Overall, this seems to be a major missing piece in the study and needs to be addressed.

We thank Reviewer #2 for this feedback and detailed questions. We hypothesized that tumors contributed to increased total GSH levels (tGSH; GSH + 2×GSSG) in the TIF. We found lower levels of tGSH in TIF from orthotopic GCLC KO tumors. This suggests that high intratumoral TIF tGSH levels derive, in part, from cancer cells (**Extended Data Fig. 3d**).

The point regarding different tumor sites having different TIF tGSH levels is important and interesting. We repeated the induction of Gclc KO in tumors using a different tumor site (subcutaneous injection of breast cancer cells into the hind flank of the mouse instead of the mammary gland of the mouse). Similar to breast tumors in the orthotopic location (mammary gland), loss of GSH synthesis in breast tumors in the hind flank did not impair their growth (**Extended Data Fig. 2**). Additionally, we observed lower levels of tGSH in TIF from GCLC KO tumors in the flank, suggesting that high intratumoral TIF tGSH levels derive, in part, from cancer cells (**Extended Data Fig. 3e**). Importantly, we did not observe any differences in tGSH levels in TIF from tumors implanted in the mammary gland (orthotopic) compared to the flank (subcutaneous) (**Extended Data Fig. 3f**). Further, we did not observe differing levels of GGT1 expression between tumors from these distinct locations (**Extended Data Fig. 9i**).

Notably, while our findings suggest that intracellular GSH contributes to extracellular GSH, intracellular GSH levels remain in excess of extracellular GSH, and thus, this process would not be a net cysteine loss for cancer cells. We found that tGSH levels in the TIF are in the range of 0.1 mM-1 mM (**Fig. 1j-1k, Extended Data Fig. 3a-3b and 3d-3f**), whereas GSH levels in the cell are reported in the 1-10 mM range, and in some cases, above 10 mM¹⁶⁻¹⁹ (Emmert S, Nature Chemistry, 2023, PMID: 37322101; Jiang X, ACS Chemical Biology, 2015, PMID: 25531746; Jeong EM, Stem Cell Reports, 2018, PMID: 29307581; Meister A, Journal of Biological Chemistry, 1988, PMID: 3053703). Also, we believe non-cancer cells in the tumor microenvironment could be an additional source of extracellular GSH. We find the immune cells in human and mouse breast tumors express high levels of GCLC (**Extended Data Fig. 1a-1c**), and others have suggested that immune cells efflux GSH²⁰ (Rouzer CA, Journal of Biological Chemistry, 1982, PMID: 6120172). **Fig. 1 and Extended Data Figs. 1, 2, 3, and 9** and their legends below. We have also included a paragraph in the Discussion to state:

“We found that GSH levels were enriched in the TIF compared with serum in mouse models of cancer and in human patients with cancer. While we find that tumors themselves contribute to GSH abundance in TIF, additional cell types most likely also contribute. Interestingly, upon activation, macrophages are known to efflux GSH²⁰.”

Figure 1
Figure 1. Intracellular production of GSH is dispensable for tumor growth. **a**, Schematic of the tumor-specific *Gclc* knockout mouse model. Autochthonous tumors from MMTV-PyMT *Gclc*^{ff} Rosa26-CreERT2 mice were excised and orthotopically transplanted into mammary fat pads of wild-type C57BL/6 mice. C57BL/6 mice were treated with vehicle (corn oil; WT) or 50 mg/kg tamoxifen for 5 days (KO). **b**, Relative *Gclc* mRNA levels of WT and KO tumors. Statistical significance assessed by an unpaired two-tailed t-test ($n = 5$ representative of $n = 3$ independent experiments). **c**, GSH levels of WT and KO tumors. Statistical significance assessed by an unpaired two-tailed t-test ($n = 6$ representative of $n = 3$ independent experiments). **d-e**, Volume over time (**d**) and mass at endpoint (**e**) of WT tumors ($n = 11$) and KO tumors ($n = 12$) (representative of $n = 3$ independent experiments). Statistical significance was assessed by two-way ANOVA followed by Šidák's multiple comparisons test for tumor volume and by an unpaired two-tailed t-test for tumor mass. **f-g**, Immunoblot of *Gclc* protein levels (**f**) and GSH levels (**g**) in HCC-1806 human breast cancer cells infected with lentiCRISPR v2 containing non-targeting guides (sgNTC) and guides against GCLC (sgGclc 1 and sgGclc 2). Statistical significance was assessed by ordinary one-way ANOVA with Tukey's multiple comparisons test. **h-i**, Volume over time (**h**) and mass at endpoint (**i**) of orthotopically implanted HCC-1806 sgNTC, sgGclc 1, and sgGclc 2 breast cancer cells. Statistical significance was assessed by ordinary one-way ANOVA with Tukey's multiple comparisons test for volume and by two-way ANOVA followed by Šidák's multiple comparisons test for tumor mass ($n = 7$ representative of $n = 2$ independent experiments). **j**, Concentration of total GSH (tGSH; GSH + 2×GSSG) in serum ($n = 11$) and tumor interstitial fluid ($n = 4$) from *Gclc* WT MMTV-PyMT autochthonous tumors from (**a**). Statistical significance was assessed by an unpaired two-tailed t-test. **k**, Concentration of tGSH in serum ($n = 8$) and tumor interstitial fluid ($n = 6$) from patients with breast cancer. Statistical significance was

assessed by an unpaired two-tailed t-test. Data represented as mean \pm s.e.m., ns, not significant; *p-value<0.05; **p-value<0.01; ***p-value<0.001; ****p-value<0.0001.

Extended Data Figure 1

Extended Data Figure 1. Gclc is expressed in non-epithelial cells, and its loss in tumors does cause redox or metabolic alterations. **a**, Single-nucleus RNAseq (snRNAseq) of *Gclc* transcripts from the indicated cells present in human mammary tissue. **b-e**, Autochthonous tumors from MMTV-PyMT *Gclc*^{f/f} Rosa26-CreERT2 mice were excised and orthotopically transplanted into mammary fat pads of wild-type C57BL/6 mice, which were treated with vehicle (corn oil; WT) or 50 mg/kg tamoxifen for 5 days (KO). Representative immunohistochemistry (**b**) and immunofluorescence (**c**) staining of indicated proteins in *Gclc* WT and *Gclc* KO mammary tumors. Relative mRNA expression of indicated genes (**d**) and representative immunohistochemistry of indicated proteins (**e**) in *Gclc* WT and *Gclc* KO tumors. Statistical significance was assessed by an unpaired two-tailed t-test in (**d**) (n = 6 representative of n = 3 independent experiments). **f-j**, Volcano plot of metabolic differences (**f**) and levels of GSSG (**g**), ophthalmic acid (**h**), glutamate (**i**), NEM-cysteine (**j**) from *Gclc* WT and *Gclc* KO tumors. Statistical significance was assessed by an unpaired two-tailed t-test in (**g-j**) (n = 6 representative of n = 3 independent

experiments). Data represented as mean \pm s.e.m., *p-value<0.05; **p-value<0.01; ***p-value<0.001; ****p-value<0.0001; ns, not significant.

Extended Data Figure 2

Extended Data Figure 2. Intracellular GSH production is dispensable for tumor growth in the mouse flank. **a-c**, Autochthonous tumors from MMTV-PyMT *Gclc*^{fl/fl} Rosa26-CreERT2 mice were excised and subcutaneously transplanted into flanks of wild-type C57BL/6 mice, which were treated with vehicle (corn oil; WT) or 50 mg/kg tamoxifen for 5 days (KO). Tumor volume over time (WT, n = 7; KO, n = 8 representative of n = 2 independent experiments) **(a)**, total tumor mass (WT, n = 6; KO, n = 7 representative of n = 2 independent experiments) **(b)**, total glutathione levels (WT, n = 4; KO, n = 3 representative of n = 2 independent experiments) **(c)**. Statistical significance was assessed by an unpaired two-tailed t-test. Data represented as mean \pm s.e.m., *p-value<0.05; **p-value<0.01; ***p-value<0.001; ****p-value<0.0001; ns, not significant.

Extended Data Figure 3

Extended Data Figure 3. Total GSH (tGSH) is enriched in the TIF compared to serum across multiple cancer models. **a-c**, Concentration of tGSH and cystine in plasma ($n = 17$) and tumor interstitial fluid ($n = 7$) from the LSL-KrasG12D/+ Trp53f/f Pdx-1-Cre (KP-/-C) mice² bearing pancreatic ductal adenocarcinoma (PDAC) (**a**), in the plasma ($n = 27$) and tumor interstitial fluid ($n = 46$) from renal cell carcinoma (RCC) human patients¹ (**b**), in cell culture media formulations (**c**). **d-i**, Autochthonous tumors from MMTV-PyMT Gclc^{fl/fl} Rosa26-CreERT2 mice were excised and orthotopically transplanted into the mammary gland or subcutaneously transplanted into the

flank of wild-type C57BL/6 mice, which were treated with vehicle (corn oil; WT) or 50 mg/kg tamoxifen for 5 days (KO). tGSH and cystine levels in TIF from orthotopic (WT, n = 4; KO, n = 6) **(d)** or flank (WT, n = 4; KO, n = 3) **(e)** tumors. tGSH in TIF from orthotopic and flank Gclc WT MMTV-PyMT tumors (n = 4) **(f)**. tGSH and cystine levels in serum of mice with orthotopic (WT, n = 11; KO, n = 12) **(g)** or flank (WT, n = 4; KO, n = 5) **(h)** tumors. Serum tGSH and cystine in serum of mice without Gclc WT MMTV-PyMT tumors (n = 8) **(i)**. **j-k**, Serum tGSH **(j)** and cystine **(k)** levels in mice bearing orthotopic (n = 11), flank (n = 4), or no Gclc WT tumors (n = 8). **l-n**, Glutamate levels in plasma and TIF from **Fig. 1k (l)** and from **a-b (m-n)**. Total GSH (tGSH) was calculated as the sum of reduced glutathione (GSH) and twice the amount of oxidized glutathione (GSSG). Statistical significance was assessed by an unpaired two-tailed t-test. Data represented as mean \pm s.e.m., *p-value<0.05; **p-value<0.01; ***p-value<0.001; ****p-value<0.0001; ns, not significant.

Extended Data Figure 9

Extended Data Figure 9. Multiple GGT isoforms are expressed in cancer cell lines, but none score as essential in genome-wide genetic screens. **a**, GGT gene family table showing all 13 GGT-related genes in the human genome. GGT1, 5, 6, and 7 are predicted to have enzymatic activity, but only two (GGT1,5) have been functionally characterized. **b**, mRNA expression (Expression Public 24Q2) of cancer cell lines (n=1,517) from DepMap's Cancer Cell Line Encyclopedia (CCLE). **c**, Protein levels from cancer cell lines from DepMap's CCLE (Proteomics GGT1: P19440 (n=339), GGT5: P36269 (n=108), GGT6: Q6P531 (n=45), GGT7: Q9UJ14 (n=375)). **d**, mRNA and protein levels of GGT isoforms in the cell line HCC1806, using data from **(b)** and **(c)**. **e**, Histogram (splined curves) of gene dependency of cancer cell lines from Dependency Map (DepMap) (Public 24Q2+Score Chronos). A lower score suggests a gene is more likely to be essential in a cell line. A score of 0 indicates a non-essential gene, while -1 reflects the median for essential genes. MTOR and GAPDH are shown as examples of common essential genes. **f**, mRNA expression of GGT enzymes was analyzed in human tumors from 32 different TCGA projects. **g**, GGT1 mRNA expression levels in human tumoral and non-tumoral tissues from the Cancer Genome Atlas Program (TCGA) and the Genotype-Tissue Expression (GTEx) databases. Black lines represent median values. **h**, The mutational landscape and somatic mutation frequency (SMF) of each GGT gene was analyzed in 105,260 cancer patient samples across 226 non-redundant studies using the cBioPortal platform revealing a low mutation frequency (mean = 0.15%). **i**, Autochthonous tumors from MMTV-PyMT Gclc/f Rosa26-CreERT2 mice were excised and orthotopically transplanted into the mammary gland or subcutaneously transplanted into the flank of wild-type C57BL/6 mice, which were treated with vehicle (corn oil; WT) or 50 mg/kg tamoxifen for 5 days (KO). Immunohistochemistry of GGT1 protein expression in Gclc WT and KO orthotopic and flank tumors. Statistical significance in **(g)** was analyzed by an unpaired t-test. **p-value<0.01; ***p-value<0.001; ****p-value<0.0001; ns, not significant.

Figure 5m is an important functional experiment to validate their mechanism, but it is surprisingly devoid of corroborative data. To tie up their mechanism, the authors should conduct metabolomics on serum and tumors from these animals to uncover how NAC treatment affects systemic (NAC, cystine, and glutathione) and intratumoral (cysteine, glutathione, (hypo)taurine) cysteine metabolism.

We appreciate this point and agree with Reviewer #2 that this data was lacking from **Fig. 5**. We have analyzed metabolites in serum and tumors from tumor-bearing mice treated with vehicle, GGsTop, NAC, or both GGsTop and NAC (**Fig. 5o-5q** and **Extended Data Fig. 10j-10o**). We did not detect NAC in the serum but did detect the oxidized form of NAC, N,N'-diacetyl-L-cystine (diNAC)²¹, in the serum of mice treated with NAC (**Extended Data Fig. 10j**). Further, we found increased NAC levels in tumors of mice treated with NAC (**Extended Data Fig. 10m**). We did not observe any differences in cystine or tGSH in the serum of mice treated with NAC and GGsTop compared to mice treated with GGsTop alone (**Extended Data Fig. 10k-10l**). Additionally, there were no differences in cysteine or tGSH in the tumor of mice treated with GGsTop and NAC compared to mice treated with GGsTop (**Fig. 5o** and **Extended Data Fig. 10n**). However, hypotaurine levels were decreased in tumors from mice treated with GGsTop (**Fig. 5l**), and hypotaurine (but not taurine) levels were increased in tumors of mice treated with GGsTop and NAC (**Fig. 5p** and **Extended Data Fig. 10o**), suggesting a rescue in cysteine-related metabolism. Finally, we found that ophthalmic acid, a metabolite synthesized in the absence of cysteine, was increased in tumors treated with GGsTop (**Fig. 5m**), and there was a trending but non-significant decrease in ophthalmic acid in mice treated with NAC and GGsTop (**Fig. 5q**). Overall, these data demonstrate that mice treated with NAC have increased levels of NAC in serum and tumors, and when NAC and GGsTop treatment is combined, cysteine-related metabolism in tumors is rescued. **Fig. 5** and **Extended Data Fig. 10** and their legend are shown below.

Figure 5

Figure 5. GSH catabolism is necessary to support cysteine supply and tumor growth. **a**, Mouse kidney extracts were assayed for GGT activity in the presence of GGT inhibitors ($n = 4$ technical replicates from $n = 3$ independent experiments). **b**, HCC-1806 cells were grown in a control medium with indicated doses of GGTsTop for 4 hours, and GGT activity was determined ($n = 2$ technical replicates representative of $n = 3$ independent experiments). **c-d**, Relative cell number of HCC-1806 cells treated with GGTsTop for 96 hours in control (208 μM cystine) (**c**) or cystine-free/GSH- or CysGly-supplemented (**d**) ($n = 4$ technical replicates representative of $n = 2$ independent experiments). **e**, GGT activity in the kidney extracts of C57BL/6 mice treated intraperitoneally with vehicle ($n=6$ mice) or with 5 mg/kg of GGTsTop every 12 hours for 1 ($n=9$), 2 ($n=9$), or 3 days ($n=8$). **f-g**, Orthotopically implanted HCC-1806 cell xenografts in mice treated intraperitoneally with vehicle (sterile saline) ($n=15$) or 5 mg/kg GGTsTop every 12 hours ($n=18$). Tumor volume over time (**f**) and mass at endpoint (**g**) (from $n = 2$ independent experiments). Statistical significance was assessed by two-way ANOVA followed by Tukey's multiple comparisons test. **h-i**, GGT activity in the kidney (**h**) and tumors (**i**) from mice in (**f**) at the endpoint. Statistical significance was assessed by an unpaired two-tailed t-test (vehicle, $n = 7$; GGTsTop, $n = 9$). **j**, Serum GSH (NEM-GSH) levels from mice in (**f**). Statistical significance was assessed by an unpaired two-tailed t-test (vehicle, $n = 8$; GGTsTop, $n = 7$). **k-m**, Tumor cysteine (NEM-cysteine) (**k**), hypotaurine (**l**), and ophthalmic acid (**m**) levels from tumors in (**f**). Statistical significance in (**k-m**) was assessed by an unpaired two-tailed t-test (vehicle, $n = 9$; GGTsTop, $n = 10$). **n**, Volume of orthotopically implanted HCC-1806 cell xenografts in mice treated with vehicle or 5 mg/kg GGTsTop every 12 hours alone or while being supplemented with n-acetyl-cysteine (NAC, 30 mM) in their drinking water (vehicle, $n = 5$; GGTsTop, $n = 8$; vehicle with NAC, $n = 7$, GGTsTop with NAC, $n = 7$). Statistical significance was assessed by two-way ANOVA (representative of $n = 2$ independent experiments). **o-q**, Tumor cysteine (**o**), hypotaurine (**p**), and ophthalmic acid (**q**) levels from tumors in (**n**). Statistical significance was assessed by an unpaired two-tailed t-test (vehicle, $n = 8$; GGTsTop, $n = 7$; vehicle with NAC, $n = 7$).

= 8, GGSTop with NAC, n = 9 from n = 2 independent experiments). Data represented as mean ± s.e.m., *p-value<0.05; **p-value<0.01; ***p-value<0.001; ****p-value<0.0001; ns, not significant.

Extended Data Figure 10

Extended Data Figure 10. GGsTop efficiently inhibits GGT activity in animals without causing toxicity. **a**, Structure of putative inhibitors of GGT. **b**, GGT activity in untreated C57BL/6 mouse tissues ($n \geq 3$ for each tissue). **c**, mRNA expression of GGT isoforms in 22 adult and 8 embryonic mouse tissues from publicly available datasets²². **d**, GGT activity in the kidney of mice 24 hours after a single i.p. injection of GGsTop at the indicated concentrations ($n = 5$ per group). **e-f**, ALT and AST liver damage serum markers ($n=6$ for Control, $n=9$ for GGsTop) (**e**) and percent change in body weight ($n=10$ per group) (**f**) in mice treated bi-daily with vehicle (saline) or 5 mg/kg i.p. GGsTop for 8 days. **g-i**, Tumor metabolites from mice with orthotopically implanted HCC-1806 cells treated with bi-daily i.p. injections of vehicle or 5 mg/kg GGsTop ($n=9$ for Control, $n=10$ for GGsTop). **j-o**, Serum and tumor metabolites from mice with orthotopically implanted HCC-1806 cells treated with the vehicle, *n*-acetyl-cysteine (30 mM in the drinking water), GGsTop (189 μ M in the drinking water), or NAC and GGsTop (Tumors; $n=8$ for vehicle, $n=7$ for vehicle+NAC, $n=9$ for GGsTop, $n=9$ for GGsTop+NAC. Serum; $n=9$ for vehicle, $n=7$ for vehicle+NAC, $n=6$ for GGsTop, $n=9$ for GGsTop+NAC). **p**, GGsTop aliquots (25 mg/mL in H₂O) were kept at room temperature for the indicated number of days, after which their inhibitory potency was assessed using an *in vitro* activity assay with mouse kidney lysates ($n=3$ technical replicates). **q**, GGT activity in the kidney of mice 72 hours after delivery of GGsTop in the drinking water at the indicated concentrations ($n=7-10$ animals per group). **r-t**, Tumor volume over time (**r**), and GGT activity in the kidney (**s**) and tumor (**t**) from mice with orthotopically implanted HCC-1806 cells treated with the vehicle, GGsTop (189 μ M in the drinking water), buthionine sulfoximine (BSO, 20 mM in the drinking water), or GGsTop and BSO ($n=8-10$ animals per group). Statistical significance in **d,j-o** was assessed by one-way ANOVA with Tukey's multiple comparisons test, while in **e,g-i,s,t** it was assessed by unpaired t-test. Data represented as mean \pm s.e.m., * p -value <0.05 ; ** p -value <0.01 ; *** p -value <0.001 ; **** p -value <0.0001 ; ns, not significant.

While much of the data focuses on glutathione catabolism delivering cysteine to cells, the authors are careful to leave open the possibility that it may also deliver the other constituent parts of glutathione. Can the authors test the relative capability of glutathione to fulfill limitations of glycine (potentially using glycine or serine/glycine deficient media) or glutamate (potentially using a glutaminase inhibitor)?

This is a very interesting point by Reviewer #2. To test this, we cultured cancer cells in serine/glycine-deficient medium (to limit glycine) and in glutamine/glutamate-deficient medium (to limit glutamate) and supplemented with GSH. Unlike in cystine-free medium, GSH failed to rescue cancer cells cultured in conditions that limit glycine or glutamate (**Extended Data Fig. 4b-4d**). Next, we cultured cancer cells in cystine-replete media supplemented with GSH and treated cells with an inhibitor of glutaminase (CB-839). Interestingly, we found that cells supplemented with GSH were less sensitive to CB-839 than cells in control conditions. This suggests that while GSH-derived glycine and glutamate cannot rescue glycine- and glutamate-free culture conditions, GSH catabolism can potentially provide support upon inhibition of amino acid production. **Extended Data Fig. 4** and its legend are shown below.

Extended Data Figure 4

Extended Data Figure 4. Supplementation with GSH or cysteinylglycine rescues cancer cells in cystine-free conditions. **a**, Cancer cell lines were grown in control (208 μ M cystine; Cys₂), cystine-free (Cys₂-free), cystine-free/GSH-supplemented (750 μ M) or cystine-free/CysGly-supplemented (750 μ M) medium, and relative cell numbers were determined ($n = 6-12$ technical replicates from $n = 3-4$ independent experiments). **b-d**, HCC-1806 cells were grown in control media or cystine-free ($n = 8$ technical replicates from $n = 2$ independent experiments) (**b**), serine- and glycine-free ($n = 8$ technical replicates from $n = 2$ independent experiments) (**c**), glutamine- and glutamate-free ($n = 11$ technical replicates from $n = 3$ independent experiments) (**d**) media alone or supplemented with GSH (750 μ M) for 72 hours, and relative cell numbers were determined. Statistical

significance was assessed by one-way ANOVA with Tukey's multiple comparisons test. **e**, HCC-1806 cells were grown in control media alone or supplemented with GSH (750 μ M) and treated with DMSO or CB-839 (5 μ M) for 72 hours, and relative cell numbers were determined. Statistical significance was assessed by two-way ANOVA followed by Šídák's multiple comparisons ($n = 12$ technical replicates). **f**, HCC-1806 cells were untreated or treated with BSO (100 μ M) for 24 hours, and relative GSH levels were determined. Statistical significance was assessed by unpaired two-tailed t-test ($n = 13-17$ from $n = 3$ independent experiments). **g**, HCC-1806 cells were grown in control (208 μ M cystine; Cys₂), cystine-free (Cys₂-free), or cystine-free/GSH-supplemented (750 μ M) medium, untreated or treated with BSO (100 μ M) for 72 hours, and relative cell numbers were determined. Statistical significance was assessed by two-way ANOVA followed by Šídák's multiple comparisons ($n = 6$ technical replicates). **h**, HCC-1806 cells were grown in control (208 μ M cystine; Cys₂), cystine-free (Cys₂-free), cystine-free/L-NAC-supplemented (400 μ M), cystine-free/D-NAC-supplemented (400 μ M) medium, untreated or treated with BSO (100 μ M) for 72 hours, and relative cell numbers were determined. Statistical significance was assessed by two-way ANOVA followed by Šídák's multiple comparisons ($n = 4$ technical replicates).

Minor Comments:

Line 77 "after its release..." The peptidase step is somewhat mysterious throughout the paper. Can the authors elaborate on what is known about these peptidase enzymes? Perhaps some citations would be useful here.

We greatly appreciate Reviewer #2's suggestion. We have included a discussion of peptidase enzymes potentially involved in cysteinylglycine breakdown, following its release from GSH. In the Discussion, it now states:

"Following its release from GSH, cysteinylglycine must be further broken down into the individual amino acids cysteine and glycine. Dipeptidases are predicted to control the breakdown of dipeptides, including cysteinylglycine; however, the exact enzymes involved are poorly understood. Carnosine dipeptidase II (CNDP2) has been described to support the catabolism not only of cysteinylglycine²³ but other dipeptides, such as glutamine-containing dipeptides⁴. Future studies involving a systematic analysis of dipeptidase-dipeptide relationships, in malignant and non-malignant tissues, are required."

Referee #3 (Remarks to the Author):

Hecht et al. discover that GSH catabolism promotes tumor growth in a gamma- glutamyltransferase dependent manner, thus providing novel evidence that extracellular glutathione in the tumor microenvironment is capable of acting as an amino acid reservoir. To test this, the authors use the Rosa26-CreERT2 Gclc/f mouse they generated and implanted formed tumors into C57BL/6 mice, in which they activated Cre-recombinase. The authors show GGT activity is necessary and sufficient to promote cancer cell survival, thus highlighting a new actionable pathway of nutrient acquisition in cancer. They show that treatment with GGSTop, a GGT inhibitor, in vivo, increased serum GSH and impaired tumor growth that was rescued by NAC (a cysteine donor).

The key novel insight from this work is that it highlights a potential reason why targeting GSH via Gclc to induce ferroptosis may be challenging in vivo. This work also offers evidence to support the importance of distinguishing between in vitro and in vivo contexts, where metabolite availability significantly differs, as related to targeting of ferroptosis. However, the use of multiple model systems and cancer types, without consistently presenting data across all models obscures the cohesiveness of the proposed mechanism. Critically, while the in vitro mechanism clearly demonstrates that GGT activity is sufficient to support GSH catabolism and promote the survival of surrounding cells under cystine-depleted conditions, its relevance to the in vivo findings shown in Fig. 1 remains uncertain. Bridging this gap will require further experiments to clearly elucidate the extent to which this mechanism operates in vivo. Specifically, several alternative possibilities (described below) need to be experimentally investigated to clarify the contribution of the observed extracellular GSH catabolism phenotype to tumor growth in vivo and it will be important to understand the extent to which this mechanism occurs in the setting of Gclc inhibition (in Gclc-deficient tumor cells, or with BSO treatments).

Major comments

1. The provided rationale for exploring that extracellular GSH may be critical to tumor growth comes from in vivo experiments demonstrating Gclc deletion in tumors failed to perturb tumor growth. However, given that in Fig 1c

there are significant decreases in tumor GSH, and yet no differences in tumor growth does this not suggest that GSH is not required for these tumor cells to survive, and that already in the absence of de novo GSH synthesis the cancer cells have adequate levels of upstream metabolites (i.e., cysteine) to sustain growth? Please clarify this rationale based on the data shown in Fig. 1.

We appreciate this point by Reviewer #3 and completely agree. We hypothesize that, in the presence or absence of de novo GSH synthesis, cancer cells have adequate levels of upstream metabolites (i.e., cysteine) to sustain growth. In the Results, we now state:

“Overall, these findings suggest that while intracellular GSH does not impact tumor growth and survival, extracellular GSH is highly abundant and potentially supports tumors. Further, this suggests that in the presence or absence of GSH synthesis, tumors have an adequate supply of upstream metabolites (i.e., cysteine) to sustain growth.”

2. Are there phenotypic, metabolic, or redox features (are basal ROS levels different?) in the Gclc^{-/-} versus WT tumors that the authors can demonstrate that would indicate extracellular GSH catabolism would confer an increased tumor survival benefit, thus strengthening the premise of their hypothesis?

This is a very good point by Reviewer #3. We have examined this and found that compared to Gclc WT tumors, Gclc KO tumors have little or no increased markers of oxidative stress, including the expression of NRF2 target genes (Gclm, Hmox1, Nqo1, Slc7a11), NRF2 protein, or oxidized DNA damage marker 8-oxoguanine (8-oxoG) (**Extended Data Fig. 1d-1e**). Upon profiling metabolites (**Extended Data Fig. 1f-1j**), we found that Gclc KO tumors had lower levels of GSH, oxidized GSH (GSSG), and ophthalmic acid, which is synthesized by GCLC but condenses glutamate and α -aminobutyric acid rather than glutamate and cysteine²⁴(Schomakers BV, The FEBS Journal, 2024, PMID: 38245827). Notably, aside from these GSH-related metabolites, very few metabolic differences were found in Gclc KO tumors compared to WT. Overall, these data would suggest that extracellular GSH catabolism is sufficient to prevent phenotypic, metabolic, or redox-related features in tumors. **Extended Data Fig. 1** and its legend are shown below.

Extended Data Figure 1

Extended Data Figure 1. Gclc is expressed in non-epithelial cells, and its loss in tumors does cause redox or metabolic alterations. **a**, Single-nucleus RNAseq (snRNAseq) of *Gclc* transcripts from the indicated cells present in human mammary tissue. **b-e**, Autochthonous tumors from MMTV-PyMT *Gclc*^{fl/fl} *Rosa26-CreERT2* mice were excised and orthotopically transplanted into mammary fat pads of wild-type C57BL/6 mice, which were treated with vehicle (corn oil; WT) or 50 mg/kg tamoxifen for 5 days (KO). Representative immunohistochemistry (**b**) and immunofluorescence (**c**) staining of indicated proteins in *Gclc* WT and *Gclc* KO mammary tumors. Relative mRNA expression of indicated genes (**d**) and representative immunohistochemistry of indicated proteins (**e**) in *Gclc* WT and *Gclc* KO tumors. Statistical significance was assessed by an unpaired two-tailed t-test in (**d**) ($n = 6$ representative of $n = 3$ independent experiments). **f-j**, Volcano plot of metabolic differences (**f**) and levels of GSSG (**g**), ophthalmic acid (**h**), glutamate (**i**), NEM-cysteine (**j**) from *Gclc* WT and *Gclc* KO tumors. Statistical significance was assessed by an unpaired two-tailed t-test in (**g-j**) ($n = 6$ representative of $n = 3$ independent experiments). Data represented as mean \pm s.e.m., * p -value <0.05 ; ** p -value <0.01 ; *** p -value <0.001 ; **** p -value <0.0001 ; ns, not significant.

3. What are the intracellular cysteine levels of the tumors shown in Fig. 1c? It is possible cysteine levels in the Gclc^{-/-} tumors are higher because cysteine is not being synthesized into GSH, which would thus decrease the cellular requirement for extracellular cysteine import.

This is a very interesting point. As mentioned above, we did not observe an accumulation of cysteine in MMTV-PyMT Gclc KO tumors compared to Gclc WT tumors (**Extended Data Fig. 1f**). However, analysis HCC-1806 Gclc WT and Gclc KO breast cancer cells *in vitro* demonstrated that cysteine accumulates in Gclc KO cells compared to Gclc WT cells (**Extended Data Fig. 7e**). The discrepancy between the *in vivo* and *in vitro* data could be attributed to HCC-1806 Gclc WT and Gclc KO breast cancer cells cultured *in vitro* in high cysteine level (208 μ M; **Extended Data Fig. 3c**) and these levels are much lower in the TIF of MMTV-PyMT Gclc WT and KO tumors (less than 10 μ M; **Fig. 2c**). The pertinent panels of **Extended Data Fig. 1f**, **Extended Data Fig. 3c**, **Extended Data Fig. 7e**, and **Fig. 2** are shown below.

Extended Data Figure 1. Gclc is expressed in non-epithelial cells, and its loss in tumors does cause redox or metabolic alterations. **a**, Single-nucleus RNAseq (snRNAseq) of Gclc transcripts from the indicated cells present in human mammary tissue. **b-e**, Autochthonous tumors from MMTV-PyMT Gclc^{fl/fl} Rosa26-CreERT2 mice were excised and orthotopically transplanted into mammary fat pads of wild-type C57BL/6 mice, which were treated with vehicle (corn oil; WT) or 50 mg/kg tamoxifen for 5 days (KO). Representative immunohistochemistry (**b**) and immunofluorescence (**c**) staining of indicated proteins in Gclc WT and Gclc KO mammary tumors. Relative mRNA expression of indicated genes (**d**) and representative immunohistochemistry of indicated proteins (**e**) in Gclc WT and Gclc KO tumors. Statistical significance was assessed by an unpaired two-tailed t-test in (**d**) (n = 6 representative of n = 3 independent experiments). **f-j**, Volcano plot of metabolic differences (**f**) and levels of GSSG (**g**), ophthalmic acid (**h**), glutamate (**i**), NEM-cysteine (**j**) from Gclc WT and Gclc KO tumors. Statistical significance was assessed by an unpaired two-tailed t-test in (**g-j**) (n = 6 representative of n = 3 independent experiments). Data represented as mean \pm s.e.m., *p-value<0.05; **p-value<0.01; ***p-value<0.001; ****p-value<0.0001; ns, not significant.

Extended Data Figure 3. Total GSH (tGSH) is enriched in the TIF compared to serum across multiple cancer models. **a-c**, Concentration of tGSH and cystine in plasma (n = 17) and tumor interstitial fluid (n = 7) from the LSL-KrasG12D/+ Trp53f/f Pdx-1-Cre (KP-/-C) mice² bearing pancreatic ductal adenocarcinoma (PDAC) (**a**), in the plasma (n = 27) and tumor interstitial fluid (n = 46) from renal cell carcinoma (RCC) human patients¹ (**b**), in cell culture media formulations (**c**). **d-i**, Autochthonous tumors from MMTV-PyMT Gclc^{fl/fl} Rosa26-CreERT2 mice were excised and orthotopically transplanted into the mammary gland or subcutaneously transplanted into the flank of wild-type C57BL/6 mice, which were treated with vehicle (corn oil; WT) or 50 mg/kg tamoxifen for 5 days (KO). tGSH and cystine levels in TIF from orthotopic (WT, n = 4; KO, n = 6) (**d**) or flank (WT, n = 4; KO, n = 3) (**e**) tumors. tGSH in TIF from orthotopic and flank Gclc WT MMTV-PyMT tumors (n = 4) (**f**). tGSH and cystine levels in serum of mice with orthotopic (WT, n = 11; KO, n = 12) (**g**) or flank (WT, n = 4; KO, n = 5) (**h**) tumors. Serum tGSH and cystine in serum of mice without Gclc WT MMTV-PyMT tumors (n = 8) (**i**). **j-k**, Serum tGSH (**j**) and cystine (**k**) levels in mice bearing orthotopic (n = 11), flank (n = 4), or no Gclc WT tumors (n = 8). **l-n**, Glutamate levels in plasma and TIF from **Fig. 1k** (**l**) and from **a-b** (**m-n**). Total GSH (tGSH) was calculated as the sum of reduced glutathione (GSH) and twice the amount of oxidized glutathione (GSSG). Statistical significance was assessed by an unpaired two-tailed t-test. Data represented as mean ± s.e.m., *p-value<0.05; **p-value<0.01; ***p-value<0.001; ****p-value<0.0001; ns, not significant.

Extended Data Figure 7. Blocking the export of intracellular GSH does not impair the ability of GSH to rescue in cystine-free conditions. **a**, Dose-response curves showing relative cell number after 72 hours of treatment of HCC-1806 cells with vincristine (0-5 nM) in the presence of the MRP1 inhibitor MK-571 (0-15 μM) (n = 8 technical replicates). **b**, Heatmap of absolute cell numbers of data shown in (**a**). **c**, Drug synergy analysis of data shown in (**a**) using the ZIP model. **d**, Relative cell numbers after 72 hours of treatment of HCC-1806 cells without or with MK-571 (15 μM) in control (208 μM cystine; Cys₂) or cystine-free/GSH-supplemented (750 μM) medium. Statistical significance assessed by two-way ANOVA followed by uncorrected Fisher's Least Significant Difference (LSD) test (n = 3 technical replicates). **e-h**, HCC-1806 cells were grown without or with MK-571 (15 μ) in control (208 μM cystine; Cys₂) or cystine-free/GSH-supplemented (750 μM) medium for 48 hours, and the relative abundance of indicated intracellular (**e-g**) and extracellular (**f-g**) metabolites was determined. Δ Abundance represents the metabolite level difference between conditioned and virgin media. Statistical significance was determined by two-way ANOVA using uncorrected Fisher's LSD test (n = 3 technical replicates). Data represented as mean ± s.e.m., *p-value<0.05; **p-value<0.01; ***p-value<0.001; ****p-value<0.0001; ns, not significant.

Figure 2. Extracellular GSH supplies amino acids to promote cancer cell growth and survival in cystine-free environments. **a**, Schematic of the different mechanisms of cysteine acquisition and utilization. **b**, Concentration of cystine in serum (n = 11) and tumor interstitial fluid (n=4) from Gclc WT MMTV-PyMT autochthonous tumors from **Fig. 1a**. Statistical significance was assessed by an unpaired two-tailed t-test. **c**, Concentration of cystine in serum (n = 8) and tumor interstitial fluid (n = 6) from patients with breast cancer. Statistical significance was assessed by an unpaired two-tailed t-test. **d-f**, HCC-1806 breast cancer cells were grown in control (208 μM cystine; Cys₂), cystine-free (Cys₂-free), cystine-free/GSH-supplemented (750 μM) or cystine-free/CysGly-supplemented (750 μM) medium, and cell numbers (**d**), percentages of proliferative (BrdU⁺) (**e**) and apoptotic (Annexin V⁺) (**f**) cells were determined at the indicated timepoints. Statistical significance was analyzed by two-way ANOVA followed by Tukey's multiple comparisons test (n = 3 technical replicates representative of n = 4 independent experiments) for (**d**), two-way ANOVA followed by Šidák's multiple comparisons test (n = 6 technical replicates from n = 3 independent experiments for **e**; n = 9 technical replicates from n = 3 independent experiments for **f**). **g**, Levels of extracellular CysGly in media in HCC-1806 breast cancer cells grown in indicated media and at 72 hours. One-way ANOVA followed by Tukey's multiple comparisons test (n = 3 technical replicates). **h**, Schematic of ¹³C-cystine stable-isotope labeling approach and metabolomics. **i**, Percent labeling of ¹³C-cystine stable-isotope in indicated species at 72 hours (n = 5 technical replicates). **j**, Schematic of ¹⁵N¹³C-cystine stable-isotope labeling approach and proteomics. **k**, Percent labeling of ¹⁵N¹³C-cystine stable-isotope in indicated species at 96 hours (average of n = 3 technical replicates). Data is representative as mean ± s.e.m., *p-value<0.05; **p-value<0.01; ***p-value<0.001; ****p-value<0.0001; ns, not significant.

4. In Figure 1f, the authors move into a 4T1 orthotopic model, and while this reviewer greatly appreciates the use of multiple breast cancer models as well as the PDAC model, it creates difficulty in data interpretation as not all of the data is shown for the different lines. Generating the GSH measurements in plasma and TIF from the mice in Fig. 1a-e would be a way to connect these two independent datasets in the same model as it is unclear

if the observed phenotypes in Fig. 1f can also be applied to the data shown in in Fig 1a-e. The difficulty comparing data in the different models also occurs later in the manuscript as the authors shift the HCC1806 and the phenotypes shown in Figure 1 are uncertain to be occurring in this model.

We greatly appreciate this point by Reviewer #3, and we have made multiple changes to the manuscript to increase clarity and continuity between Figures.

First, we have analyzed total GSH levels (tGSH; GSH + 2×GSSG) in serum and TIF from the MMTV-PyMT tumors in Fig. 1a-1e (now **Fig. 1a-1c**). Similar to the 4T1 orthotopic tumor model and the PDAC model, we find that tGSH levels are increased in TIF from MMTV-PyMT tumors compared to serum (**Fig. 1j**). To extend these studies, we also analyzed tGSH levels in TIF and plasma from patients with breast and renal cell cancer. Similar to the murine tumor models, we find tGSH levels are increased in TIF from human tumors compared to serum (**Fig. 1k and Extended Data Fig. 3b**).

Next, we have generated HCC-1806 Gclc WT and Gclc KO breast cancer cell lines (**Fig. 1f-1g**) and tested their ability to grow in vivo as orthotopic tumors. Similar to MMTV-PyMT Gclc WT and Gclc KO tumors, we find that HCC-1806 Gclc KO tumors grow at similar rates as Gclc WT tumors (**Fig. 1h-1i**).

Finally, we agree with Reviewer #3 that the use of two different immunocompetent mouse strains could introduce phenotypic differences (C57BL/6 mouse strain for MMTV-PyMT breast tumors and PDAC tumors and Balb/c mouse strain for 4T1 orthotopic tumor model). To add clarity, we have decided to remove the TIF and serum data with the 4T1 orthotopic tumors in Balb/c mice and focus on one mouse strain (C57/BL6) for murine immunocompetent tumor experiments with MMTV-PyMT breast tumors (**Fig. 1j**) and PDAC tumors (**Extended Data Fig. 3a**). **Fig. 1 and Extended Data Fig. 3** and their legends are shown below.

Figure 1
Figure 1. Intracellular production of GSH is dispensable for tumor growth. **a**, Schematic of the tumor-specific *Gclc* knockout mouse model. Autochthonous tumors from MMTV-PyMT *Gclc^{fl/fl}* Rosa26-CreERT2 mice were excised and orthotopically transplanted into mammary fat pads of wild-type C57BL/6 mice. C57BL/6 mice were treated with vehicle (corn oil; WT) or 50 mg/kg tamoxifen for 5 days (KO). **b**, Relative *Gclc* mRNA levels of WT and KO tumors. Statistical significance assessed by an unpaired two-tailed t-test ($n = 5$ representative of $n = 3$ independent experiments). **c**, GSH levels of WT and KO tumors. Statistical significance assessed by an unpaired two-tailed t-test ($n = 6$ representative of $n = 3$ independent experiments). **d-e**, Volume over time (**d**) and mass at endpoint (**e**) of WT tumors ($n = 11$) and KO tumors ($n = 12$) (representative of $n = 3$ independent experiments). Statistical significance was assessed by two-way ANOVA followed by Šidák's multiple comparisons test for tumor volume and by an unpaired two-tailed t-test for tumor mass. **f-g**, Immunoblot of *Gclc* protein levels (**f**) and GSH levels (**g**) in HCC-1806 human breast cancer cells infected with lentiCRISPR v2 containing non-targeting guides (sgNTC) and guides against GCLC (sgGclc 1 and sgGclc 2). Statistical significance was assessed by ordinary one-way ANOVA with Tukey's multiple comparisons test. **h-i**, Volume over time (**h**) and mass at endpoint (**i**) of orthotopically implanted HCC-1806 sgNTC, sgGclc 1, and sgGclc 2 breast cancer cells. Statistical significance was assessed by ordinary one-way ANOVA with Tukey's multiple comparisons test for volume and by two-way ANOVA followed by Šidák's multiple comparisons test for tumor mass ($n = 7$ representative of $n = 2$ independent experiments). **j**, Concentration of total GSH (tGSH; GSH + 2×GSSG) in serum ($n = 11$) and tumor interstitial fluid ($n = 4$) from *Gclc* WT MMTV-PyMT autochthonous tumors from (a). Statistical significance was assessed by an unpaired two-tailed t-test. **k**, Concentration of tGSH in serum ($n = 8$) and tumor interstitial fluid ($n = 6$) from patients with breast cancer. Statistical significance was assessed by an unpaired two-tailed t-test. Data represented as mean \pm s.e.m., ns, not significant; * p -value <0.05 ; ** p -value <0.01 ; *** p -value <0.001 ; **** p -value <0.0001 .

Extended Data Figure 3

Extended Data Figure 3. Total GSH (tGSH) is enriched in the TIF compared to serum across multiple cancer models. **a-c**, Concentration of tGSH and cystine in plasma (n = 17) and tumor interstitial fluid (n = 7) from the LSL-KrasG12D/+ Trp53f/f Pdx-1-Cre (KP-/-C) mice² bearing pancreatic ductal adenocarcinoma (PDAC) (**a**), in the plasma (n = 27) and tumor interstitial fluid (n = 46) from renal cell carcinoma (RCC) human patients¹ (**b**), in cell culture media formulations (**c**). **d-i**, Autochthonous tumors from MMTV-PyMT Gclc^{fl/fl} Rosa26-CreERT2 mice were excised and orthotopically transplanted into the mammary gland or subcutaneously transplanted into the flank of wild-type C57BL/6 mice, which were treated with vehicle (corn oil; WT) or 50 mg/kg tamoxifen for 5 days (KO). tGSH and cystine levels in TIF from orthotopic (WT, n = 4; KO, n = 6) (**d**) or flank (WT, n = 4; KO, n = 3) (**e**) tumors. tGSH in TIF from orthotopic and flank Gclc WT MMTV-PyMT tumors (n = 4) (**f**). tGSH and cystine levels in serum of mice with orthotopic (WT, n = 11; KO, n = 12) (**g**) or flank (WT, n = 4; KO, n = 5) (**h**) tumors. Serum tGSH and cystine in serum of mice without Gclc WT MMTV-PyMT tumors (n = 8) (**i**). **j-k**, Serum tGSH (**j**) and cystine (**k**) levels in mice bearing orthotopic (n = 11), flank (n = 4), or no Gclc WT tumors (n = 8). **l-n**, Glutamate levels in plasma and TIF from **Fig. 1k** (**l**) and from **a-b** (**m-n**). Total GSH (tGSH) was calculated as the sum of reduced glutathione (GSH) and twice the amount of oxidized glutathione (GSSG). Statistical significance was assessed by an unpaired two-tailed t-test. Data represented as mean ± s.e.m., *p-value<0.05; **p-value<0.01; ***p-value<0.001; ****p-value<0.0001; ns, not significant.

5. Another interpretation of the data shown in Fig. 1f is that the tumor cells are exporting glutathione (such as via MRPs/ABC transporters or GSH solute exchangers), thus increasing its presence in TIF. Please provide experiments to account for this possibility, as this could change the interpretation of the data. Although it may not mimic the in vivo conditions, demonstrating in vitro in WT or Gclc-/- cells and conducting intracellular and extracellular GSH, GSSG, GSH/GSSG, cysteine, cystine, glutamate +/- exogenous GSH and +/- export transporter inhibition may be one way to provide insight into this possibility and would be sufficient exploration of this alternate explanation.

This is a great suggestion, as drug exporters can facilitate exports of GSH²⁵ (Cao JY, Cell Reports, 2019, PMID: 30726737). Further, the export of certain chemotherapies, such as vincristine, depends on GSH²⁶ (Loe DW, Journal of Biological Chemistry, 1996, PMID: 8621643). To investigate this, we used MK-571, an inhibitor of MRPs²⁷. First, we determined the concentration of MK-571 that blocks MRP activity in cancer cells by performing a double-dose curve with increasing doses of MK-571 and the chemotherapy vincristine. We found that MK-571 at 15 μM provided the most sensitization of cancer cells to vincristine (**Extended Data Fig. 7a-7c**). Next, we tested whether drug export inhibition impacted rescue upon cystine-free/GSH-supplemented conditions. We found that in the presence of drug export inhibition by MK-571, GSH could still rescue cancer cells in cystine-depleted conditions (**Extended Data Fig. 7d**). Overall, this suggests that, in vitro, export of GSH by MRPs does not limit rescue of GSH in cystine-free conditions.

Next, we performed metabolomic analysis of Gclc WT and Gclc KO cells, in cystine-replete conditions or cystine-free/GSH-supplemented conditions, alone or in combination with drug export inhibition by MK-571 (**Extended Data Fig. 7e-7h**). For clarity, we separated out the graphs to show intracellular metabolites in cystine-replete conditions (**Extended Data Fig. 7e**) and cystine-free/GSH-supplemented conditions (**Extended Data Fig. 7f**), followed by extracellular metabolites in cystine-replete conditions (**Extended Data Fig. 7g**) and cystine-free/GSH-supplemented conditions (**Extended Data Fig. 7h**). Notably, for intracellular metabolites, relative abundances were normalized to the cystine-replete alone condition (the first bar in the graphs in **Extended Data Fig. 7e**). For extracellular metabolites, relative abundances were normalized to virgin media that had not been exposed to cells.

First, we found that inhibition of MRPs by MK-571 in cystine-replete conditions resulted in an accumulation of intracellular GSH and GSSG in Gclc WT cancer cells (**Extended Data Fig. 7e**). This result is in accordance with previous findings that MRP1 loss prevents GSH export²⁵. We did not observe an accumulation of intracellular GSH in Gclc KO cancer cells in cystine-replete conditions with MK-571 (**Extended Data Fig. 7e**); however, this is most likely because intracellular GSH synthesis was impaired in Gclc KO cells. Additionally, we found, as expected, Gclc KO cells to have higher levels of the precursors cysteine and glutamate than Gclc WT cells.

Next, we found that MRP inhibition by MK-571 in Gclc WT cells under cystine-free/GSH-supplemented conditions did not affect metabolite levels, aside from an increase in intracellular glutamate, the mechanism of which was unclear (**Extended Data Fig. 7f**). For Gclc KO cells, we found GSSG levels to be elevated compared to Gclc

WT, potentially caused by the lack of de novo GSH synthesis and reliance on regeneration of GSH from GSSG. Additionally, drug export inhibition by MK-571 in Gclc KO cells lowered GSSG and glutamate levels, a mechanism that was also unclear.

Finally, we examined extracellular metabolites in Gclc WT and Gclc KO cells in cystine-replete conditions or cystine-free/GSH-supplemented conditions, alone or in combination with MK-571. In cystine-replete conditions, we found that MRP inhibition by MK-571 did not impact extracellular GSH, GSSG, or cystine levels but did lower extracellular glutamate levels for Gclc KO cells, the mechanism of which was unclear (**Extended Data Fig. 7g**). In cystine-free/GSH-supplemented conditions, MRP1/4 inhibition by MK-571 lowered extracellular GSH, cystine, and glutamate levels, but only in Gclc KO cells (**Extended Data Fig. 7h**). This data could suggest that, upon limiting GSH synthesis, export of the remaining GSH by MRPs is required to sustain levels of extracellular GSH and its catabolic products. **Extended Data Fig. 7** and its legend are shown below.

Extended Data Figure 7

Extended Data Figure 7. Blocking the export of intracellular GSH does not impair the ability of GSH to rescue in cystine-free conditions. **a**, Dose-response curves showing relative cell number after 72 hours of treatment of HCC-1806 cells with vincristine (0-5 nM) in the presence of the MRP1 inhibitor MK-571 (0-15 μ M) (n = 8 technical replicates). **b**, Heatmap of absolute cell numbers of data shown in **(a)**. **c**, Drug synergy analysis of data shown in **(a)** using the ZIP model. **d**, Relative cell numbers after 72 hours of treatment of HCC-1806 cells without or with MK-571 (15 μ M) in control (208 μ M cystine; Cys₂) or cystine-free/GSH-supplemented (750 μ M) medium. Statistical significance assessed by two-way ANOVA followed by uncorrected Fisher's Least Significant Difference (LSD) test (n = 3 technical replicates). **e-h**, HCC-1806 cells were grown without or with MK-571 (15 μ) in control (208 μ M cystine; Cys₂) or cystine-free/GSH-supplemented (750 μ M) medium for 48 hours, and the relative abundance of indicated intracellular (**e-g**) and extracellular (**f-g**) metabolites was determined. Δ Abundance represents the metabolite level difference between conditioned and virgin media. Statistical significance was determined by two-way ANOVA using uncorrected Fisher's LSD test (n = 3 technical replicates). Data represented as mean \pm s.e.m., *p-value<0.05; **p-value<0.01; ***p-value<0.001; ****p-value<0.0001; ns, not significant.

6. Please include non-tumor bearing controls in Fig. 1f for the plasma measurements (this reviewer appreciates that fluid from non-tumor bearing mammary fat pad conditions is not feasible). Plasma levels of mice bearing 4T1 WT and 4T1 Gclc- deficient tumors are also needed to connect the data in Fig 1-e and Fig 1f between the different models, as it could change the data interpretation if TIF from 4T1 Gclc- deficient tumors contains the same GSH, cystine, and glutamate levels as TIF from WT tumors.

We appreciate the Reviewer's suggestion (which was also suggested by Reviewer #1). We analyzed the serum of C57/BL6 mice not harboring tumors and found no difference between tGSH and cystine levels in the serum (**Extended Data Fig. 3j**). Interestingly, we found a decrease in serum tGSH (but not serum cystine) levels between mice with tumors compared with mice without tumors (**Extended Data Fig. 3k-3l**). Previous studies have found that serum GSH levels are lower in patients with colorectal cancer than in healthy volunteers⁶ (Baltruskeviciene E, Tumori Journal, 2018, PMID: 28777429). This suggests that tumors potentially contribute to GSH breakdown and lower levels of GSH in the serum. **Extended Data Fig. 3** and its legend are shown below.

As mentioned above in Point 4, to provide clarity, we have removed data involving 4T1 orthotopic tumors in Balb/c mice and only focus on the C57/BL6 mouse strain for immunocompetent murine tumor experiments with the MMTV-PyMT breast tumor model and PDAC model.

Extended Data Figure 3

Extended Data Figure 3. Total GSH (tGSH) is enriched in the TIF compared to serum across multiple cancer models. **a-c**, Concentration of tGSH and cystine in plasma ($n = 17$) and tumor interstitial fluid ($n = 7$) from the LSL-KrasG12D/+ Trp53f/f Pdx-1-Cre (KP-/-C) mice² bearing pancreatic ductal adenocarcinoma (PDAC) (**a**), in the plasma ($n = 27$) and tumor interstitial fluid ($n = 46$) from renal cell carcinoma (RCC) human patients¹ (**b**), in cell culture media formulations (**c**). **d-i**, Autochthonous tumors from MMTV-PyMT Gclc^{fl/fl} Rosa26-CreERT2 mice

were excised and orthotopically transplanted into the mammary gland or subcutaneously transplanted into the flank of wild-type C57BL/6 mice, which were treated with vehicle (corn oil; WT) or 50 mg/kg tamoxifen for 5 days (KO). tGSH and cystine levels in TIF from orthotopic (WT, n = 4; KO, n = 6) (d) or flank (WT, n = 4; KO, n = 3) (e) tumors. tGSH in TIF from orthotopic and flank Gclc WT MMTV-PyMT tumors (n = 4) (f). tGSH and cystine levels in serum of mice with orthotopic (WT, n = 11; KO, n = 12) (g) or flank (WT, n = 4; KO, n = 5) (h) tumors. Serum tGSH and cystine in serum of mice without Gclc WT MMTV-PyMT tumors (n = 8) (i). j-k, Serum tGSH (j) and cystine (k) levels in mice bearing orthotopic (n = 11), flank (n = 4), or no Gclc WT tumors (n = 8). l-n, Glutamate levels in plasma and TIF from Fig. 1k (l) and from a-b (m-n). Total GSH (tGSH) was calculated as the sum of reduced glutathione (GSH) and twice the amount of oxidized glutathione (GSSG). Statistical significance was assessed by an unpaired two-tailed t-test. Data represented as mean \pm s.e.m., *p-value<0.05; **p-value<0.01; ***p-value<0.001; ****p-value<0.0001; ns, not significant.

7. Different mouse strains are known to have significantly different GSH and GSH/GSSG profiles (PMID: 24613380) and this should be taken into consideration when comparing data from different mouse strains (such as C57BL/6 and Balb/c). In Fig. 1f, the interpretation of the data with respect to these different strains would benefit from the inclusion of GSH measurements of plasma from all strains used in this manuscript.

As mentioned above in Point 4, to provide clarity, we have removed data involving 4T1 orthotopic tumors in Balb/c mice and only focus on the C57/BL6 mouse strain for murine immunocompetent tumor experiments with the MMTV-PyMT breast tumor model and PDAC model.

However, this is a very important point, and we have included the Reviewer's point in the Discussion, which now states:

“Notably, GSH levels and GSH/GSSG ratios can vary across mouse strains²⁸, and thus could also impact cysteine supply to tissues (and tumors) in these mice.”

8. The authors show that extracellular GSH can be used as a cysteine source in vitro, but have not yet demonstrated if extracellular GSH can be used as a cysteine source in vivo. In vivo tracing experiments, such as tracing extracellular cystine, such as using a ¹³C₆-cystine infusion, and tracking glutathione labeling within the tumors in vivo in WT compared to Gclc^{-/-} tumors may be one way to show the extent to which tumor cells rely on cysteine uptake in vivo and strengthen the connections between the mechanism observed in vitro and the in vivo phenotype.

This is a very good point and suggestion by Reviewer #3. Recently, a co-author of our manuscript (Dr. Gina DeNicola) published a comprehensive study of cysteine utilization in normal and malignant tissues²⁹ (Yoon SJ, Cancer Research, 2023, PMID: 36862034). Here, the authors performed in vivo tracing experiments using ¹³C₆-cystine infusion to trace extracellular cystine and track GSH labeling in tumors in vivo. First, they show that circulating cystine pools are fully labeled (Figure 3B), suggesting complete incorporation of ¹³C₆-cystine into extracellular cystine. Next, they show that, in hepatocellular carcinoma (HCC), pancreatic ductal adenocarcinoma (PDAC), and lung adenocarcinoma (LUAD), a significant fraction of cysteine in GSH comes from unlabeled cystine (Figure 6A). This is especially clear in HCC tumors, where only a small fraction of cysteine in GSH is labeled by extracellular cystine. In the Discussion, they state, “Interestingly, we find that despite an increase in the total cysteine pool in HCC, this cannot be accounted for by an increase in transsulfuration or cystine uptake, suggesting that HCC tumors have an alternative source of cysteine. Glutathione degradation via gamma-glutamyl transpeptidase may locally generate available cysteine (49), or tumors may recycle micropinocytosis-derived protein to contribute to the cysteine pool as has been shown in HCC cell lines (50). Additional work is needed to understand the reliance of HCC tumors on other cysteine sources.” Overall, this data suggests that GSH-derived cysteine accounts for a significant fraction of cysteine in GSH. Figure 3, Figure 6, and the Figure Legends from Yoon SJ et al. are below. Also, in the Discussion, we have added:

“Interestingly, previous studies measuring the incorporation of cysteine into metabolites suggested that additional sources of cysteine for tumors, beyond extracellular cystine or de novo synthesized cysteine, most likely exist²⁹.”

Figure 3. Cyst(e)ine supplies the cysteine pool in all tissues. **A**, Schematic depicting $^{13}\text{C}_6$ -cystine infusion and its metabolism to glutathione and taurine. **B-G**, Healthy C57BL/6J mice were infused with $^{13}\text{C}_6$ -cystine, followed by analysis of the fraction labeling in cysteine (**B**), γ -glutamylcysteine (**C**), cysteine (**D**), glutathione (**E**), hypotaurine (**F**) and taurine (**G**). For **B-G**, data are presented as mean \pm SD and $N = 5$ mice. N.D., not detected. **H**, Immunoblots of xCT, CDO1, CSAD, ADO, FMO1, GCLC, GCLM, and GSS for each tissue. HSP90 was used for the loading control. α KB, α -ketobutyrate; Cth, cystathionine; Cys, cysteine; Cys_2 , cystine; γ -Glu-Cys, γ -glutamylcysteine; Glut, glutamate; Gly, glycine; GSH, glutathione; Hcy, homocysteine; Htau, hypotaurine; Tau, taurine. (**A**, Created with BioRender.com.)

Figure 6. Cysteine is a major contributor to the cysteine pool in tumors. **A**, Analysis of the fraction labeling in cysteine, γ -glutamylcysteine, and glutathione in liver tissues ($N=9$), HCC tumors ($N=9$), lung tissues ($N=10$), Nrf2^{WT} LUAD tumors ($N=16$), Nrf2^{D29H} LUAD tumors ($N=10$), pancreas tissues ($N=3$), PDAC tumors ($N=12$), and their matched serum from normal control mice for HCC ($N=6$), HCC ($N=7$), normal control mice for PDAC ($N=3$), PDAC ($N=6$), normal control mice for LUAD ($N=5$), Nrf2^{WT} LUAD ($N=8$), and Nrf2^{D29H} LUAD serum ($N=5$) following infusion with $^{13}\text{C}_6$ -cysteine. **B**, Total signal of glutathione in the tissues from **A**. **C**, Total signal of cysteine in the tissues from **A**. For **A-C**, data are presented as mean \pm SD. N.D., not detected. **D**, Immunoblots of xCT, GCLC, GCLM, and GSS for each tissue. HSP90 was used for the loading control. *, $P < 0.05$; **, $P < 0.01$; ***, $P < 0.001$; ****, $P < 0.0001$. Cys, cysteine; GSH, glutathione; γ -Glu-Cys, γ -glutamylcysteine.

9. One major outstanding question is that although the in vitro findings show GSH can be broken down to fuel intracellular metabolism of cysteine in vitro in the absence of de novo GSH metabolism, this mechanism is not yet fully supported in vivo. Based on the authors hypothesis, Gclc^{-/-} tumors, or BSO combinatorial treatments with GGsTOP, should have increased sensitivity to GGsTop because of the increased dependency on GSH catabolism. Is this the case?

This is an excellent suggestion by the Reviewer. We have performed experiments with human orthotopic breast tumor xenografts in mice treated with vehicle, GGsTop, BSO, or GGsTop and BSO combination. We found that BSO did not increase the sensitivity of tumors to GGsTop (**Extended Data Fig. 10r**). Surprisingly, it appeared that BSO dampened the anti-tumor effect of GGsTop. We profiled the efficacy of GGsTop (alone or in combination with BSO) in blocking GGT activity in the kidney and tumors. We found that GGsTop was less effective at blocking GGT activity in the kidney (but not tumors) when combined with BSO (**Extended Data Fig. 10s-10t**). This suggests that BSO potentially diminishes GGsTop's anti-tumor effect by lowering GGsTop efficacy at blocking GGT. Since BSO treatment lowers GSH levels, it may be inducing a stress response and increased

expression of GGTs; however, further investigation is required. Further, examining additional rationale combinations with GGsTop, such as combinations with IKE (to block cystine uptake), PPG (to block cysteine generation from the transsulfuration pathway), and cysteinase (to degrade cysteine/cystine), is required. **Extended Data Fig. 10** and their legend are shown below. In the Results, we state:

“We hypothesized that combining GGsTop with BSO (an inhibitor of GSH synthesis) would synergize in their anti-tumor effect. Surprisingly, this was not the case, and GGsTop appeared to be less effective at blocking tumor growth when combined with BSO (Extended Data Fig. 10r). The dampening of GGsTop’s anti-tumor effect was potentially due to the combination of GGsTop and BSO being less effective at inhibiting GGT activity (Extended Data Fig. 10s-10t). Further studies are required to examine the combination of GGsTop and BSO, along with additional rational combinations with GGsTop, such as IKE (to block cystine uptake)³⁰, PPG (to block cysteine generation from the transsulfuration pathway)³¹, and cysteinase (to degrade cysteine/cystine)³². Together, these findings suggest that GGT activity maintains GSH catabolism to supply cysteine and support tumor growth. Furthermore, the data indicate that blocking GGT is a potential therapeutic strategy for patients with cancer.”

Extended Data Figure 10

Extended Data Figure 10. GGsTop efficiently inhibits GGT activity in animals without causing toxicity. **a**, Structure of putative inhibitors of GGT. **b**, GGT activity in untreated C57BL/6 mouse tissues ($n \geq 3$ for each tissue). **c**, mRNA expression of GGT isoforms in 22 adult and 8 embryonic mouse tissues from publicly available datasets²². **d**, GGT activity in the kidney of mice 24 hours after a single i.p. injection of GGsTop at the indicated concentrations ($n = 5$ per group). **e-f**, ALT and AST liver damage serum markers ($n=6$ for Control, $n=9$ for GGsTop) (**e**) and percent change in body weight ($n=10$ per group) (**f**) in mice treated bi-daily with vehicle (saline) or 5 mg/kg i.p. GGsTop for 8 days. **g-i**, Tumor metabolites from mice with orthotopically implanted HCC-1806 cells treated with bi-daily i.p. injections of vehicle or 5 mg/kg GGsTop ($n=9$ for Control, $n=10$ for GGsTop). **j-o**, Serum and tumor metabolites from mice with orthotopically implanted HCC-1806 cells treated with the vehicle, *n*-acetyl-cysteine (30 mM in the drinking water), GGsTop (189 μ M in the drinking water), or NAC and GGsTop (Tumors; $n=8$ for vehicle, $n=7$ for vehicle+NAC, $n=9$ for GGsTop, $n=9$ for GGsTop+NAC. Serum; $n=9$ for vehicle, $n=7$ for vehicle+NAC, $n=6$ for GGsTop, $n=9$ for GGsTop+NAC). **p**, GGsTop aliquots (25 mg/mL in H₂O) were kept at room temperature for the indicated number of days, after which their inhibitory potency was assessed using an *in vitro* activity assay with mouse kidney lysates ($n=3$ technical replicates). **q**, GGT activity in the kidney of mice 72 hours after delivery of GGsTop in the drinking water at the indicated concentrations ($n=7-10$ animals per group). **r-t**, Tumor volume over time (**r**), and GGT activity in the kidney (**s**) and tumor (**t**) from mice with orthotopically implanted HCC-1806 cells treated with the vehicle, GGsTop (189 μ M in the drinking water), buthionine sulfoximine (BSO, 20 mM in the drinking water), or GGsTop and BSO ($n=8-10$ animals per group). Statistical significance in **d,j-o** was assessed by one-way ANOVA with Tukey's multiple comparisons test, while in **e,g-i,s,t** it was assessed by unpaired t-test. Data represented as mean \pm s.e.m., **p*-value<0.05; ***p*-value<0.01; ****p*-value<0.001; *****p*-value<0.0001; ns, not significant.

Minor comments

1. Where possible please show GSH, GSSG, and GSH/GSSG ratios throughout where GSH measurements are provided (as in Fig. 1c, Fig. 1f)

We have provided the relative abundance of GSSG levels for Gclc WT and KO tumors (**Extended Data Fig. 1g**). We did not conduct quantitative metabolomics for MMTV-PyMT Gclc WT and KO breast tumor samples and were unable to provide GSH/GSSG ratios. For serum and TIF analysis of patients with breast cancer, we were unable to perform the NEM-derivatization required to distinguish the GSH and GSSG pools. Thus, we have reported "Total GSH" which is a summation of GSH and 2*GSSG pools from quantitative metabolomics. **Extended Data Fig. 1** and its legend are shown below:

Extended Data Figure 1

Extended Data Figure 1. Gclc is expressed in non-epithelial cells, and its loss in tumors does cause redox or metabolic alterations. **a**, Single-nucleus RNAseq (snRNAseq) of *Gclc* transcripts from the indicated cells present in human mammary tissue. **b-e**, Autochthonous tumors from MMTV-PyMT *Gclc*^{fl/fl} *Rosa26-CreERT2* mice were excised and orthotopically transplanted into mammary fat pads of wild-type C57BL/6 mice, which were treated with vehicle (corn oil; WT) or 50 mg/kg tamoxifen for 5 days (KO). Representative immunohistochemistry (**b**) and immunofluorescence (**c**) staining of indicated proteins in *Gclc* WT and *Gclc* KO mammary tumors. Relative mRNA expression of indicated genes (**d**) and representative immunohistochemistry of indicated proteins (**e**) in *Gclc* WT and *Gclc* KO tumors. Statistical significance was assessed by an unpaired two-tailed t-test in (**d**) ($n = 6$ representative of $n = 3$ independent experiments). **f-j**, Volcano plot of metabolic differences (**f**) and levels of GSSG (**g**), ophthalmic acid (**h**), glutamate (**i**), NEM-cysteine (**j**) from *Gclc* WT and *Gclc* KO tumors. Statistical significance was assessed by an unpaired two-tailed t-test in (**g-j**) ($n = 6$ representative of $n = 3$ independent experiments). Data represented as mean \pm s.e.m., * p -value <0.05 ; ** p -value <0.01 ; *** p -value <0.001 ; **** p -value <0.0001 ; ns, not significant.

2. It is unclear if the *in vivo* experiments in Figure 1 represent only 1 independent experiment, please clarify and justify as two why a second independent replication is not provided.

In addition to the *in vivo* experiments previously shown in Figure 1, we have repeated this experiment twice more using distinct autochthonous tumor chunks MMTV-PyMT *Gclc^{fl/fl}* Rosa26-CreERT2 mice. In experiment 1, there were WT (n=6) and KO (n=6); in experiment 2, there were WT (n=5) and KO (n=6). The data provided in **Fig. 1d-1e** represent volume over time (**d**) and mass at endpoint (**e**) of WT tumors (n=11) and KO tumors (n=12) from two independent experiments. **Fig. 1** and its legend are shown below.

Figure 1

Figure 1. Intracellular production of GSH is dispensable for tumor growth. **a**, Schematic of the tumor-specific *Gclc* knockout mouse model. Autochthonous tumors from MMTV-PyMT *Gclc^{fl/fl}* Rosa26-CreERT2 mice were excised and orthotopically transplanted into mammary fat pads of wild-type C57BL/6 mice. C57BL/6 mice were treated with vehicle (corn oil; WT) or 50 mg/kg tamoxifen for 5 days (KO). **b**, Relative *Gclc* mRNA levels of WT and KO tumors. Statistical significance assessed by an unpaired two-tailed t-test (n = 5 representative of n = 3 independent experiments). **c**, GSH levels of WT and KO tumors. Statistical significance assessed by an unpaired two-tailed t-test (n = 6 representative of n = 3 independent experiments). **d-e**, Volume over time (**d**) and mass at endpoint (**e**) of WT tumors (n = 11) and KO tumors (n = 12) (representative of n = 3 independent experiments). Statistical significance was assessed by two-way ANOVA followed by Šidák's multiple comparisons test for tumor volume and by an unpaired two-tailed t-test for tumor mass. **f-g**, Immunoblot of *Gclc* protein levels (**f**) and GSH levels (**g**) in HCC-1806 human breast cancer cells infected with lentiCRISPR v2 containing non-targeting guides (sgNTC) and guides against GCLC (sgGclc 1 and sgGclc 2). Statistical significance was assessed by ordinary one-way ANOVA with Tukey's multiple comparisons test. **h-i**, Volume

over time (**h**) and mass at endpoint (**i**) of orthotopically implanted HCC-1806 sgNTC, sgGclc 1, and sgGclc 2 breast cancer cells. Statistical significance was assessed by ordinary one-way ANOVA with Tukey's multiple comparisons test for volume and by two-way ANOVA followed by Šídák's multiple comparisons test for tumor mass (n = 7 representative of n = 2 independent experiments). **j**, Concentration of total GSH (tGSH; GSH + 2×GSSG) in serum (n = 11) and tumor interstitial fluid (n = 4) from Gclc WT MMTV-PyMT autochthonous tumors from (**a**). Statistical significance was assessed by an unpaired two-tailed t-test. **k**, Concentration of tGSH in serum (n = 8) and tumor interstitial fluid (n = 6) from patients with breast cancer. Statistical significance was assessed by an unpaired two-tailed t-test. Data represented as mean ± s.e.m., ns, not significant; *p-value<0.05; **p-value<0.01; ***p-value<0.001; ****p-value<0.0001.

3. Please also show the levels of glutamate for both Figure 1f and ED Data Fig 1

We have analyzed glutamate levels in the serum and TIF from human patients with breast cancer (current **Fig. 1k**) and found that glutamate levels are enriched in TIF compared to serum (**Extended Data Fig. 3l**). Additionally, we see elevated glutamate in TIF compared to serum from murine PDAC tumors and human renal cell carcinomas (**Extended Data Fig. 3m-3n**). This is interesting since glutamate can counteract xCT-mediated cystine uptake. **Extended Data Fig. 3** and its legend are shown below.

Extended Data Figure 3

Extended Data Figure 3. Total GSH (tGSH) is enriched in the TIF compared to serum across multiple cancer models. **a-c**, Concentration of tGSH and cystine in plasma ($n = 17$) and tumor interstitial fluid ($n = 7$) from the LSL-KrasG12D/+ Trp53f/f Pdx-1-Cre (KP-/-C) mice² bearing pancreatic ductal adenocarcinoma (PDAC) (**a**), in the plasma ($n = 27$) and tumor interstitial fluid ($n = 46$) from renal cell carcinoma (RCC) human patients¹ (**b**), in cell culture media formulations (**c**). **d-i**, Autochthonous tumors from MMTV-PyMT Gclc^{fl/fl} Rosa26-CreERT2 mice were excised and orthotopically transplanted into the mammary gland or subcutaneously transplanted into the

flank of wild-type C57BL/6 mice, which were treated with vehicle (corn oil; WT) or 50 mg/kg tamoxifen for 5 days (KO). tGSH and cystine levels in TIF from orthotopic (WT, n = 4; KO, n = 6) **(d)** or flank (WT, n = 4; KO, n = 3) **(e)** tumors. tGSH in TIF from orthotopic and flank Gclc WT MMTV-PyMT tumors (n = 4) **(f)**. tGSH and cystine levels in serum of mice with orthotopic (WT, n = 11; KO, n = 12) **(g)** or flank (WT, n = 4; KO, n = 5) **(h)** tumors. Serum tGSH and cystine in serum of mice without Gclc WT MMTV-PyMT tumors (n = 8) **(i)**. **j-k**, Serum tGSH **(j)** and cystine **(k)** levels in mice bearing orthotopic (n = 11), flank (n = 4), or no Gclc WT tumors (n = 8). **l-n**, Glutamate levels in plasma and TIF from **Fig. 1k (l)** and from **a-b (m-n)**. Total GSH (tGSH) was calculated as the sum of reduced glutathione (GSH) and twice the amount of oxidized glutathione (GSSG). Statistical significance was assessed by an unpaired two-tailed t-test. Data represented as mean \pm s.e.m., *p-value<0.05; **p-value<0.01; ***p-value<0.001; ****p-value<0.0001; ns, not significant.

4. Please conduct the experiments shown in ED Fig. 2e also using NAC to distinguish between gclc-associated affects versus cysteine replenishment effects to support the interpretation of the data with differences in cell survival versus proliferation with ferrostatin-1.

We have conducted the experiments using cystine-free/NAC-supplemented conditions with vehicle or BSO treatments to block Gclc and GSH synthesis. Additionally, we have included the chiral isoforms L-NAC and D-NAC. While only L-NAC contributes to cysteine supply for proteins and metabolites³³ (Pedre B, Pharmacology & Therapeutics, 2021, PMID: 34171332), both L-NAC and D-NAC can act as cofactors for GPX4-mediated redox buffering³⁴ (Zheng J, Cell Chemical Biology, 2025, PMID: 40311609). We find that L-NAC, but not D-NAC, can rescue cancer cells in cystine-free conditions, with or without Gclc inhibition by BSO (**Extended Data Fig. 4h**). This means that, upon cystine-depletion, both maintenance of proteogenic/metabolic demands and redox buffering are required to sustain cancer cell survival. **Extended Data Fig. 4** and its legend are shown below.

Extended Data Figure 4

Extended Data Figure 4. Supplementation with GSH or cysteinylglycine rescues cancer cells in cystine-free conditions. **a**, Cancer cell lines were grown in control (208 μ M cystine; Cys₂), cystine-free (Cys₂-free), cystine-free/GSH-supplemented (750 μ M) or cystine-free/CysGly-supplemented (750 μ M) medium, and relative cell numbers were determined ($n = 6-12$ technical replicates from $n = 3-4$ independent experiments). **b-d**, HCC-1806 cells were grown in control media or cystine-free ($n = 8$ technical replicates from $n = 2$ independent experiments) (**b**), serine- and glycine-free ($n = 8$ technical replicates from $n = 2$ independent experiments) (**c**), glutamine- and glutamate-free ($n = 11$ technical replicates from $n = 3$ independent experiments) (**d**) media alone or supplemented with GSH (750 μ M) for 72 hours, and relative cell numbers were determined. Statistical significance was assessed by one-way ANOVA with Tukey's multiple comparisons test. **e**, HCC-1806 cells were

grown in control media alone or supplemented with GSH (750 μ M) and treated with DMSO or CB-839 (5 μ M) for 72 hours, and relative cell numbers were determined. Statistical significance was assessed by two-way ANOVA followed by Šídák's multiple comparisons (n = 12 technical replicates). **f**, HCC-1806 cells were untreated or treated with BSO (100 μ M) for 24 hours, and relative GSH levels were determined. Statistical significance was assessed by unpaired two-tailed t-test (n = 13-17 from n = 3 independent experiments). **g**, HCC-1806 cells were grown in control (208 μ M cystine; Cys₂), cystine-free (Cys₂-free), or cystine-free/GSH-supplemented (750 μ M) medium, untreated or treated with BSO (100 μ M) for 72 hours, and relative cell numbers were determined. Statistical significance was assessed by two-way ANOVA followed by Šídák's multiple comparisons (n = 6 technical replicates). **h**, HCC-1806 cells were grown in control (208 μ M cystine; Cys₂), cystine-free (Cys₂-free), cystine-free/L-NAC-supplemented (400 μ M), cystine-free/D-NAC-supplemented (400 μ M) medium, untreated or treated with BSO (100 μ M) for 72 hours, and relative cell numbers were determined. Statistical significance was assessed by two-way ANOVA followed by Šídák's multiple comparisons (n = 4 technical replicates).

5. Please describe the data and the data interpretation more extensively for the panels in Extended Data 3 and 4 in the manuscript text (currently one sentence description in the text). There are a few additional panels (such as in Fig. 5, among others) that are not directly described in the manuscript text.

We have updated the manuscript to include additional description of the data and the data interpretations for **Extended Data Fig. 8** (previously Extended Data Fig. 3 and Extended Data Fig. 4). **Extended Data Fig. 8** and its legend are shown below. Additionally, in the Results section, we state:

“Several enzymes possess gamma-glutamyl transferase activity¹⁰, with GGT1 being the most catalytically active isoform¹¹. We took three separate approaches to examine the necessity of GGT1 in sustaining GSH catabolism and cysteine supply to cancer cells. We generated GGT1 knockdown cell lines using CRISPRi approaches and GGT1 knockout cell lines (polyclonal and single-cell clones) using CRISPR-Cas9 approaches. GGT1 knockdown cell lines had minimal GGT1 mRNA (Extended Data Fig. 8a) but still retained significant GGT activity (Extended Data Fig. 8b) and grew similarly to control cell lines in cystine-replete and cystine-depleted/GSH-supplemented conditions (Extended Data Fig. 8c). GGT1 knockout cell lines showed decreased GGT activity (especially in GGT1 KO single-cell clones); however, in both models, we observed a rescue of cancer cells under cystine-free/GSH-supplemented conditions (Extended Data Fig. 8d-8g). Cancer cells express multiple GGT isoforms (Extended Data Fig. 9a-9d), but none of these isoforms display dependency in cancer cell line genetic screens (Extended Data Fig. 9e). Expression of GGT isoforms varies across tumor subtypes, in most cases with increased expression in tumors compared to normal tissue (Extended Data Fig. 9f-9g); however, GGT isoforms are rarely found mutated in cancers (Extended Data Fig. 9h). Additionally, expression of GGT1 in murine tumors was not dependent on GCLC expression or site of tumor growth (Extended Data Fig. 9i). Further studies are required to better understand the interplay of GGT1, other GGT family members, and uncharacterized proteins with GGT activity in cancer cells.”

Extended Data Figure 8

Extended Data Figure 8. Reduction or deletion of GGT1 expression does not impair GSH-mediated rescue of cells in cystine-free conditions. **a-b**, HCC-1806 cells were transduced with CRISPRi lentiviral vector containing sgRNA guides targeting the GGT1 promoter. GGT1 mRNA (**a**) (n = 2 technical replicates) and GGT activity (**b**) (n = 8 technical replicates from n = 3 independent experiments) in control (208 μM cystine) medium were determined. **c**, GGT1 knockdown cells from (**a-b**) were grown in control (208 μM cystine; Cys₂), cystine-free (Cys₂-free), or cystine-free/GSH-supplemented (750 μM) medium for the indicated timepoints, and cell numbers were determined. **d-e**, HCC-1806 cells were transduced with CRISPR-Cas9 lentiviral vector containing sgRNA guides targeting GGT1 and maintained as a polyclonal population. GGT activity (**d**) (n = 6 technical replicates from n = 3 independent experiments) and cell numbers (**e**) (n = 3-6 technical replicates from n = 3 independent experiments) for cells grown in control (208 μM cystine; Cys₂), cystine-free (Cys₂-free), or cystine-free/GSH-supplemented (750 μM) medium for the indicated time points were determined. (**f-g**) Single-cell clones were established from the polyclonal population, and GGT activity (**f**) (n = 3 technical replicates from n = 2 independent experiments) and cell numbers (**g**) (n = 4-6 technical replicates from n = 2 independent experiments) for cells grown in control (208 μM cystine; Cys₂), cystine-free (Cys₂-free), or cystine-free/GSH-supplemented (750 μM) medium for the indicated timepoints were determined.

6. Take care to ensure appropriate replications of experiments are completed (as in Extended Data Fig. 6b, which appears to be 1 replication)

We have ensured that the appropriate replications of experiments have been completed for all experiments, including **Fig. 3g** (previously Extended Data Fig. 6b). **Fig. 3** and its legend are shown below.

Figure 3

Figure 3. GGT1 is sufficient to promote GSH catabolism and tumor growth. **a-b**, Immunoblot analysis of human GGT1 (**a**) and GGT activity (**b**) in wild-type (WT) and GGT1-overexpressing (GGT1⁺) PC3 prostate cancer cells. Statistical significance was assessed in (**b**) by an unpaired two-tailed t-test ($n = 3$ technical replicates from $n = 2$ independent experiments). **c**, GGT histochemical stain (GMNA method) of wild-type (WT) and GGT1-overexpressing (GGT1⁺) PC3 prostate cancer. Serine-Borate was used to competitively inhibit GGT (negative control). **d**, WT and GGT1⁺ PC3 cells were grown in control media (208 μ M cystine) or cystine-free media supplemented with indicated GSH concentrations. After 72 hours, cell numbers were quantified, and the percentage of rescue ((cell numbers in indicated media – cell numbers in cystine-free media)/(cell numbers in control media) was determined ($n = 6$ technical replicates from $n = 3$ independent experiments). **e**, Schematic of non-contact co-culture experiments using 0.4 μ m PET membrane transwell inserts in media containing low concentrations of GSH (250 μ M), which were insufficient to rescue the growth of WT cells in cystine-depleted conditions. **f**, Relative cell numbers of WT, GGT1⁺, or WT cells co-cultured with GGT1⁺ cells in control, cystine-depleted, or cystine-depleted/GSH-supplemented (250 μ M) conditions. Statistical significance was assessed by two-way ANOVA with Tukey's multiple comparisons test ($n = 8$ technical replicates from $n = 4$ independent experiments). **g**, Fold change in cell numbers at indicated time points for WT and GGT1⁺ cells grown in medium containing 200 μ M cystine/0 μ M GSH (left) or 0 μ M cystine/250 μ M GSH (right). Statistical significance was assessed by two-way ANOVA ($n = 7$ technical replicates from $n = 3$ independent experiments). **h-i**, Volume over time (**h**) and mass at endpoint (**i**) of xenograft tumors from WT ($n = 26$) and GGT1⁺ ($n = 28$) PC3 cells. Statistical significance was assessed by two-way ANOVA for (**h**) and an unpaired two-tailed t-test for (**i**) ($n = 2$ independent experiments). Data represented as mean \pm s.e.m., *** p -value <0.001 ; **** p -value <0.0001 ; ns, not significant.

7. In Fig. 3f, the statistical bars are obscuring the data points

We have corrected the graph in **Fig. 3f** so that the statistical bars are not obscuring the data points. **Fig. 3** and its legend are shown below.

Figure 3

Figure 3. GGT1 is sufficient to promote GSH catabolism and tumor growth. **a-b**, Immunoblot analysis of human GGT1 (**a**) and GGT activity (**b**) in wild-type (WT) and GGT1-overexpressing (GGT1⁺) PC3 prostate cancer cells. Statistical significance was assessed in (**b**) by an unpaired two-tailed t-test ($n = 3$ technical replicates from $n = 2$ independent experiments). **c**, GGT histochemical stain (GMNA method) of wild-type (WT) and GGT1-overexpressing (GGT1⁺) PC3 prostate cancer. Serine-Borate was used to competitively inhibit GGT (negative control). **d**, WT and GGT1⁺ PC3 cells were grown in control media (208 μ M cystine) or cystine-free media supplemented with indicated GSH concentrations. After 72 hours, cell numbers were quantified, and the percentage of rescue ((cell numbers in indicated media – cell numbers in cystine-free media)/(cell numbers in control media) was determined ($n = 6$ technical replicates from $n = 3$ independent experiments). **e**, Schematic of non-contact co-culture experiments using 0.4 μ m PET membrane transwell inserts in media containing low concentrations of GSH (250 μ M), which were insufficient to rescue the growth of WT cells in cystine-depleted conditions. **f**, Relative cell numbers of WT, GGT1⁺, or WT cells co-cultured with GGT1⁺ cells in control, cystine-depleted, or cystine-depleted/GSH-supplemented (250 μ M) conditions. Statistical significance was assessed by two-way ANOVA with Tukey's multiple comparisons test ($n = 8$ technical replicates from $n = 4$ independent experiments). **g**, Fold change in cell numbers at indicated time points for WT and GGT1⁺ cells grown in medium containing 200 μ M cystine/0 μ M GSH (left) or 0 μ M cystine/250 μ M GSH (right). Statistical significance was assessed by two-way ANOVA ($n = 7$ technical replicates from $n = 3$ independent experiments). **h-i**, Volume over time (**h**) and mass at endpoint (**i**) of xenograft tumors from WT ($n = 26$) and GGT1⁺ ($n = 28$) PC3 cells. Statistical

significance was assessed by two-way ANOVA for **(h)** and an unpaired two-tailed t-test for **(i)** (n = 2 independent experiments). Data represented as mean \pm s.e.m., ***p-value<0.001; ****p-value<0.0001; ns, not significant.

8. Please show tumor GSH, GSSG, GSH/GSSG, glutamate levels in Figure 5

We have included tumor GSH, GSSG, and glutamate levels from tumor-bearing mice treated with vehicle or GGsTop in **Extended Data Fig. 10g-10i**. Since quantitative metabolomics was not performed, we could not calculate GSH/GSSG levels. **Extended Data Fig. 10** and its legend are shown below.

Extended Data Figure 10

Extended Data Figure 10. GGsTop efficiently inhibits GGT activity in animals without causing toxicity. a, Structure of putative inhibitors of GGT. **b,** GGT activity in untreated C57BL/6 mouse tissues (n ≥ 3 for each

tissue). **c**, mRNA expression of GGT isoforms in 22 adult and 8 embryonic mouse tissues from publicly available datasets²². **d**, GGT activity in the kidney of mice 24 hours after a single i.p. injection of GGsTop at the indicated concentrations (n = 5 per group). **e-f**, ALT and AST liver damage serum markers (n=6 for Control, n=9 for GGsTop) (**e**) and percent change in body weight (n=10 per group) (**f**) in mice treated bi-daily with vehicle (saline) or 5 mg/kg i.p. GGsTop for 8 days. **g-i**, Tumor metabolites from mice with orthotopically implanted HCC-1806 cells treated with bi-daily i.p. injections of vehicle or 5 mg/kg GGsTop (n=9 for Control, n=10 for GGsTop). **j-o**, Serum and tumor metabolites from mice with orthotopically implanted HCC-1806 cells treated with the vehicle, n-acetyl-cysteine (30 mM in the drinking water), GGsTop (189 μ M in the drinking water), or NAC and GGsTop (Tumors; n=8 for vehicle, n=7 for vehicle+NAC, n=9 for GGsTop, n=9 for GGsTop+NAC. Serum; n=9 for vehicle, n=7 for vehicle+NAC, n=6 for GGsTop, n=9 for GGsTop+NAC). **p**, GGsTop aliquots (25 mg/mL in H₂O) were kept at room temperature for the indicated number of days, after which their inhibitory potency was assessed using an *in vitro* activity assay with mouse kidney lysates (n=3 technical replicates). **q**, GGT activity in the kidney of mice 72 hours after delivery of GGsTop in the drinking water at the indicated concentrations (n=7-10 animals per group). **r-t**, Tumor volume over time (**r**), and GGT activity in the kidney (**s**) and tumor (**t**) from mice with orthotopically implanted HCC-1806 cells treated with the vehicle, GGsTop (189 μ M in the drinking water), buthionine sulfoximine (BSO, 20 mM in the drinking water), or GGsTop and BSO (n=8-10 animals per group). Statistical significance in **d,j-o** was assessed by one-way ANOVA with Tukey's multiple comparisons test, while in **e,g-i,s,t** it was assessed by unpaired t-test. Data represented as mean \pm s.e.m., *p-value<0.05; **p-value<0.01; ***p-value<0.001; ****p-value<0.0001; ns, not significant.

References

- 1 Abbott, K. L. *et al.* Metabolite profiling of human renal cell carcinoma reveals tissue-origin dominance in nutrient availability. *Elife* **13** (2024). <https://doi.org:10.7554/eLife.95652>
- 2 Sullivan, M. R. *et al.* Quantification of microenvironmental metabolites in murine cancers reveals determinants of tumor nutrient availability. *Elife* **8** (2019). <https://doi.org:10.7554/eLife.44235>
- 3 Liebergesell, T. C. E., Murdock, E. G. & Puri, A. W. Detection of Inverse Stable Isotopic Labeling in Untargeted Metabolomic Data. *Anal Chem* **96**, 16330-16337 (2024). <https://doi.org:10.1021/acs.analchem.4c03528>
- 4 Guzelsoy, G. *et al.* Cooperative nutrient scavenging is an evolutionary advantage in cancer. *Nature* **640**, 534-542 (2025). <https://doi.org:10.1038/s41586-025-08588-w>
- 5 Meadow, M. E. *et al.* Proteome Birthdating Reveals Age-Selectivity of Protein Ubiquitination. *Mol Cell Proteomics* **23**, 100791 (2024). <https://doi.org:10.1016/j.mcpro.2024.100791>
- 6 Baltruskeviciene, E. *et al.* Changes of reduced glutathione and glutathione S-transferase levels in colorectal cancer patients undergoing treatment. *Tumori* **104**, 375-380 (2018). <https://doi.org:10.5301/tj.5000674>
- 7 Han, L., Hiratake, J., Kamiyama, A. & Sakata, K. Design, Synthesis, and Evaluation of γ -Phosphono Diester Analogues of Glutamate as Highly Potent Inhibitors and Active Site Probes of γ -Glutamyl Transpeptidase. *Biochemistry* **46**, 1432-1447 (2007). <https://doi.org:10.1021/bi061890j>
- 8 Mandal, P. K. *et al.* System x(c)- and thioredoxin reductase 1 cooperatively rescue glutathione deficiency. *J Biol Chem* **285**, 22244-22253 (2010). <https://doi.org:10.1074/jbc.M110.121327>
- 9 Pader, I. *et al.* Thioredoxin-related protein of 14 kDa is an efficient L-cystine reductase and S-denitrosylase. *Proc Natl Acad Sci U S A* **111**, 6964-6969 (2014). <https://doi.org:10.1073/pnas.1317320111>
- 10 Heisterkamp, N., Groffen, J., Warburton, D. & Sneddon, T. P. The human gamma-glutamyltransferase gene family. *Hum Genet* **123**, 321-332 (2008). <https://doi.org:10.1007/s00439-008-0487-7>
- 11 Wickham, S., West, M. B., Cook, P. F. & Hanigan, M. H. Gamma-glutamyl compounds: substrate specificity of gamma-glutamyl transpeptidase enzymes. *Anal Biochem* **414**, 208-214 (2011). <https://doi.org:10.1016/j.ab.2011.03.026>

- 12 Hanigan, M. H., Gallagher, B. C., Townsend, D. M. & Gabarra, V. Gamma-glutamyl transpeptidase accelerates tumor growth and increases the resistance of tumors to cisplatin in vivo. *Carcinogenesis* **20**, 553-559 (1999). <https://doi.org/10.1093/carcin/20.4.553>
- 13 Bansal, A. *et al.* Gamma-Glutamyltransferase 1 Promotes Clear Cell Renal Cell Carcinoma Initiation and Progression. *Mol Cancer Res* **17**, 1881-1892 (2019). <https://doi.org/10.1158/1541-7786.MCR-18-1204>
- 14 Zhang, L. *et al.* Hypersensitivity to ferroptosis in chromophobe RCC is mediated by a glutathione metabolic dependency and cystine import via solute carrier family 7 member 11. *Proc Natl Acad Sci U S A* **119**, e2122840119 (2022). <https://doi.org/10.1073/pnas.2122840119>
- 15 Priolo, C. *et al.* Impairment of gamma-glutamyl transferase 1 activity in the metabolic pathogenesis of chromophobe renal cell carcinoma. *Proc Natl Acad Sci U S A* **115**, E6274-E6282 (2018). <https://doi.org/10.1073/pnas.1710849115>
- 16 Emmert, S., Quargnali, G., Thallmair, S. & Rivera-Fuentes, P. A locally activatable sensor for robust quantification of organellar glutathione. *Nat Chem* **15**, 1415-1421 (2023). <https://doi.org/10.1038/s41557-023-01249-3>
- 17 Jiang, X. *et al.* Quantitative imaging of glutathione in live cells using a reversible reaction-based ratiometric fluorescent probe. *ACS Chem. Biol.* **10**, 864-874 (2015). <https://doi.org/10.1021/cb500986w>
- 18 Jeong, E. M. *et al.* Real-Time Monitoring of Glutathione in Living Cells Reveals that High Glutathione Levels Are Required to Maintain Stem Cell Function. *Stem Cell Reports* **10**, 600-614 (2018). <https://doi.org/10.1016/j.stemcr.2017.12.007>
- 19 Meister, A. Glutathione metabolism and its selective modification. *J. Biol. Chem.* **263**, 17205-17208 (1988).
- 20 Rouzer, C. A., Scott, W. A., Griffith, O. W., Hamill, A. L. & Cohn, Z. A. Glutathione metabolism in resting and phagocytizing peritoneal macrophages. *J Biol Chem* **257**, 2002-2008 (1982).
- 21 Sarnstrand, B. *et al.* N,N'-Diacyl-L-cystine-the disulfide dimer of N-acetylcysteine-is a potent modulator of contact sensitivity/delayed type hypersensitivity reactions in rodents. *J Pharmacol Exp Ther* **288**, 1174-1184 (1999).
- 22 Yue, F. *et al.* A comparative encyclopedia of DNA elements in the mouse genome. *Nature* **515**, 355-364 (2014). <https://doi.org/10.1038/nature13992>
- 23 Kobayashi, S. *et al.* Carnosine dipeptidase II (CNDP2) protects cells under cysteine insufficiency by hydrolyzing glutathione-related peptides. *Free Radic Biol Med* **174**, 12-27 (2021). <https://doi.org/10.1016/j.freeradbiomed.2021.07.036>
- 24 Schomakers, B. V. *et al.* Ophthalmic acid is a glutathione regulating tripeptide. *FEBS J* **291**, 3317-3330 (2024). <https://doi.org/10.1111/febs.17061>
- 25 Cao, J. Y. *et al.* A Genome-wide Haploid Genetic Screen Identifies Regulators of Glutathione Abundance and Ferroptosis Sensitivity. *Cell Rep* **26**, 1544-1556 e1548 (2019). <https://doi.org/10.1016/j.celrep.2019.01.043>
- 26 Loe, D. W., Almquist, K. C., Deeley, R. G. & Cole, S. P. Multidrug resistance protein (MRP)-mediated transport of leukotriene C4 and chemotherapeutic agents in membrane vesicles. Demonstration of glutathione-dependent vincristine transport. *J Biol Chem* **271**, 9675-9682 (1996). <https://doi.org/10.1074/jbc.271.16.9675>
- 27 Gekeler, V., Ise, W., Sanders, K. H., Ulrich, W. R. & Beck, J. The leukotriene LTD4 receptor antagonist MK571 specifically modulates MRP associated multidrug resistance. *Biochem Biophys Res Commun* **208**, 345-352 (1995). <https://doi.org/10.1006/bbrc.1995.1344>
- 28 Zhou, Y. *et al.* Genetic analysis of tissue glutathione concentrations and redox balance. *Free Radic Biol Med* **71**, 157-164 (2014). <https://doi.org/10.1016/j.freeradbiomed.2014.02.027>

- 29 Yoon, S. J. *et al.* Comprehensive Metabolic Tracing Reveals the Origin and Catabolism of Cysteine in Mammalian Tissues and Tumors. *Cancer Res* **83**, 1426-1442 (2023). <https://doi.org:10.1158/0008-5472.CAN-22-3000>
- 30 Zhang, Y. *et al.* Imidazole Ketone Erastin Induces Ferroptosis and Slows Tumor Growth in a Mouse Lymphoma Model. *Cell Chem Biol* **26**, 623-633 e629 (2019). <https://doi.org:10.1016/j.chembiol.2019.01.008>
- 31 Yamauchi, T. *et al.* Epigenetic repression of de novo cysteine synthetases induces intra-cellular accumulation of cysteine in hepatocarcinoma by up-regulating the cystine uptake transporter xCT. *Cancer Metab* **12**, 23 (2024). <https://doi.org:10.1186/s40170-024-00352-4>
- 32 Cramer, S. L. *et al.* Systemic depletion of L-cyst(e)ine with cyst(e)inase increases reactive oxygen species and suppresses tumor growth. *Nat Med* **23**, 120-127 (2017). <https://doi.org:10.1038/nm.4232>
- 33 Pedre, B., Barayeu, U., Ezerina, D. & Dick, T. P. The mechanism of action of N-acetylcysteine (NAC): The emerging role of H(2)S and sulfane sulfur species. *Pharmacol Ther* **228**, 107916 (2021). <https://doi.org:10.1016/j.pharmthera.2021.107916>
- 34 Zheng, J. *et al.* N-acetyl-l-cysteine averts ferroptosis by fostering glutathione peroxidase 4. *Cell Chem Biol* **32**, 767-775 e765 (2025). <https://doi.org:10.1016/j.chembiol.2025.04.002>

Isaac S. Harris, Ph.D.
Associate Professor of Biomedical Genetics
Associate Professor of Pharmacology and Physiology

Manuscript: 2024-09-20778A

Specific responses to Reviewer comments:

Reviewer comments – Blue

Author responses – Black

Text from the manuscript – Bolded in Black

Referee #1 (Remarks to the Author):

The authors have answered my queries. I commend them for the amount of work they conducted.
liron bar-peled

We appreciate the kind words from Reviewer #1.

Referee #2 (Remarks to the Author):

The authors have done a commendable job of addressing my critiques. Overall, I find the results compelling and highly interesting. I support publication.

We appreciate the kind words from Reviewer #2.

Referee #3 (Remarks to the Author):

The authors have sufficiently addressed the raised concerns with the inclusion of new experiments and interpretations, which contribute to the improved strength of the manuscript's conclusions.

We appreciate the kind words from Reviewer #3.

Pertaining to major comment #9, the outcomes of the BSO in vivo experiments in ED Fig. 10 involve multiple unresolved and untested variables, including whether the phenotype is recapitulated in *gclc*^{-/-} lines or is a BSO-specific effect, and whether the modest increase in BSO-induced GGT activity explains the observed functional differences. As the authors appropriately note, these data cannot be interpreted conclusively at this stage. Accordingly, inclusion of this experiment is left to the authors' discretion as conclusions from these findings remain speculative and thus neither strengthen nor undermine the manuscripts' central conclusions.

We agree with Reviewer #3 and have decided to remove the experimental data from the BSO in vivo experiments in Extended Data Fig. 10 (shown below).

Extended Data Figure 10. GGsTop efficiently inhibits GGT activity in animals without causing toxicity. **a**, Structure of putative inhibitors of GGT. **b**, GGT activity in untreated C57BL/6 mouse tissues (kidney, spleen, and liver; $n = 6$ animals analyzed in 2 independent experiments. pancreas, epididymis, seminal vesicles, brain, testis, lungs, large intestine, brown adipose tissue, heart, and muscle; $n = 3$ animals analyzed in one experiment. Mammary fat pad; $n = 4$ animals analyzed in 2 independent experiments. PyMT breast tumors; $n = 2$ animals analyzed in one experiment). **c**, mRNA expression of GGT isoforms in 22 adult and 8 embryonic mouse tissues from publicly available datasets⁷³. **d**, GGT activity in the kidney of mice 24 hours after a single i.p. injection of GGsTop at the indicated concentrations ($n = 5$ animals from one experiment; **** $p < 0.0001$). **e-f**, ALT and AST liver damage serum markers ($n = 6$ for Control, $n = 9$ for GGsTop) and percent change in body weight ($n = 10$ per group) (**f**) in mice treated bi-daily with vehicle (saline) or 5 mg/kg i.p. GGsTop for 8 days. **g-i**, Tumor metabolites from mice with orthotopically implanted HCC-1806 cells treated with bi-daily i.p. injections of vehicle or 5 mg/kg GGsTop ($n = 9$ for Control, $n = 10$ for GGsTop). **j-o**, Serum and tumor metabolites from mice with orthotopically implanted HCC-1806 cells treated with the vehicle, n-acetylcysteine (30 mM in the drinking water), GGsTop (189 μ M in the drinking water), or NAC and

Isaac S. Harris, Ph.D.

Associate Professor of Biomedical Genetics

Associate Professor of Pharmacology and Physiology

GGsTop (Tumors; n = 8 for vehicle, n = 7 for vehicle+NAC, n = 9 for GGsTop, n = 9 for GGsTop+NAC. Serum; n = 9 for vehicle, n = 7 for vehicle+NAC, n = 6 for GGsTop, n = 9 for GGsTop+NAC). In **j**, *p = 0.020, ****p < 0.0001; in **m**, *p = 0.0208, **p = 0.0080. **p**, GGsTop aliquots (25 mg/mL in H₂O) were kept at room temperature for the indicated number of days, after which their inhibitory potency was assessed using an *in vitro* activity assay with mouse kidney lysates (n = 3 technical replicates). **q**, GGT activity in the kidney of mice 72 hours after delivery of GGsTop in the drinking water at the indicated concentrations (n = 7 for 0 and 189 μ M, n = 9 for 38 μ M, n = 10 for 94 μ M, ****p < 0.0001). **r-t**, Tumor volume over time (*p = 0.0254) (**r**), and GGT activity in the kidney (****p < 0.0001) (**s**) and tumor (*p = 0.0101) (**t**) from mice with orthotopically implanted HCC-1806 cells treated with the vehicle (n = 10) or GGsTop (189 μ M in the drinking water, n = 8). Statistical significance was assessed by one-way ANOVA followed with Dunnet's multiple comparisons test in **d**, **q**, by ordinary two-way ANOVA followed by Šídák's multiple comparisons test in **e**, **r**, and by ordinary two-way ANOVA followed Tukey's multiple comparisons test in **j-o**. Statistical significance was assessed by two-tailed unpaired t-test in **g-l**, **s**, **t**. Data represented as mean \pm s.e.m., *p-value < 0.05; **p-value < 0.01; ***p-value < 0.001; ****p-value < 0.0001; ns, not significant.

Within the set of new *in vivo* experiments generated to address related comments, the authors show that GGT1 expression in tumors does not appear to depend on GCLC status (new ED Fig. 9i). This result provides adequate *in vivo* context to address the original question of major comment #9 regarding GSH catabolism capacity in the absence of *de novo* GSH metabolism *in vivo*, although quantification or representation of the IHC staining across multiple independent tumors could support this conclusion more rigorously.

We agree with Reviewer #3 and have included a representation of the IHC staining for GGT1 protein across multiple independent tumors in Extended Data Fig. 9i (shown below).

Extended Data Figure 9. Multiple GGT isoforms are expressed in cancer cell lines, but none score as essential in genome-wide genetic screens. **a**, GGT gene family table showing all 13 GGT-related genes in the human genome. GGT1, 5, 6, and 7 are predicted to have enzymatic activity, but only two (GGT1,5) have been functionally characterized. **b**, mRNA expression (Expression Public 24Q2) of cancer cell lines (n = 1,517) from DepMap's Cancer Cell Line Encyclopedia (CCLE). **c**, Protein levels from cancer cell lines from DepMap's CLE (Proteomics GGT1: P19440 (n = 339), GGT5: P36269 (n = 108), GGT6: Q6P531 (n = 45), GGT7: Q9UJ14 (n = 375)). **d**, mRNA and protein levels of GGT isoforms in the cell line HCC-1806, using data from **(b)** and **(c)**. **e**, Histogram (splined curves) of gene dependency of cancer cell lines from Dependency Map (DepMap) (Public 24Q2+Score Chronos). A lower score suggests a gene is more likely to be essential in a cell line. A score of 0 indicates a non-essential gene, while -1 reflects the median for essential genes. *MTOR* and *GAPDH* are shown as examples of common essential genes. **f**, mRNA expression of GGT enzymes was analyzed in human tumors from 32 different TCGA projects. **g**, GGT1 mRNA expression levels in human tumoral and non-tumoral tissues from the Cancer Genome

Isaac S. Harris, Ph.D.

Associate Professor of Biomedical Genetics

Associate Professor of Pharmacology and Physiology

Atlas Program (TCGA) and the Genotype-Tissue Expression (GTEx) databases. Black lines represent median values. Statistical significance was analyzed by two-tailed unpaired t-test. ns, not significant; ****p-value < 0.0001; **p = 0.0029 (papillary renal cell carcinoma); *p = 0.0038 (clear cell renal cell carcinoma); ***p = 0.0005 (hepatocellular carcinoma). **h**, The mutational landscape and somatic mutation frequency (SMF) of each GGT gene was analyzed in 105,260 cancer patient samples across 226 non-redundant studies using the cBioPortal platform revealing a low mutation frequency (mean = 0.15%). **i**, Autochthonous tumors from MMTV-PyMT *Gclc/f* *Rosa26-CreERT2* mice were excised and orthotopically transplanted into the mammary gland or subcutaneously transplanted into the flank of wild-type C57BL/6 mice, which were treated with vehicle (corn oil; WT) or 50 mg/kg tamoxifen for 5 days (KO). Immunohistochemistry of GGT1 protein expression in *Gclc* WT and KO orthotopic and flank tumors (n = 4 animals representative of 2 independent experiments). Numbers displayed on the top left of each panel represent the animal identification number. Scale bar = 50 μ m.